# Minute-scale oscillatory sequences in medial entorhinal cortex

Soledad Gonzalo Cogno[1✉], Horst A. Obenhaus[1], Ane Lautrup[1], R. Irene Jacobsen[1], Claudia Clopath[2], Sebastian O. Andersson[1,4], Flavio Donato[1,3,5], May-Britt Moser[1,5✉] & Edvard I. Moser[1,5✉]

The medial entorhinal cortex (MEC) hosts many of the brain's circuit elements for spatial navigation and episodic memory, operations that require neural activity to be organized across long durations of experience[1]. Whereas location is known to be encoded by spatially tuned cell types in this brain region[2,3], little is known about how the activity of entorhinal cells is tied together over time at behaviourally relevant time scales, in the second-to-minute regime. Here we show that MEC neuronal activity has the capacity to be organized into ultraslow oscillations, with periods ranging from tens of seconds to minutes. During these oscillations, the activity is further organized into periodic sequences. Oscillatory sequences manifested while mice ran at free pace on a rotating wheel in darkness, with no change in location or running direction and no scheduled rewards. The sequences involved nearly the entire cell population, and transcended epochs of immobility. Similar sequences were not observed in neighbouring parasubiculum or in visual cortex. Ultraslow oscillatory sequences in MEC may have the potential to couple neurons and circuits across extended time scales and serve as a template for new sequence formation during navigation and episodic memory formation.

Brain function emerges from the dynamic coordination of interconnected neurons[4–7]. At sub-second time scales, cells are coordinated within and across brain regions by way of neuronal oscillations[8]. Studies have also reported oscillations at slower time scales, with frequencies lower than 0.1 Hz and periods lasting from tens of seconds to minutes (ultraslow oscillations), in individual neurons[9–11] and in local field potentials[12–14]. However, it remains unknown how pervasive these ultraslow oscillations are. Moreover, it remains to be determined whether and how they organize the activity of participating neurons in space and time across the neural circuit.

We directed our search for ultraslow oscillations to the MEC, a brain circuit that by containing many of the elements involved in navigational behaviour[1–3] and episodic memory formation[1,15], may possess mechanisms to organize neural activity at behavioural time scales, from seconds to minutes. Activity was recorded from hundreds of MEC cells at the same time using either two-photon calcium imaging or Neuropixels probes (Extended Data Fig. 1). To rule out variations in external stimuli as sources of modulation, we allowed head-fixed mice to run on a rotating wheel for 30 or 60 min, in darkness and with no scheduled rewards[16,17] (Fig. 1a and Extended Data Fig. 2a).

## Ultraslow oscillations in MEC neurons

To determine whether neural activity in MEC exhibits ultraslow oscillations, for each recorded cell we deconvolved the calcium signal and binarized the obtained signal ('calcium activity', bin size = 129 ms). For each cell, we then calculated the autocorrelation of the calcium activity and the corresponding power spectral density (PSD). Autocorrelation diagrams for stacks of cells from the same session showed vertical bands (Fig. 1b), suggesting that the calcium activity of many cells was oscillatory and oscillated at similar frequencies. Some cells had only one prominent peak in their PSD (Fig. 1c), suggesting that they were active at a fixed frequency. Other cells had several peaks, often with the higher frequencies appearing as harmonics of a fundamental frequency (Fig. 1d). In the example session in Fig. 1b, for most of the cells (72%, 348 out of 484) the frequency at which the PSD peaked (the 'primary frequency') was lower than 0.01 Hz (44% of the cells had a primary frequency within the range 0.006–0.008 Hz), and there were no cells whose PSD peaked at frequencies higher than 0.1 Hz. In the complete dataset (15 sessions over 5 mice), the oscillations were detectable in the majority of the recorded neurons (91%, 5,691 out of 6,231) but not in shuffled versions of the same data (Extended Data Fig. 3 and Methods). Although there was some variation in frequencies across sessions and mice, the primary frequency was always below 0.1 Hz (all oscillatory 5,691 cells; range of maximum frequencies across 15 sessions: 0.036–0.057 Hz).

To verify that the ultraslow oscillations manifest in spiking activity, we implanted two mice with Neuropixels 2.0 probes in the MEC (Extended Data Fig. 1d). Similar to the calcium imaging data, we observed oscillations at frequencies lower than 0.1 Hz in the majority of the units (78%, 683 out of 879 units, bin size = 120 ms; Fig. 1e,f).

[1]Kavli Institute for Systems Neuroscience and Centre for Algorithms in the Cortex, Fred Kavli Building, Norwegian University of Science and Technology, Trondheim, Norway. [2]Department of Bioengineering, Imperial College London, London, UK. [3]Biozentrum Universität Basel, Basel, Switzerland. [4]Present address: Max Planck Institute for Brain Research, Frankfurt am Main, Germany. [5]These authors contributed equally: Flavio Donato, May-Britt Moser, Edvard I. Moser. ✉e-mail: soledad.g.cogno@ntnu.no; may-britt.moser@ntnu.no; edvard.moser@ntnu.no

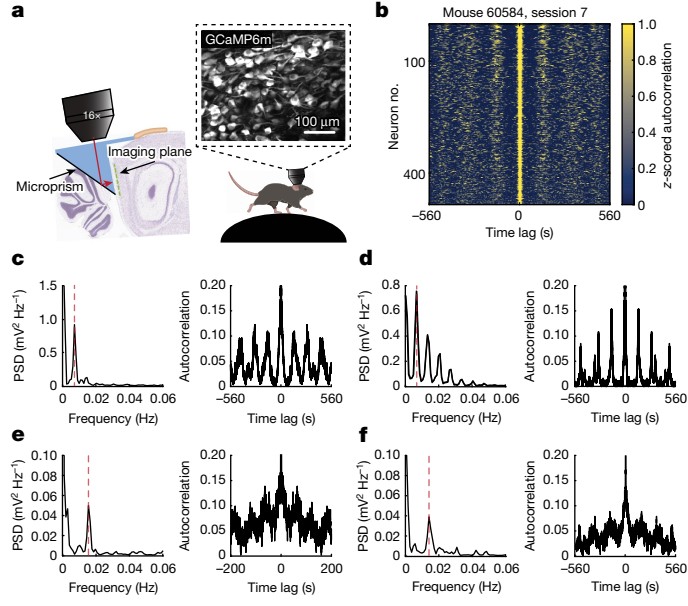

**Fig. 1 | Ultraslow oscillations in MEC neurons. a**, Neural activity was recorded through a prism from GCaMP6m-expressing neurons of the MEC in head-fixed mice running in darkness on a non-motorized wheel. Cartoon of a running mouse on the right created with BioRender.com. **b**, Stacked *z*-scored autocorrelations of single-cell calcium activity for one example session (484 neurons), plotted as a function of time lag. Neurons are sorted according to the maximum power of the PSD calculated on each autocorrelation separately, in descending order. **c**, PSD (left) calculated on the autocorrelation (right) of one example cell's calcium activity. The dashed red line indicates the frequency at which the PSD peaks (0.0066 Hz). **d**, As in **c** but for another example cell. The PSD peaks at 0.0066 Hz and has harmonics at 0.0132, 0.0207 and 0.0273 Hz. **e**,**f**, As in **c**,**d** but for two example cells recorded using Neuropixels probes. The PSDs peak at 0.016 Hz (**e**) and 0.015 Hz (**f**).

## Oscillatory sequences in MEC activity

To determine whether the ultraslow oscillations of different cells are coordinated at the neural population level, we first calculated, for the calcium imaging data, instantaneous correlations between the calcium activity of all pairs of cells. The cell pair with the highest correlation value was identified and one of the two cells was defined as the 'seed' cell. The remaining cells were sorted based on their correlation value with the seed cell, in a descending manner. Using this sorting procedure, we observed periodic sequences of neuronal activation (Fig. 2a and Extended Data Fig. 4a). The sequences unfolded successively with no interruption for tens of minutes (Fig. 2a). Because sequences of activity constitute low-dimensional dynamics, we also sorted the cells using dimensionality reduction methods, which do not depend on hyperparameters. For each recording session, we applied principal component analysis (PCA) to the matrix of calcium activity and measured, for each cell, the angle of the vector defined by the pair of loadings on principal components 1 and 2, and sorted the neurons based on these angles in a descending manner (Extended Data Fig. 4b). This sorting ('PCA method') revealed the same stereotyped periodic sequences of neuronal activation, which we hereafter refer to as oscillatory sequences; however, the sequential organization was now more salient (Fig. 2b and Extended Data Fig. 5a). When projecting the population activity onto a two-dimensional embedding, the manifold resembled a ring (Fig. 2c and Extended Data Fig. 4c). The instantaneous population activity was estimated from the position on the ring ('phase of the oscillation', Fig. 2d). The oscillatory sequences were not evident if cells were not sorted, nor if the PCA method was applied to shuffled data (Extended Data Fig. 4d). The sequences were similarly apparent when neurons

were sorted according to non-linear dimensionality reduction techniques (Extended Data Fig. 4d), as well as when the neurons were sorted using subsets of data (Extended Data Fig. 4e and Methods), and when the neurons' calcium activity was visualized using the unprocessed calcium signals (Fig. 2e).

We also observed ultraslow oscillatory sequences in the data from two mice with Neuropixels probes (469 and 410 units, respectively), indicating that our findings do not reflect factors unique to calcium imaging (Fig. 2f and Extended Data Fig. 4f,g). Some of the Neuropixels sequences were noisier than those of the calcium imaging data, possibly reflecting a broader mix of cell types located more ventrally and across several cell layers (Extended Data Fig. 1d). To maximize the number of cells recorded in layer II, and to minimize variability, we focused on calcium imaging data for the rest of the study.

Although striking oscillatory sequences were observed across multiple sessions and mice, the population activity exhibited considerable variability (Extended Data Figs. 4f,g and 5a–c). To capture this variability, we calculated an oscillation score that ranged from 0 (no oscillations) to 1 (oscillations throughout the session). The distribution of scores in the calcium imaging data was bimodal (Extended Data Fig. 5d), with oscillatory sequences showing up in 15 sessions (Extended Data Fig. 5a). All Neuropixels sessions were classified as oscillatory (Fig. 2f and Extended Data Fig. 4f,g). For each oscillatory session, we identified all sequences (Extended Data Fig. 6a–c) and found that sequence durations ranged from tens of seconds to minutes (Fig. 2g), with high variability across sessions and mice but little variability within individual sessions (Extended Data Fig. 6d–g). Inter-sequence intervals (ISI) were similarly present at different lengths, ranging from 0 s when sequences were consecutive (279 out of 406 ISIs (69%)) to a maximum of 452 s (Fig. 2h and Extended Data Fig. 6h,i).

## MEC neurons are locked to the sequences

To determine the extent to which calcium activity was tuned to the oscillatory sequences, we computed for each neuron its degree of locking to the phase of the oscillation, which ranged from 0 (no locking) to 1 (perfect locking). Significant locking degrees were observed for the vast majority of the recorded cells (Fig. 3a, left; 458 out of 484 significantly locked neurons (95%)). Results were upheld with the mutual information between calcium events and phase of the oscillation (Fig. 3a, right and Extended Data Fig. 7a). The predominance of phase-locked neurons was observed in all 15 oscillatory sessions (Fig. 3b, 5,841 out of 6,231 locked neurons (93.7%)). Each locked neuron exhibited a preference for activity within a narrow range of phases of the oscillation ('preferred phase', Fig. 3c and Extended Data Fig. 7b–e). Although sequences were still observed if high phase locking neurons were excluded, suggesting that sequences recruit widespread networks, the more cells that were excluded the more difficult it was to observe the sequences, indicating that the dynamics manifests more clearly at the neural population level (Extended Data Fig. 7f). Because the oscillatory sequences involve the vast majority of neurons recorded in MEC, and multiple cell types can be recorded within fields of view (FOV) of comparable size[18,19], the sequences most probably include a mixture of functional cell types such as grid and head-direction cells, with grid cells spanning more than one module.

Not all neurons participated in each individual sequence. We quantified the degree to which cells skipped sequences through a participation index (Extended Data Fig. 7g). Participation index variability was observed both within and across oscillatory sessions (Fig. 3d and Extended Data Fig. 7h).

## MEC sequences are not travelling waves

We next explored whether the oscillatory sequences in MEC could have features of travelling waves, in which the population activity moves

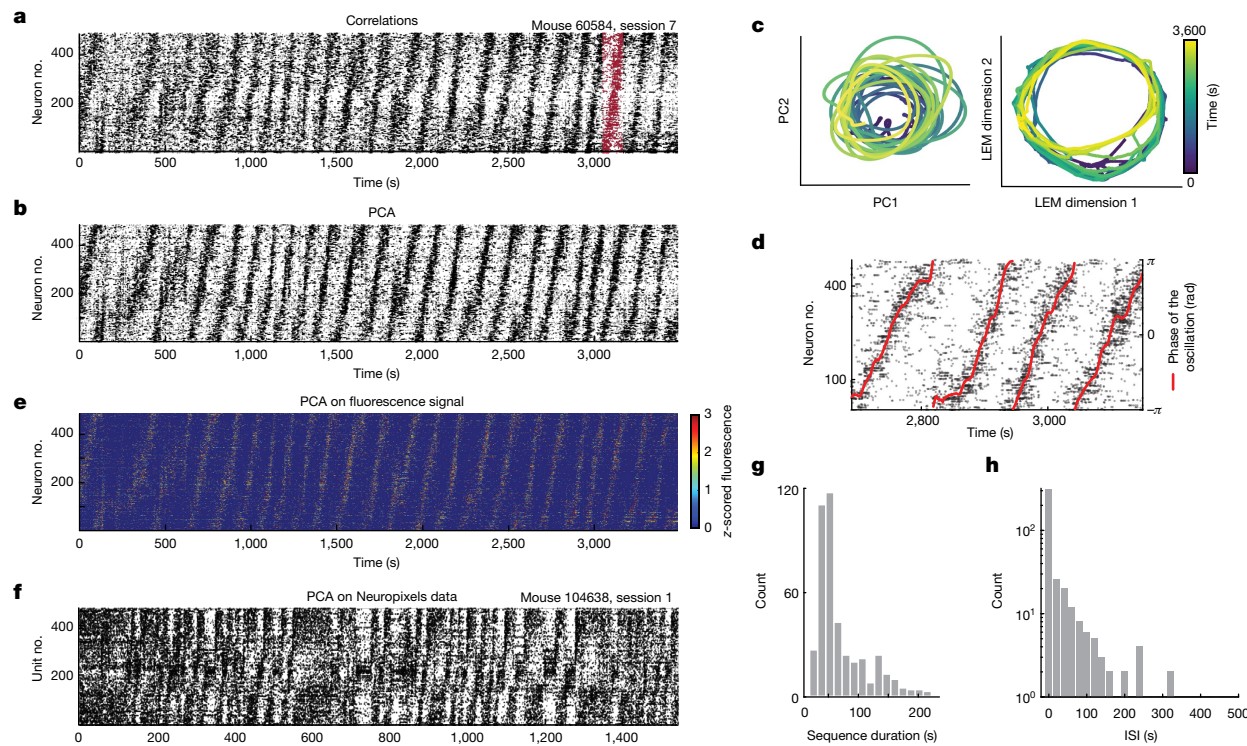

**Fig. 2 | Ultraslow oscillations are organized into oscillatory sequences.**
**a**, Raster plot of calcium activity of all cells recorded in the example session shown in Fig. 1b (bin size = 129 ms, *n* = 484 cells). Time bins with calcium events are indicated with black dots; those without calcium events are indicated with white dots. Cells were sorted according to their correlation values with one arbitrary cell, in a descending manner. The example sequence indicated in red is 121 s long. **b**, As in **a** but now with neurons sorted according to the PCA method. **c**, Projection of neural activity of the session in **a**,**b** onto the first two principal components of PCA (left), and the first two dimensions of a Laplacian eigenmaps (LEM) analysis (right). Time is colour coded. One sequence is equivalent to one rotation along the ring-shaped manifold. **d**, Raster plot as in

**b**. The phase of the oscillation, overlaid in red, was used to track the position of the population activity on the sequence. **e**, As in **b**, but showing the *z*-scored fluorescence calcium signals. **f**, Raster plot of binarized spiking activity of all units recorded in one example session using Neuropixels probes (bin size = 120 ms, *n* = 469 units). Neurons are sorted according to the PCA method. **g**, Distribution of sequence durations across 15 oscillatory sessions over 5 mice (imaging data only; one mouse did not have detectable sequences; 421 sequences in total). Each count is one sequence. **h**, Distribution of ISI (406 ISIs in total across 15 oscillatory sessions). Each count is an ISI. During periodic sequences the ISI is 0. Note that the *y* axis has a log scale.

progressively across anatomical space[20,21]. First, we found that cells with similar and dissimilar preferred phases were anatomically intermingled (Fig. 3e, Extended Data Fig. 8a and Supplementary Video 1), suggesting the absence of travelling waves with a constant direction in the propagation of activity across sequences. We next investigated the presence of travelling waves in individual sequences by calculating the preferred phase of each cell in the sequence and correlating, for all cell pairs, their difference in preferred phases with their anatomical distance (Fig. 3f). Across sequences, the correlation values were very small, ranging from −0.068 to 0.147, and below the level of statistical significance (Fig. 3g, 421 sequences across 15 oscillatory sessions over 5 mice), suggesting a lack of topographical organization (see complementary analyses in Extended Data Fig. 8b,c and Methods). In agreement with the proposed absence of travelling waves, we observed that during a single sequence, the neural activity spread across the entire FOV, and that the distance traversed by the centre of mass was similar in experimental and shuffled data (Extended Data Fig. 8d–f).

## Sequential activation of ensembles

To quantify the sequential activation of neural activity in the population, and to average out single-cell variability, we next studied ensembles of co-active cells (Extended Data Fig. 9a,b). We assigned neurons to a total of 10 ensembles, based on their proximity in the sorting obtained through the PCA method (Extended Data Fig. 9c) and

then calculated the probability by which activity transitioned between ensembles across adjacent time bins (Extended Data Fig. 9d–f), with probabilities displayed in a transition matrix (Extended Data Fig. 9g). Transitions occurred mostly between adjacent ensembles and with a preferred directionality (Extended Data Fig. 9g,h). In the oscillatory sessions the sequential activation of three or more ensembles was 2.3 times more likely in the recorded data than in shuffled data (Extended Data Fig. 9i). The probability of observing sequential activation of three or more ensembles ('sequence score') was significant in 100% of the oscillatory sessions (15 out of 15). Significant sequential activity was demonstrated also in 41% of the non-oscillatory sessions (5 out of 12, Extended Data Fig. 9j).

## Sequences do not map position

Fast oscillations and single-cell firing in the entorhinal-hippocampal system can be modulated by a number of movement-associated parameters, such as position and running state[2,3,22,23]. We next investigated whether similar dependencies are present for the minute-scale oscillatory sequences (Fig. 4a). We first calculated the probability of observing the oscillatory sequences given that the mouse was either running (mouse moves along the wheel) or immobile (position on the wheel remains unchanged) (Extended Data Fig. 2a). The oscillatory sequences were predominant during running bouts, but they were also observed during immobility (Fig. 4b). During immobility, oscillatory

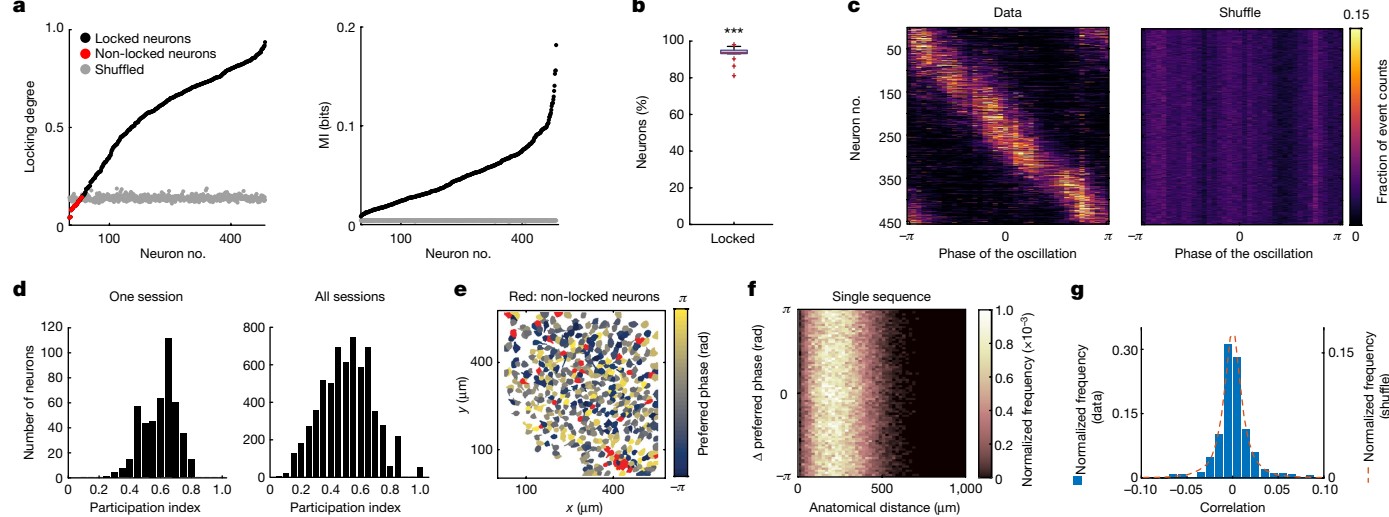

**Fig. 3 | Nearly all MEC neurons are locked to the oscillatory sequences.**
**a**, Left, locking degrees of neurons from the session shown in Fig. 2a. Black dots indicate locked neurons; red dots indicate non-locked neurons; and grey dots show the 99th percentiles of the corresponding shuffle distributions, one per cell (458 out of 484 cells were significantly locked to the phase of the oscillation). Right, similar to left, but for mutual information (MI) between phase of the oscillation and count of calcium events. Black dots indicate MI and grey dots show the estimated bias in the MI. For all cells, the MI is larger than the bias. Neurons are sorted according to ascending locking degree (left) or MI (right). **b**, Box plot showing percentage of locked neurons over all oscillatory sessions (median = 94%; two-sided Wilcoxon signed-rank test, $n = 15$ sessions, $P = 6.1 \times 10^5$, $W = 120$). Red line shows median across sessions; blue bottom and top lines delineate bottom and top quartiles, respectively; whiskers extend to 1.5 times the interquartile range; and red crosses show outliers exceeding 1.5 times the interquartile range. **c**, Each row shows the tuning curve (colour coded) to the phase of the oscillation of one locked neuron in Fig. 2a ($n = 458$) calculated on experimental (left) and shuffled (right) data. **d**, Distribution of participation indexes across neurons in the session in Fig. 2a ($n = 484$ cells, left) and across all 15 oscillatory sessions ($n = 6,231$ cells, right). **e**, Anatomical distribution of neurons in the FOV of the session in Fig. 2a. Neuronal preferred phase is colour coded. Neurons in red are not significantly locked. Dorsal MEC on top, medial on the right. **f**, A two-dimensional histogram of differences in preferred phase between pairs of neurons for sequence no. 19 of the session in Fig. 2a, and their distance in the FOV. In the presence of travelling waves, high values along the diagonal would be expected. Normalized frequency is colour coded. Each count is a cell pair ($n = 116,886$ cell pairs for 484 recorded cells). Correlation = 0.0026, cutoff for significance = 0.0099. **g**, Distribution of correlation values between differences in preferred phase and anatomical distance in experimental data (blue bars, $n = 421$ sequences across 15 oscillatory sessions) and shuffled data (orange dotted line, $n = 42,100$, 100 shuffled iterations per sequence) (Methods). ***$P < 0.001$, **$P < 0.01$, *$P < 0.05$; NS, not significant ($P > 0.05$).

sequences were continuous for durations spanning from 1 s to 258 s (Fig. 4c and Extended Data Fig. 2b). The continued presence of the oscillatory sequences during long epochs of immobility suggests that behavioural state and running distance have a limited role in driving the progression of the sequences in MEC, in contrast to previous observations in CA1 of the hippocampus[16]. In line with this result, the number of laps the mice completed on the wheel during one sequence was highly heterogeneous, ranging from 0 to 86 laps per sequence across all mice (lap length = 53.7 cm, Fig. 4d and Extended Data Fig. 2c).

Sequences took place during a wide range of speed and acceleration values (Extended Data Fig. 2d,e). Although we found no difference in speed 10 s before and after sequence onset (Extended Data Fig. 2f–j), new epochs of sequences were more likely to be initiated during running bouts (onset of sequences was 3.1 times more frequent in running bouts than in immobility bouts).

## Sequences are specific to MEC

Since ultraslow oscillations have been reported in widely different brain areas[9–14], we investigated whether the oscillatory sequences were observed in other regions too. We recorded the activity of hundreds of cells in two regions: (1) the parasubiculum (PaS), a parahippocampal region abundant with grid and head-direction cells but with a different circuit structure than MEC[24] (25 sessions over 4 mice, Extended Data Fig. 10a,b), and (2) the visual cortex (VIS), which differs from MEC[25] in its network architecture and in the high dimensionality of its neural population activity[26] (19 sessions over 3 mice, Extended Data Fig. 10c). The mice performed the same minimalistic self-paced running task as in the MEC recordings. We found that while the calcium activity of a fraction

of cells in both brain areas was ultraslow and periodic (Fig. 5a–d), in neither brain region were these oscillations organized into oscillatory sequences (Fig. 5e,f and Extended Data Fig. 11a–h), and for all sessions the oscillation scores were lower than the threshold defined from the MEC data to classify sessions as oscillatory (Extended Data Fig. 11i, threshold = 0.72) (Fig. 5g). Moreover, data from VIS were more synchronous than PaS data (Extended Data Fig. 11j,k), consistent with previous observations[17]. Finally, calcium activity was more correlated with the speed of the mouse in VIS than in MEC and PaS (Extended Data Fig. 11l), suggesting that ultraslow oscillations in VIS might reflect slow changes in the running speed of the mouse. Altogether, these results suggest that MEC has network mechanisms for sequential coordination of single-cell oscillations that are not present in PaS or VIS.

## Sequences may enable specific patterns

The ultraslow time scale of the oscillatory sequences raises questions as to their possible function. To determine whether they could serve as a scaffold—or 'template'—for the formation of new activity patterns, we developed a simple model. In this model, 500 units that fired in a sequential manner, the template, were connected to an output neuron (Extended Data Fig. 12a; the results can be generalized to more output neurons). We trained the weights of the connections to enable a specific 'target' activity pattern in the output neuron. As example targets we considered first a ramp of activity (Extended Data Fig. 12b, left), mirroring activity observed in many neurons in decision making tasks[27] or during free foraging[28], and second a less stereotyped target generated with a stochastic process (Extended Data Fig. 12b, right). The output unit could reproduce the target activity when the input sequence was

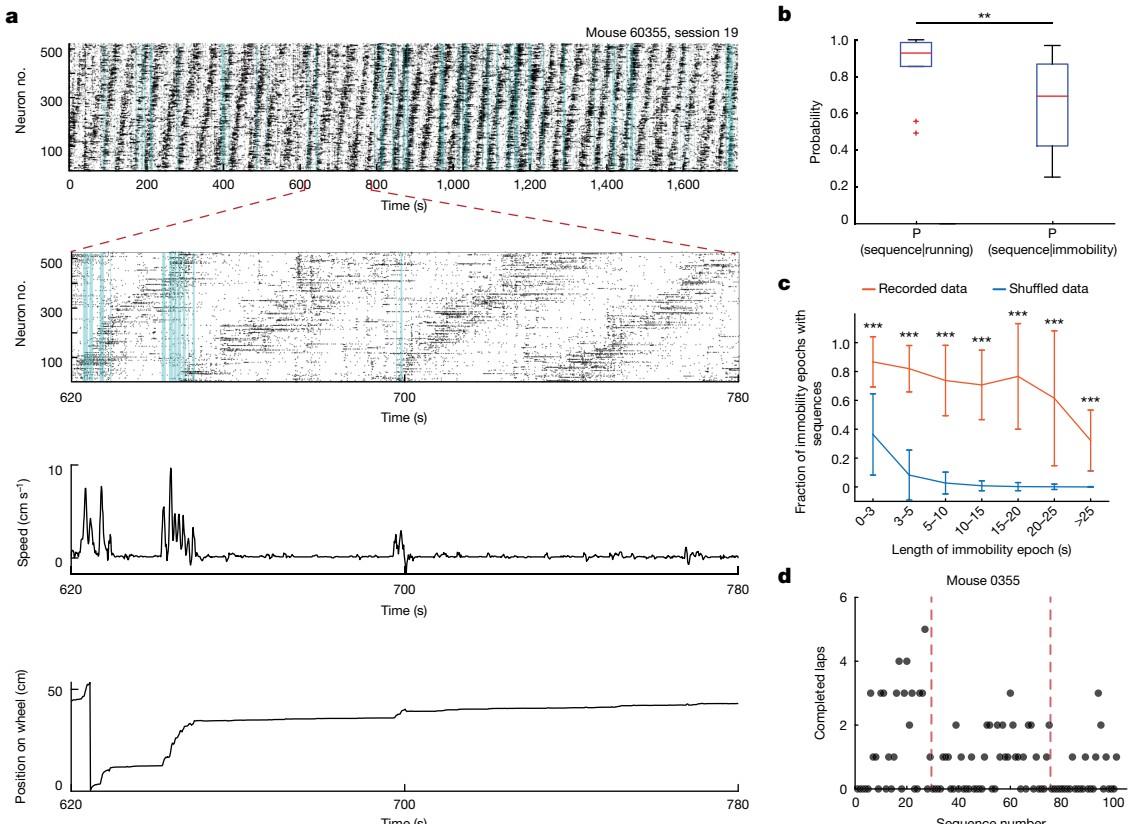

**Fig. 4 | The oscillatory sequences transcend periods of running and immobility. a**, Top, raster plot of one recorded session (520 neurons). Time bins in aquamarine indicate that the mouse ran faster than 2 cm s⁻¹. Second from top, expanded view showing 160 s of neural activity. Third from top, instantaneous speed of the mouse. Bottom, position of the mouse on the wheel. **b**, Probability of observing the oscillatory sequences given that the mouse was either running or immobile (median probability during running and immobility was 0.93 and 0.69, respectively; two-sided two sample Wilcoxon signed-rank test, $n = 10$ sessions over 3 mice, $P = 0.002$, $W = 55$). **c**, Fraction of immobility epochs with oscillatory sequences as a function of length of the immobility epoch (data are mean ± s.d.). For each length bin, the fraction of epochs was averaged across sessions. Orange, recorded data ($n = 10$ per length bin); blue: shuffled data ($n = 5,000$ per length bin, 500 shuffled realizations per session). Recorded versus shuffled data: $P \le 2.62 \times 10^{-6}$, $4.7 \le Z \le 47.5$, two-sided Wilcoxon rank-sum test. **d**, Number of completed laps as a function of sequence number for one mouse. Each dot indicates one sequence. Dashed lines indicate separation between recorded sessions.

slower or as slow as the target pattern, but not when the input sequences were faster (Extended Data Fig. 12c,d). These results suggest that neural activity patterns that unfold at behavioural time scales may only be supported by sequences that unfold at similarly slow or slower time scales—that is, over durations of many seconds or more.

## Discussion

Our experiments identify sequences of neural activity in MEC that repeat periodically during running as well as during intermittent periods of rest. Across recording sessions, the duration of individual sequences can range from tens of seconds to minutes, but the time scale is generally fixed within an individual recording session. In Neuropixels data, the sequences were somewhat noisier than in the calcium imaging data, as expected when sampling from multiple layers, across a wider dorso–ventral range, and with better capture of the fast dynamics of interneurons. The ultraslow periodic sequences observed in our data stand out from instances of slow sequential neural activity that have not been described in terms of oscillations. In the hippocampus, neural activity in CA1 cells that is organized into stereotypic sequences[29,30] is more coupled to ongoing behavioural activity and running distance than in our data[16]. Moreover, whereas nearly 94% of MEC neurons in the present study were significantly locked to the oscillatory sequences, reported hippocampal sequences involve only a small fraction of the network (5% in ref. 16). This difference in participation would be in

agreement with the view that the MEC supports a low-dimensional population code where the cells' responses covary across environments[31], whereas the hippocampus supports a more high-dimensional population code that may orthogonalize distinct experiences[32,33]. The MEC oscillatory sequences also differ from travelling waves[20,21], which move progressively through anatomical space.

The widespread nature of the ultraslow oscillatory activity in individual neurons would be consistent with a role for ascending neuromodulatory arousal-associated brain-stem circuits in controlling these oscillations[14,34,35]. In contrast to the oscillations, sequential organization of neural population activity was only present in MEC, pointing to MEC as having unique network mechanisms for sequence formation. The oscillatory sequences of the MEC are consistent with dynamics expected in a ring-shaped continuous attractor network[36,37]. However, sequential activity could also be generated in recurrently connected networks[38] or in feedforward networks through synfire chains or rate propagation[39,40], or by plasticity rules operating on slow time scales[41].

The oscillatory sequences might have a role in large-scale coordination of entorhinal circuit elements[5], either by synchronizing faster oscillatory activity, such as theta and gamma[1,4,6,8], or by organizing neural activity across functionally dissociable cell classes, such as grid and head-direction cells[2,3]. Coordination may help functional cell classes, for example different grid cell modules, keeping the same phase relationships over time, enabling a consistent readout of position or other variables represented in MEC activity[42,43]. As illustrated by our

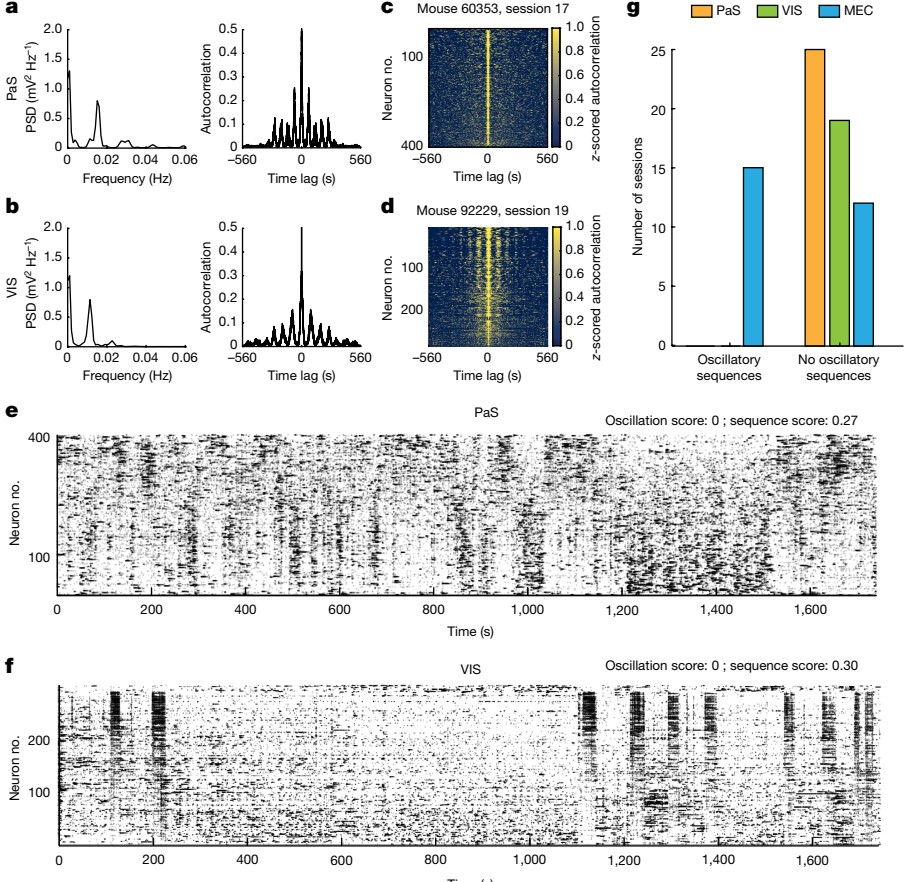

**Fig. 5 | The oscillatory sequences are not observed in PaS or VIS. a,b,** PSD (left) calculated on the autocorrelation (right) of calcium activity in one example cell recorded in PaS (**a**) or VIS (**b**). The PSDs peaked at 0.015 Hz (**a**) and 0.011 Hz (**b**). **c,d,** Stacked autocorrelations (as in Fig. 1b) for two example sessions recorded in PaS (**c**; 402 neurons) and VIS (**d**; 289 neurons). **e,f,** PCA-sorted raster plots (as in Fig. 2b) for two example sessions recorded in PaS (**a,c**) and VIS (**b,d**). Oscillation score and sequence score are indicated at the top. **g,** Number of sessions with and without oscillatory sequences in MEC (blue, 27 sessions), VIS (green, 19 sessions) and PaS (yellow, 25 sessions) based on oscillation scores and threshold defined from the MEC dataset (Extended Data Fig. 5d).

model, the oscillatory sequences may also act as a template to enable the formation of new firing patterns over long and behaviourally relevant time scales. By doing so, they may facilitate storage of memories associated with one-time experiences in downstream networks[17,44,45]. Downstream sequences may be generated via plasticity in connections from MEC, in reminiscence of sequence formation during zebra finch song learning[46]. The MEC sequences may also serve a role in temporal coding during extended behavioural experiences, by enabling the circuit to keep track of time[47,48] or by facilitating the slowly drifting neural population activity in lateral entorhinal cortex[28].

It remains an open question whether the ultraslow oscillatory sequences are present across a broader spectrum of behaviours, including sleep and free exploration, and in the presence of salient visual feedback. If so, it is possible that the sequences reset in the presence of strong landmarks or sensory stimulation and that only subpopulations of the neurons demonstrate it. The potentially richer dynamics of the periodic sequences during more natural behaviours must interface with the dynamics of MEC cells on a number of manifolds, such as in ensembles of head-direction cells and grid cells[25,49,50].

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

# Methods

All experiments were performed in accordance with the Norwegian Animal Welfare Act and the European Convention for the Protection of Vertebrate Animals used for Experimental and Other Scientific Purposes, Permit numbers 18011 and 29893.

## Subjects

Male C57/Bl6 mice were housed in social groups of 2–6 individuals per cage (calcium imaging experiments) or individually (electrophysiology experiments, after implantation). The mice had access to nesting material and a planar running wheel and were kept on a 12 h light/12 h darkness schedule in a temperature and humidity-controlled vivarium. Food and water were provided ad libitum. Two-photon calcium imaging data were collected from a cohort of 12 mice (5 implanted in MEC, 4 in PaS, and 3 in VIS). Electrophysiological data from the MEC were collected from 2 mice.

## Surgeries

For all surgeries, anaesthesia was induced by placing the subjects in a plexiglass chamber filled with isoflurane vapour (5% isoflurane in medical air, flow of 1 l min$^{-1}$). Surgery was performed on a heated surgery table (38 °C). Air flow was kept at 1 l min$^{-1}$ with 1–3% isoflurane as determined from physiological monitoring of breathing and heartbeat. The mice were allowed to recover from surgery in a heated chamber (33 °C) until they regained complete mobility and alertness. Postoperative analgesia was given in the form of subcutaneous injections of Metacam (5 mg kg$^{-1}$) 24 and 48 h after the first Metacam injection as long as was deemed necessary. Additionally, the mice were given subcutaneous injections or oral administration of Temgesic (0.05–0.1 mg kg$^{-1}$) with 6- to 8-h (injections) or 12-h (oral) intervals for the first 36 h after the first Temgesic injection.

**Surgeries for calcium imaging.** Surgeries were performed according to a two-step protocol. During the first procedure, newborn pups or adult mice were injected in MEC or PaS, or adult mice were injected in VIS with a virus carrying a construct for the expression of the calcium indicator GCaMP6m. The virus (for all injections: AAV1-Syn-GcaMP6m; titre $3.43 \times 10^{13}$ genome copies per ml, AV-1-PV2823, UPenn Vector Core, University of Pennsylvania, USA) was diluted 1:1 in sterile DPBS (1× Dulbecco's Phosphate Buffered Saline, Gibco, ThermoFisher). During the second procedure, two weeks later, a microprism was implanted to gain optical access to infected neurons located in MEC and PaS, or a glass window was inserted to obtain similar access in VIS.

**Virus injection and microprism implantation in MEC and PaS.** In the first surgical procedure, newborn pups received injections of AAV1-Syn-GCaMP6m one day after birth[51]. An analgesic was provided immediately before the surgery (Rymadil, Pfizer, 5 mg kg$^{-1}$). Pre-heated ultrasound gel (39 °C, Aquasonic 100, Parker) was generously applied on the pup's head in order to create a large medium for the transmission of ultrasound waves. Real-time ultrasound imaging (Vevo 1100 System, Fujifilm Visualsonics) allowed for targeted delivery of the viral mixture to specific areas of the brain. During ultrasound imaging, the pup was immobilized through a custom-made mouth adapter. The ultrasound probe (MS-550S) was lowered to be in close contact with the gel and thus the pup's head to allow visualization of the targeted structures. The probe was kept in place for the whole duration of the procedure via the VEVO injection mount (VEVO Imaging Station. Imaging in B-Mode, frequency: 40 MHz; power: 100%; gain: 29 dB; dynamic range: 60 dB). Target regions were identified by structural landmarks: the MEC or PaS were identified in the antero–posterior and medio–lateral axis by the appearance of the aqueduct of Sylvius and the lateral sinus. The target area for injection was comparable to a coronal section at ∼−4.7 mm from bregma in the adult mouse. The solution containing the virus

(250 ± 50 nl per injection) was injected in the target regions via beveled glass micropipettes (Origio, custom made; outer tip opening: 200 µm; inner tip opening: 50 µm) using a pressure-pulse system (Visualsonics, 5 pulses, 50 nl per pulse). The pipette tip was pushed through the brain without any incision on the skin, or a craniotomy, and, to reduce the duration of the procedure, retracted immediately after depositing the virus in the target area. The anatomical specificity of the infection was verified by imaging serial sections of the infected hemispheres after experiment completion (see 'Histology of calcium imaging mice and reconstruction of field-of-view location').

Two weeks after the viral injection, we performed a second procedure, in which a microprism was implanted in the left hemisphere to gain optical access to the superficial layers of MEC and PaS[52]. The implanted microprism was a right-angle prism with 2 mm side length and reflective enhanced aluminium coating on the hypotenuse (Tower Optical). The prism was glued to a 4-mm-diameter (CS-4R, thickness no. 1) round coverslip with UV-curable adhesive (Norland). On the day of surgery, mice were anaesthetized with isoflurane (IsoFlo, Zoetis, 5% isoflurane vapourised in medical air delivered at 0.8–1 l min$^{-1}$) after which two analgesics were provided through intraperitoneal injection (Metacam, Boehringer Ingelheim, 5 mg kg$^{-1}$ or Rimadyl, Pfizer, 5 mg kg$^{-1}$, and Temgesic, Indivior, 0.05–0.1 mg kg$^{-1}$) and one local analgesic was applied underneath the skin covering the skull (Marcain, Aspen, 1–3 mg kg$^{-1}$). Their scalp was removed with surgical scissors and the surface of the bone was dried before being generously covered with optibond (Kerr). To increase the thickness and stability of the skull and overall preparation, a thin layer of dental cement (Charisma, Kulzer) was applied on the exposed skull, except in the location above the implant, where a 4-mm-wide circular craniotomy was made. The craniotomy was positioned over the dorsal surface of the cortex and cerebellum, with the centre positioned ∼4 mm lateral from the centre of the medial sinus, and above the transverse sinus just above the MEC and PaS. After the dura was removed above the cerebellum, the lower edge of the prism was slowly pushed in the empty space between the forebrain and the cerebellum, just posterior to the transverse sinus. The edges of the coverslip were secured to the surrounding skull with UV-curable dental cement (Venus Diamond Flow, Kulzer). A custom-designed steel headbar was attached to the dorsal surface of the skull, centred upon and positioned parallel to the top face of the microprism. All exposed areas of the skull, including the headbar, were finally covered with dental cement (Paladur, Kulzer) and made opaque by adding carbon powder (Sigma Aldrich) until the dental cement powder became dark grey.

**Virus injection and glass window implantation in VIS.** In a different cohort of mice than those used for MEC/PaS imaging, we induced the expression of GCaMP6m in neurons of the adult VIS for subsequent imaging. We targeted the injection of the same AAV1-Syn-GCaMP6m viral solution used in the developing MEC and PaS to the primary visual cortex. On the day of surgery, 3- to 5-month-old mice were anaesthetized with isoflurane (IsoFlo, Zoetis, 5 % isoflurane vapourized in medical air delivered at 0.8–1 l min$^{-1}$) after which two analgesics were provided through intraperitoneal injection (Metacam, Boehringer Ingelheim, 5 mg kg$^{-1}$ or Rimadyl, Pfizer, 5 mg kg$^{-1}$, and Temgesic, Indivior, 0.05–0.1 mg kg$^{-1}$) and one local anaesthetic was applied underneath the skin covering the skull (Marcain, Aspen, 1–3 mg kg$^{-1}$). The virus was injected at three locations in VIS, all of which were within the following anatomical ranges in the right hemisphere: 2.3–2.5 mm lateral from the midline, 0.9–1.3 mm anterior from lambda[53]. At each injection site, 50 nl of the virus was injected 0.5 mm below the dura and the pipette was left in place for 3–4 min to enable the virus to diffuse. The pipette was then brought to 0.3 mm below the dura and another 50 nl was injected. The pipette was then left in place for 5–10 min before retracting it completely. The speed of the injections was 5 nl s$^{-1}$.

Two weeks after the viral injection, a surgery to chronically implant a glass window over VIS was performed. The mice were handled as

previously described for the prism surgery in MEC/PaS, including anaesthesia, delivery of analgesics, and scalp removal. Optibond was applied to the exposed skull except in the location of the craniotomy. A 4-mm-wide craniotomy was made, centred on the virus injection coordinates, and a 4-mm glass window was placed underneath the skull edges of the craniotomy. The glass was slightly larger than the craniotomy, so after it was manoeuvred in place, the upward pressure exerted by the brain secured it in place against the skull, thereby minimizing the presence of empty gaps that might favour tissue and bone regrowth. The edges of the window were secured with UV-curable dental cement and superglue before the positioning of the headbar as described for the MEC–PaS implantation. All exposed areas of the skull, including the headbar, were finally covered with dental cement (Paladur, Kulzer) that was made opaque by adding carbon powder (Sigma Aldrich) until the dental cement powder became dark grey.

**Neuropixels probe implants.** Two adult mice (4 to 5 months old) were implanted with four-shank Neuropixels 2.0 silicon probes[54] targeting the superficial layers of MEC in the left hemisphere. Prior to the surgery, the mice were given general analgesics (Metacam, Boehringer Ingelheim, 5 mg kg$^{-1}$ and Temgesic, Indivior, 0.05–0.1 mg kg$^{-1}$) subcutaneously and one local anaesthetic was applied underneath the skin covering the skull (Marcain, Aspen, 1–3 mg kg$^{-1}$). After incision, a hole was drilled over the cerebellum for an anchor screw connected to a ground wire. Craniotomies were then drilled. Probes targeting the MEC were lowered from the surface to depths between 2.5 mm and 2.7 mm relative to the dura mater. They were implanted with the most medial shank placed on the brain surface 3.2 mm lateral to the midline and 0.4 mm anterior to the transverse sinus edge. The four shanks were oriented with the electrode sites on the posterior side. In one of the two mice (no. 104638), the probe was first rotated 7° in the horizontal plane (angle with reference to the coronal plane), with the most lateral shank in the most posterior position such that the shanks were parallel to the transverse sinus. The four shanks were then lowered vertically from this position.

The Neuropixels probe of the second mouse (no. 102335) was not rotated in the horizontal plane—that is, all shanks had the same anterior–posterior coordinates. The electrode shanks of this mouse were lowered from the surface with a 2° angle relative to the coronal plane, such that the shank tips were the most posterior. The shanks remained within the same sagittal plane as they were lowered. This second mouse was also implanted with a probe targeting the CA1 region in the right hemisphere, 1.225–1.975 mm relative to the midline, at a depth of 3 mm relative to dura mater, with all shanks 2.1 mm posterior to bregma. The hippocampal data were not used in the present study. The probes were secured to the skull using an adhesive (OptiBond, Kerr), UV-curable dental cement (Venus Diamond Flow, Kulzer), and dental cement (Meliodent, Kulzer). A headbar was attached as described above for the calcium imaging studies.

### Self-paced running behaviour under sensory-minimized conditions

Training of mice began 2 days after the prism implantation in MEC and PaS, 12 days after the implantation of a cranial window in VIS, and 5–7 days after Neuropixels probe implantation. All mice used for calcium imaging recordings and one Neuropixels-implanted mouse (no. 104638) were head-restrained by a headbar with their limbs resting on a freely rotating styrofoam wheel with a metal shaft fixed through the centre. The radius of the wheel was ~85 mm and the width 70 mm. Low friction ball bearings (HK 0608, Kulelager) were affixed to the ends of the metal shaft and held in place on the optical table using a custom mount. This arrangement allowed the mice to self-regulate their movement. The position of the mouse on the rotating wheel was measured using a rotary encoder (E6B2-CWZ3E, YUMO) attached to its centre axis. Step values of the encoder (4,096 per full revolution,

~130 µm resolution) were digitized by a microcontroller (Teensy 3.5, PJRC) and recorded using custom Python scripts at 40–50 Hz. Wheel tracking was triggered at the start of imaging and synchronized to the ongoing image acquisition through a digital input from the 2-photon microscope. In a subset of mice recorded with calcium imaging (3 out of 12; 2 implanted in MEC, 1 implanted in PaS), the precise synchronization was not available to us and these data were hence not used for comparison of movement and imaging data. A T-slot photo interrupter (EE-SX672, Omron) served as a lap (full revolution) counter. Design and code of the wheel are publicly available under https://github.com/kavli-ntnu/wheel_tracker.

The other Neuropixels probe-implanted mouse (no. 102335) was head-restrained by a headbar while resting on a circular disc coated with rubber spray. The radius of this wheel was ~85 mm. The mouse was allowed self-paced movement on the wheel. Three-dimensional motion capture (OptiTrack Flex 6 cameras and Motive recording software) was used to track the rotation of the wheel by tracking retroreflective markers placed on the wheel edge. Digital pulses were generated using an Arduino microcontroller which were used to align the Neuropixels acquisition system and the OptiTrack system via direct TTL input and infra-red LEDs.

In all mice, the self-paced task was performed under conditions of minimal sensory stimulation, in darkness, and with no rewards to signal elapsed time or distance run[16,17]. Prior to the imaging sessions, the calcium imaging mice were accustomed to the setup through daily exposures over the course of between 5 and 15 sessions, one session per day. Neuropixels-implanted mice were habituated to the setup by gradually increasing the time spent on the wheel over four days. In each session, after the mice were positioned on the wheel, they were gently head-restrained and free to run or rest[55,56] for 30, 45 or 60 min.

Recording sessions of Neuropixels-implanted mice also consisted of trials where the mice were freely foraging in a 80 cm × 80 cm open field arena for 30 min. These open field trials preceded the self-paced wheel trials and were not used in the present study.

### Two-photon imaging in head-fixed mice

A custom-built 2-photon benchtop microscope (Femtonics, Hungary) was used for 2-photon imaging of the target areas (that is, superficial layers of MEC, PaS and VIS). A Ti:Sapphire laser (MaiTai Deepsee eHP DS, Spectra-Physics) tuned to a wavelength of 920 nm was used as the excitation source. Average laser power at the sample (after the objective) was 50–120 mW. Emitted GCaMP6m fluorescence was routed to a GaAsP detector through a 600 nm dichroic beamsplitter plate and 490–550 nm band-pass filter. Light was transmitted through a 16×/0.8 NA water-immersion objective (MRP07220, Nikon) carefully lowered in close contact to the coverslip glued to the microprism (for MEC–PaS imaging) or above the coverslip in contact with the brain surface (for VIS imaging). For the microprism-implanted mice, the objective lens was aligned to the ventro–lateral corner of the prism, to consistently identify the position of MEC and PaS across mice. Ultrasound gel (Aquasonic 100, Parker) or water was used to fill the gap between the objective lens and the glass coverslips. The software MESc (v 3.3 and 3.5, Femtonics, Hungary) was used for microscope control and data acquisition. Imaging time series of either ~30 min or ~60 min were acquired at 512 × 512 pixels (sampling frequency: 30.95 Hz, frame duration: ~32 ms; pixel size: either 1.78 × 1.78 µm$^2$ or 1.18 × 1.18 µm$^2$). Time series acquisition was initiated arbitrarily after the mouse was head-restrained on the setup.

### Neuropixels recordings in head-fixed mice

Signals were recorded using a Neuropixels acquisition system as described previously[25,57]. In short, the electrophysiological signal was amplified with a gain of 80, low-pass-filtered at 0.5 Hz, high-pass-filtered at 10 kHz, and digitized at 30 kHz on the probe circuit board. The digitized signal was then multiplexed by the 'headstage' circuit board and

transmitted along a 5 m tether cable using twisted pair wiring to a Neuropixels PXIe acquisition module. The data was visualized and recorded using SpikeGLX version 20201103 software (https://billkarsh.github.io/SpikeGLX).

### Histology

**Histology of calcium imaging mice and reconstruction of field-of-view location.** On the last day of imaging, after the imaging session, the mice were anaesthetized with isoflurane (IsoFlo, Zoetis) and then received an overdose of sodium pentobarbital before transcardial perfusion with freshly prepared PFA (4% in PBS). After perfusion, the brain was extracted from the skull and kept in 4% PFA overnight for post-fixation. The PFA was then exchanged with 30% sucrose to cryoprotect the tissue.

To verify the anatomical location of the imaged FOVs in the microprism-implanted mice, we used small, custom-made pins, derived from a thin piano wire coated with a solution of 1,1′-dioctadecyl-3,3,3′,3′-tetramethylindocarbocyanine perchlorate (DiI; DiIC18(3)) (ThermoFischer), to mark the location of the imaged tissue in relation to the prism footprint. A DiI-coated pin was inserted into the brain tissue at the location left empty by the prism footprint, and specifically targeted to the ventro–lateral corner of the footprint (see 'Surgeries'). The pin was left in place to favour transfer of DiI from the metal pin to the brain tissue, and to leave a fluorescent mark on the location of the imaged FOV. After 30 to 60 s, the pin was removed and the brain was sliced on a cryostat in 30–50 μm thick sagittal sections. All slices were collected sequentially in a 24-well plate filled with PBS, before being mounted in their appropriate anatomical order on a glass slide in custom-made mounting medium. For confocal imaging, a Zeiss LSM 880 microscope (Carl Zeiss) was used to scan through the whole series of slices and locate the position of the DiI fluorescent mark. Images were then acquired using an EC Plan-Neofluar 20×/0.8 NA air immersion, 40×/1.3 oil immersion, or 63×/1.4 oil immersion objective (Zeiss, laser power: 2–15%; optical slice: 1.28–1.35 airy units, step size: 2 μm). Before acquisition, gain and digital offset were established to optimize the dynamic range of acquisition to the dynamic range of the GCaMP6m and DiI signals. Settings were kept constant during acquisition across brains. Based on the location of the red fluorescent mark, we could infer where, on the medio–lateral and dorso–ventral extent of the brain, the ventro–lateral corner of the microprism (and hence the 2-photon FOV aligned to it) was located.

We used the Paxinos mouse brain atlas[53] to produce a reference flat map representing the medio–lateral and dorso–ventral extent of the MEC and PaS. Flat maps helped delineate the extent of the FOV that fell within the anatomical boundaries of either the MEC and adjacent PaS, and allowed for a standardized comparison across mice. For each imaged mouse, we mapped the dorso–ventral and medio–lateral location of the DiI mark on the refence flat map (Extended Data Fig. 1c). Mice were assigned to 'MEC imaging' or 'PaS imaging' groups depending on the location of the FOV: a mouse would be further analysed as being part of the MEC imaging group if more than 50% of the area of the FOV occupied by GCaMP6m-expressing cells could be located in the MEC.

To verify the anatomical location of the FOVs in VIS in the glass window implanted mice, we sliced the brain until we reached the anatomical coordinates at which the virus was infused (see 'Surgeries'). Coronally cut slices of 50 μm thickness were collected sequentially in a 24-well plate, and immediately mounted in their appropriate anatomical order on a glass slide in custom-made mounting medium. For confocal imaging, a Zeiss LSM 880 microscope (Carl Zeiss) was used according to the same specification as described above for MEC/PaS.

**Histology and reconstruction of Neuropixels probe placement.** After the end of experiments, the mice were anaesthetized and received an overdose of isoflurane (IsoFlo, Zoetis) before transcardial perfusion with saline followed by 4% formaldehyde. The brain was

either extracted after perfusion or kept overnight in 4% formaldehyde for post-fixation before extraction. The brains were then stored in 4% formaldehyde. Frozen 30 μm thick sagittal sections were cut on a cryostat, mounted on glass, and stained with Cresyl violet (Nissl). To estimate the shank locations, we used an Axio Scan.Z1 (Carl Zeiss) slide scanner microscope for brightfield detection at 20x magnification. We used Paxinos mouse brain atlas[53] and the Allen Mouse Brain Common Coordinate Framework[58] version 3 through the siibra-explorer (Forschungszentrum Juelich, https://atlases.ebrains.eu/viewer/) to estimate anatomical location of recording sites. A map of the probe shank was aligned to the histology assuming that the cutting plane was near-parallel to the sagittal plane. When possible, the anatomical locations were calculated using the tip of the probe shanks and the intersection of the shank with the brain surface as reference frames. When this was not possible, the profile of a nearby brain region (for example, the hippocampus) was used to estimate the MEC implant site. We observed theta-rhythmicity of neural activity on all recorded shanks, as expected for recording locations in the MEC.

### Analysis of imaging time series

Imaging time series data were analysed using the Suite2p[59] Python library (https://github.com/MouseLand/suite2p). We used its built-in routines for motion correction, region of interests (ROI) extraction, neuropil signal estimation, and spike deconvolution. Non-rigid motion correction was chosen to align each frame iteratively to a template. Quality was assessed by visual inspection of the corrected stacks and built-in motion correction metrics. The Suite2p GUI was used to manually sub-select putative neurons based on anatomical and signal characteristics and to discard obvious artefacts that accumulated during the analysis—for example, ROIs with footprints spanning large areas of the FOV, ROIs that did not have clearly delineated circumferences in the generated maximum intensity projection, or ROIs that were extracted automatically but showed no visible calcium transients.

Raw fluorescence calcium traces of each ROI were neuropil-corrected to create a fluorescence calcium signal $F_{corr}$ by subtracting 0.7 times the neuropil signal from the raw fluorescence traces. We used the Suite2p integrated version of non-negative deconvolution[60] with tau = 1 s to deconvolve $F_{corr}$, yielding the basis for the binarized sequences that we refer to as the calcium activity (see 'Binary deconvolved calcium activity and matrix of calcium activity'). Due to the absence of ground truth data for our combination of indicator, region, and imaging conditions, we used a decay tau that was at the lower end of biologically plausible values (tau = 1 s), which allowed even short and low amplitude spiking responses to be picked up by the analysis and therefore did not bias our analysis towards large-amplitude calcium transients (presumed bursting responses). To estimate the signal-to-noise ratio (SNR) of each cell individually, we further thresholded the calcium activity (without binarization) at 1 s.d. over the mean, yielding filtered calcium activity, and classified the remaining activity as noise. We additionally ensured that noise was temporally well segregated from filtered calcium activity by requiring data points classified as noise to be separated by at least one second before and ten seconds after filtered calcium activity. The SNR of the cell was then estimated as the ratio of the mean amplitude of $F_{corr}$ during episodes of filtered calcium activity over the s.d. of $F_{corr}$ during episodes of noise. If no data points remained after the filtering of calcium activity, the cell was assigned a SNR of zero.

### Binary deconvolved calcium activity and matrix of calcium activity

In order to denoise the recorded fluorescence calcium signals and have good temporal resolution, all analyses in the study were performed using the deconvolved calcium activity of the recorded cells. For each cell whose SNR was larger than 4, the deconvolved calcium activity (see 'Analysis of imaging time series') was downsampled by a factor of 4 by calculating the mean over time windows of ~129 ms

(original sampling frequency = 30.95 Hz, sampling frequency used in the analyses = 7.73 Hz). Because the ultraslow oscillations and periodic sequences unfolded at the time scales of seconds to minutes, this downsampling step gave a good temporal resolution for all quantifications while allowing us to work with smaller arrays (ultraslow oscillations and the oscillatory sequences were also detectable when using the original sampling frequency), which in some of the analyses reduced the computing time. Next, the downsampled deconvolved calcium activity was averaged over time and its s.d. was calculated. A threshold equal to this average plus 1.5 times the s.d. was used to convert the deconvolved calcium activity into a binary deconvolved calcium activity, such that all values above the threshold were set to 1 (calcium events), and all values below or equal to that threshold were set to 0. Unless stated otherwise, for all analyses throughout the study we used the deconvolved and binary calcium activity, to which for simplicity we refer to as 'deconvolved calcium activity' or simply 'calcium activity'. The calcium activity of all cells in a session with SNR > 4 was stacked to construct a binary matrix of calcium activity which had as many rows as neurons, and as many columns as time bins sampled at 7.73 Hz. The population vectors are the columns of the matrix of calcium activity.

Note that the recorded calcium signals likely reflect a combination of groups of single spikes and higher-frequency bursts, although it was not possible to distinguish between the two types of firing. The sensitivity of the calcium indicator was likely not high enough to detect subthreshold potentials.

## Spike Sorting and single-unit selection

Spike sorting of Neuropixels data was performed using a version of KiloSort 2.5 (ref. 54) with some customizations to improve performance on recordings from the MEC region as described previously[25]. All trials in a session were clustered together. Single units were discarded from analysis based on a < 20% estimated contamination rate with spikes from other neurons. These units were automatically labelled by the KiloSort 2.5 algorithm as 'good' units. In the example session from mouse no. 104638 only good units were considered. In the example session of mouse no. 102335, because the number of good units was lower (<250), we also used multi-unit activity (MUA).

## Autocorrelations and spectral analysis of single-cell calcium activity

To determine if the calcium activity of single cells displays ultraslow oscillations, for each neuron the PSD was calculated on the autocorrelation of its calcium activity. The PSD was computed using Welch's method (pwelch, built-in Matlab function), with Hamming windows of 17.6 min (8,192 bins of 129 ms in each window) and 50% of overlap between consecutive windows. Note that when calculating the PSD a large window was needed to identify oscillation frequencies ≪0.1 Hz.

To visualize whether specific oscillatory patterns at fixed frequencies were present in the neural population, all autocorrelations from one session were sorted and stacked into a matrix, where rows are cells and columns are time lags. The sorting of autocorrelations was performed according to the maximum power of each PSD in a descending manner. The frequency at which the PSD peaked was used as an estimate of the oscillatory frequency of the cell's calcium activity.

In order to determine significance for the peak of the PSD, we considered two extreme and opposite shuffling procedures: On the one hand, given that circularly shuffling the data preserves all inter calcium events (Extended Data Fig. 3c,d), taking this approach would preserve the shape and the position of the peak in the PSD calculated on experimental data. On the other hand, destroying the inter calcium event intervals by assigning a random position to each calcium event in the time series would lead to a flat PSD (Extended Data Fig. 3c,d). In the latter approach, all cells would be classified as oscillatory. To bridge these two approaches we developed a new shuffling procedure. For each cell we divided its calcium activity vector into $n$ epochs of

length $W$, with $n = T /(W \cdot SF)$, where $T$ is the total number of time bins sampled at a frequency SF = 7.73 Hz (that is, bin size = 129 ms). We next shuffled those epochs (and preserved the ordering of the time bins within each epoch). This method preserved the inter calcium event interval, but at the same time disrupted the periodicity. In the limit where $W$ = 129 ms, this method coincides with shuffling all calcium events without preserving the inter calcium event intervals; in the limit where $W = T/SF$, this method is equivalent to circularly shuffling the data. For each of the 200 shuffled realizations we calculated the PSD and the fraction of cells for which the peak of the PSD in experimental data was above the 95th percentile of a shuffled distribution built with the values of the PSDs calculated on shuffled data (and at the frequency at which the PSD computed on experimental data peaked). Here we present the results for 5 different epoch lengths:

$W$ = 1 s: 6226 oscillatory cells out of 6231 (99%)
$W$ = 10 s: 6153 oscillatory cells out of 6231 (99%)
$W$ = 20 s: 5695 oscillatory cells out of 6231 (91%)
$W$ = 50 s: 4642 oscillatory cells out of 6231 (74%)
$W$ = 100 s: 3521 oscillatory cells out of 6231 (56%)

When $W$ is below the typical duration of the sequences ($W$ < 50 s), the great majority of cells are classified as having a peak in the PSD. As expected, when $W$ is similar to the duration of the sequences ($W \geq 50$ s), the fraction of oscillatory cells quickly drops. This fraction is no longer significantly above a chance level of 5%.

This approach was used for determining the fraction of oscillatory cells both in calcium imaging and in Neuropixels data. In the main text we present the results corresponding to $W$ = 20 s.

Finally, we note that there was some variability in the frequency at which the PSD peaked across cells within a session. For example, in the example session shown in Fig. 1b–d and Fig. 2a, some single-cell PSDs peaked at a frequency of 0.0066 Hz, while others did so at a frequency of 0.0075 Hz. However, in many cases the PSDs were wide enough to exhibit high power in neighbouring frequencies too, providing support to the frequencies being rather clustered among a subset of values, with some slight variability around those values. When all cells were analysed ($n$ = 6,231 cells pooled across 15 oscillatory sessions, 5 mice), in approximately half of the MEC data the oscillatory frequency at the single-cell level was very similar to the frequency at the population level (Extended Data Fig. 7e). This finding points to a small variability in the frequency of single-cell activity in MEC, as expected in the presence of recurring sequences.

## Correlation and PCA sorting methods

To determine whether neural population activity exhibits temporal structure we visualized the population activity by means of raster plots in which we sorted all cells according to different methods.

**Correlation method.** This method sorts cells such that those that are nearby in the sorting are more synchronized than those that are further away. First, each calcium activity was downsampled by a factor 4 by calculating the mean over counts of calcium events in bins of 0.52 s. The obtained calcium activity was then smoothed by convolving it with a gaussian kernel of width equal to four times the oscillation bin size, a bin size that was representative of the temporal scale of the population dynamics (see 'Oscillation bin size'). The cross correlations between all pairs of cells were calculated using time bins as data points, and a maximum time lag of 10 time points, equivalent to ~ 5 s. This small time lag allowed us to identify near instantaneous correlation while keeping information about the temporal order of activity between cell pairs. The maximum value of the cross-correlation between cell $i$ and cell $j$ was stored in the entry $(i, j)$ of the correlation matrix $C$, which was a square matrix of N rows and N columns, where $N$ was the total number of recorded neurons in the session with SNR > 4. If the cross-correlation peaked at a negative time lag the value in the entry $(i, j)$ was multiplied

by −1. The entry with the highest cross-correlation value was identified and its row, denoted by $i_{max}$, was used as the 'seed' cell for the sorting procedure and chosen to be the first cell in the sorting. Cells were then sorted according to the values in the entries $(i_{max}, j)$, $j = 1, 2, …, N, j \neq i_{max}$, that is, their correlations with the seed cell, in a descending manner.

**PCA method.** Computing correlations from the calcium activity or the calcium signals can be noisy due to fine tuning of hyperparameters (for example, the size of the kernel used to smooth the calcium activity of all cells). To avoid this, we leveraged the fact that the periodic sequences of neural activity constitute low-dimensional dynamics with intrinsic dimensionality equal to 1, and sorted the cells based on an unsupervised dimensionality reduction[61] approach (a similar approach was used in ref. 62). For each recording session, PCA was applied to the matrix of calcium activity (bin size = 129 ms; using Matlab's built-in pca function), including all epochs of movement and immobility and using the rows (neurons) as variables and the columns (time bins) as observations. The first two principal components (PCs) were kept, since 2 is the minimum number of components needed to embed non-linear 1-dimensional dynamics. Cells were sorted according to their loadings in PC1 and PC2, expecting that the relationship between these loadings would express the ordering in cell activation during the sequences.

The plane spanned by PC1 and PC2 was named the PC1–PC2 plane. In the PC1–PC2 plane, the loadings of each neuron (the components of the eigenvectors without being multiplied by the eigenvalues) defined a vector, for which we computed its angle $\theta_i = \arctg\left(\frac{l^i_{PC2}}{l^i_{PC1}}\right) \in [-\pi, \pi)$, $1 \leq i \leq N$, with respect to the axis of PC1, where $l^i_{PCj}$ is the loading of cell $i$ on $PCj$. Cells were sorted according to their angle $\theta$ in a descending manner.

Note that while we keep the first 2 principal components to sort the neurons, all principal components and the full matrices of calcium activity were used in the analyses (except for visualization purposes— for example, see 'Manifold visualization for MEC sessions'). Finally, note that because in PCA a principal component is equivalent to −1 times the principal component, the sorting and an inversion of the sorting are equivalent. The sorting was chosen so that sequences would progress from the bottom to the top in the raster plot.

The PCA method was used throughout the paper for sorting the recorded cells unless otherwise stated.

**Random sorting of cell identities.** A random ordinal integer $\in [1, N]$, where $N$ is the total number of recorded cells with SNR > 4, was assigned to each neuron without repetition across cells. Neurons were sorted according to those assigned numbers (see example session in Extended Data Fig. 4d, top row).

**Sorting of circularly shuffled data.** A shuffled matrix of calcium activity was built by circularly shuffling the calcium activity of each cell separately. For each cell a random ordinal integer $\in [1, T]$, where $T$ is the total number of time bins (bin size = 129 ms), was chosen and the calcium activity was rigidly shifted by this integer using periodic boundary conditions. The assignment of random ordinal integers was made separately for each cell. The PCA method was then applied to the shuffled matrix of calcium activity (see example session in Extended Data Fig. 4d, second row).

**Sorting of temporally shuffled data.** Because circularly shuffling the data preserves the oscillations in the single-cell calcium activity, a second shuffling approach was considered (for single-cell data shuffling procedures see 'Autocorrelations and spectral analysis of single-cell calcium activity'). A shuffled matrix of calcium activity was built by temporally shuffling the calcium activity of each cell separately. For each cell, each time bin of the calcium activity was assigned a random ordinal integer $\in [1, T]$ without repetition across time bins, where $T$ is the total number of time bins (bin size = 129 ms), and time bins were

ordered according to their assigned number. The assignment of random ordinal integers was made separately for each cell, so that the obtained random orderings were not shared across cells. The PCA method was then applied to the shuffled matrix of calcium activity.

**Sortings are preserved when different portions of data are used for obtaining the sortings.** To determine whether using different portions of the session for sorting the neurons lead to different sortings, the PCA method was applied to: (i) all data within a session; (ii) the first half of the session; and (iii) the second half of the session. This procedure gave three sortings per session. Next, for each cell pair in a session the distance between the two cells in each of the three sortings was calculated. We illustrate this calculation with a toy example: if 5 neurons were recorded, and sorting (i) was: (1,4,5,2,3), the distance between cells 1 and 5 was 2, because those two cells were 2 positions apart in the sorting. The distance between cells 1 and 3 was 1 and not 4, however, because in the calculation of distances we took into account that the sorting mirrors the position of the cells in the ring, which has periodic boundary conditions.

We next calculated the correlation between the distances in: sorting (i) versus sorting (ii), sorting (i) versus sorting (iii) and sorting (ii) versus sorting (iii). If sortings obtained with different LEM portions of data preserve the ordering of the neurons, we would expect high correlation values. We compared the obtained correlation values with the 95th percentile of a shuffled distribution obtained by assigning, to each cell, a random position in each of the sortings.
- Sorting (i) versus sorting (ii): 15 of 15 oscillatory sessions (see 'Oscillation score') were above the cutoff of significance. Correlation values in experimental data ranged from 0.38 to 0.85. The 95th percentile of shuffled data ranged from 0.004 to 0.015 ($n = 15$ in both experimental and shuffled data).
- Sorting (i) versus sorting (iii): 15 of 15 oscillatory sessions were above the cutoff of significance. Correlation values in experimental data ranged from 0.52 to 0.86. The 95th percentile of shuffled data ranged from 0.005 to 0.013 ($n = 15$ in both experimental and shuffled data).
- Sorting (ii) versus sorting (iii): 15 of 15 oscillatory sessions were above the cutoff of significance. Correlation values in experimental data ranged from 0.17 to 0.53. The 95th percentile of shuffled data ranged from 0.005 to 0.013 ($n = 15$ in both experimental and shuffled data).

The high correlation values obtained provide support for what is illustrated in Extended Data Fig. 4e: using different portions of data for sorting the cells unveils the same dynamics.

### Sorting methods based on non-linear dimensionality reduction techniques

The PCA method for sorting cells relies on a two-dimensional linear embedding. This linear embedding might not be optimal if the population vectors describe temporal trajectories that, despite being low-dimensional, lie on a curved surface. To take into account potential non-linearities, four additional sorting methods were implemented, based on the following non-linear dimensionality reduction techniques[63]: $t$-distributed stochastic neighbour embedding ($t$-SNE), LEM, Isomap and uniform manifold approximation and projection (UMAP)[64] (see parameters below). First, to express in the sortings the ordering of the cells during the slow temporal progression of the sequences, the four methods used a resampled matrix of calcium activity as input. To compute this matrix, for each session, we downsampled each calcium activity by a factor 4 by calculating its mean in bins of 0.52 s. The calcium activity of all cells was then smoothed by convolving them with a gaussian kernel whose width was given by the oscillation bin size (see 'Oscillation bin size'). After applying $t$-SNE, LEM, Isomap or UMAP to the resampled matrix of calcium activity, we kept the first two dimensions obtained with each method, for the same reasons as presented for the PCA sorting method. To obtain the sorting, the following procedure

was applied: We let Dim1 and Dim2 be the first two dimensions obtained with the chosen dimensionality reduction technique that we had applied to the resampled matrix. In analogy with the PCA method, the Dim1–Dim2 plane was spanned by Dim1 and Dim2 and for each cell the components on those dimensions defined a vector in this plane for which the angle $\theta \in [-\pi, \pi)$ with respect to the axis of Dim1 was computed. Cells were then sorted according to their angles in a descending manner.

To apply $t$-SNE to the population activity we used a perplexity value of 50. First, we applied PCA to the resampled matrix of calcium activity, and then we used the projection of the neural activity onto the first 50 principal components as input to $t$-SNE. To apply LEM to the population activity, we used as hyperparameters $k = 15$ and $\sigma = 2$. Similarly, we used $k = 15$ for running isomap. Finally, we used n_neighbors=30, min_dist=0.3 and correlation as metric for running UMAP.

We used the MATLAB implementation of UMAP[65] and the Matlab Toolbox for Dimensionality Reduction (https://lvdmaaten.github.io/drtoolbox/). Finally, when displaying the raster plots that resulted from the different sortings, the first cell (located at the bottom of the raster plot) was always the same. This was accomplished by circularly shifting the cells in the different sortings such that the initial cell in all sortings coincided with the initial cell of the sorting obtained with the PCA method.

## Manifold visualization for MEC sessions

Sorting the cells and visualizing their combined neural activity through raster plots revealed the presence of oscillatory sequences of neural activity in the recorded data. To visualize the topology of the manifold underlying the oscillatory sequences of activity, both PCA and LEM were used.

PCA was applied to the matrix of calcium activity, which first had each row convolved with a gaussian kernel of width equal to four times the oscillation bin size (see 'Oscillation bin size'). The manifold was visualized by plotting the neural activity projected onto the embedding defined by PC1 and PC2. In Fig. 2c (left) the neural activity of the entire session was projected onto the low-dimensional embedding. In Extended Data Fig. 4c, the neural activity corresponding to the concatenated epochs of uninterrupted oscillatory sequences was projected onto the embedding.

For the LEM approach, first PCA was applied to the matrix of calcium activity, which was previously resampled to bins of 0.52 s as in 'Sorting methods based on non-linear dimensionality reduction techniques', and the first five principal components were kept. Next LEM was applied to the matrix composed of the 5 principal components, using as parameters $k = 15$ and $\sigma = 2$. We decided to keep 5 principal components prior to applying LEM to denoise the data, for which we leveraged the fact that sequences of activity constitute low-dimensional dynamics with intrinsic dimensionality equal to 1, and therefore truncating the data to the first 5 principal components should preserve the sequential activity. The manifold was visualized by plotting the neural activity projected onto the embedding defined by the first two LEM dimensions. In Fig. 2c (right) the neural activity of the entire session was projected onto the embedding.

Both approaches revealed a ring-shaped manifold along which the population activity propagated repeatedly with periodic boundary conditions. One sequence was equivalent to one full turn of the population activity along the ring-shaped manifold. Finally, we note that when using PCA for visualizing the manifold, in some sessions the ring was less evident (Extended Data Fig. 4c). This is because the population activity had more variations from sequence to sequence, which resulted on the rings that corresponded to each sequence not completely overlapping in the PC1 versus PC2 plane. While recovering rings with PCA is challenging due to PCA being a linear method, using a non-linear method would have helped in visualizing the ring (as in Fig. 2c, right), but we decided not to do this for all quantifications because non-linear methods require more fine tuning and are usually harder to interpret.

## Phase of the oscillation

To track the progression of the population activity over time, we leveraged the low dimensionality of the ring-shaped manifold and the circular nature of the population activity, and parametrized the population activity with a single time-dependent parameter, which we called the phase of the oscillation. Hence, the phase of the oscillation varied as a function of time (bin size = 129 ms) and tracked the progression of the neural population activity during the oscillatory sequences. The neural activity was projected onto a two-dimensional plane using PCA. The use of PCA avoided the selection of hyperparameters, which is required in all non-linear dimensionality reduction techniques including LEM. Let $PCi_t(t)$ be the projection of the neural population activity onto principal component $i$ (PC$i$). The neural population activity at time point $t$ projected onto the plane defined by PC1 and PC2 is then given by $(PC1_t(t), PC2_t(t))$, which defines a vector in this plane. The phase of the oscillation is defined as the angle of this vector with respect to the PC1 axis and is given by

$$\varphi(t) = \arctan\left(\frac{PC2_t(t)}{PC1_t(t)}\right). \tag{1}$$

During one sequence, the phase of the oscillation continuously traversed the range $[-\pi, \pi)$ rad, which was consistent with the population activity propagating through the network and describing one turn along the ring-shaped manifold. The repetitive and almost linear dependence between the phase of the oscillation and time illustrates how stereotyped the sequences were (Fig. 2d).

We note that the quantity $\varphi(t)$ is always defined, regardless of whether the session is or is not classified as oscillatory. In the case of the oscillatory sessions, $\varphi(t)$ tracks the progression of the oscillatory sequences.

## Joint distribution of cross-correlation time lag and angular distance in the PCA sorting

To further characterize the sequential activation in the MEC neural population and to introduce a score that would determine the extent to which a session exhibited oscillatory sequences (see 'Oscillation score'), we determined the relationship between the time lags that maximized the cross-correlation between the calcium activity of two cells ($\tau$) and their angular distances in the PCA sorting ($d$). In the plane generated by PC1 and PC2, the loadings of each neuron defined a vector, for which we computed the angle $\theta_i = \arctan\left(\frac{l^i_{PC2}}{l^i_{PC1}}\right) \in [-\pi, \pi)$, $1 \leq i \leq N$, with respect to the axis of PC1, where $l^i_{PCj}$ is the loading of cell $i$ on PC$j$ and $N$ is the total number of recorded neurons (see 'Correlation and PCA sorting methods'). The angular distance $d$ between any two cells in the PCA sorting was calculated as the difference between their angles wrapped in the interval $[-\pi, \pi)$ (see Extended Data Fig. 5b, left),

$$d_{i,j} = (\theta_i - \theta_j), \tag{2}$$

where $1 \leq i \leq N$, $1 \leq j \leq N$. The Matlab function angdiff was used for computing this distance. Note that the angular distance maps how far apart two cells are in the raster plot when cells are sorted according to the PCA method.

To estimate the joint distribution of cross-correlation time lags and angular distances in the PCA sorting, the cross correlations between all pairs of cells were calculated using a maximum time lag of 248 s. For each cell pair the time lag at which the cross-correlation peaked ($\tau$) and the angular distance in the PCA sorting ($d$) were calculated. A discrete representation was used for these two variables: in all analyses, and unless stated otherwise, the range of possible $\tau$ values—that is, $[-248, 248]$ s—was discretized into 96 bins of size $\Delta\tau = \frac{496\,s}{96} \sim 5$ s and the range of possible $d$ values—that is, $[-\pi, \pi)$ rad—was discretized into 11 bins of size $\Delta d = \frac{2\pi}{11} \sim 0.57$ rad. Using those bins, the joint distribution

of $\tau$ and $d$ was expressed as a two-dimensional histogram that counted the number of cell pairs observed for every combination of $\tau$ bins and $d$ bins, normalized by the total number of cell pairs.

An example of joint distribution of cross-correlation time lags and angular distances in the PCA sorting is presented in Extended Data Fig. 5b, right, built on the example session shown in Fig. 2a. In sessions with clear periodic sequences, the time lag $\tau$ increased with the distance $d$. This dependence was observed a discrete number of times in each session, which indicated that cells were active periodically and at a fixed frequency or at an integer multiple of it (see Extended Data Fig. 5c, top for another example with a different time scale). In sessions without detectable periodic sequences such structure was not observed (Extended Data Fig. 5c, bottom).

## Oscillation score

While striking oscillatory sequences were observed in multiple sessions and mice, the population activity exhibited considerable variability, ranging from non-patterned activity to highly stereotypic and periodic sequences (Extended Data Fig. 5a). This variability prompted us to quantify, for each session, the extent to which the population activity was oscillatory, which we did by computing an oscillation score. For each session, we first calculated the phase of the oscillation $\varphi(t)$ (bin size = 129 ms, equation (1)), which tracks the progression of the population activity in the presence of oscillatory sequences (see 'Phase of the oscillation' and Fig. 2d). Next the PSD of $\sin(\varphi(t))$ was calculated using Welch's method with Hamming windows of 17.6 min (8,192 bins of 129 ms in each window) and 50% of overlap between consecutive windows (pwelch Matlab function, see 'Autocorrelations and spectral analysis of single-cell calcium activity'). If the PSD peaked at 0 Hz and the PSD was strictly decreasing, the phase of the oscillation was not oscillatory and hence the population activity was not periodic in the analysed session. In this case the oscillation score was set to zero. Otherwise, prominent peaks in the PSD at a frequency larger than 0 Hz were identified. In order to disentangle large-amplitude peaks from small fluctuations in the PSD, a peak at frequency $f_{max}$ was considered prominent and indicative of periodic activity if its amplitude was larger than (1) 9 times the mean of the tail of the PSD (that is, $<PSD(f > f_{max})>$, where $<x>$ indicates the average over frequencies $x$) and (2) 9 times the minimum of the PSD between 0 Hz and $f_{max}$ (that is, $\min(PSD(f < f_{max}))$). If no peak in the PSD met these criteria the oscillation score was set to zero. Otherwise, the presence of a prominent peak in the PSD calculated on $\sin(\varphi(t))$ was considered indicative of periodic activity at the population level. Yet a crucial component for observing oscillatory sequences is that cells fire periodically and that the time lag that maximizes the cross correlations between the calcium activity of pairs of cells that are located at a fixed distance in the sequence comes in integer multiples of a minimum time lag, which ensures that cells oscillate at a fixed frequency and that the calcium activity of one cell is temporally shifted with respect to the other. To quantify the extent to which these features were present in the data, we computed the joint distribution of time lags and angular distance in the PCA sorting ($\tau$ was discretized into 240 bins and $d$ was discretized into 11 bins, see 'Joint distribution of cross-correlation time lag and angular distance in the PCA sorting'). Next for each bin $i$ of $d$, $1 \le i \le 11$, we calculated the PSD of the distribution of $\tau$ conditioned on the distance bin $i$ (Welch's methods, Hamming windows of 128 $\tau$ bins with 50% overlap between consecutive windows, pwelch Matlab function). The presence of a peak in this signal indicated that for bin $i$ of $d$, the time lag that maximizes the cross correlations between cells was oscillatory (that is, it peaked at multiples of one specific time lag), as expected when cells are active periodically with an approximately fixed frequency and also with harmonics of the primary frequency (see example joint distribution in Extended Data Fig. 5b, right). The presence (or absence) of a peak that satisfied the condition of being larger than (1) 10 times the mean of the tail of the PSD (same definition as above), and (2) 4.5 times larger than the minimum between 0 Hz and the frequency

at which the PSD peaked, was identified (same definition as above, the parameters are different from the ones used above because the signals are very different). The oscillation score was then calculated as the fraction of angular distance bins for which a peak was identified.

Based on the bimodal distribution of oscillation scores obtained in the calcium imaging data from MEC (Extended Data Fig. 5d), a session was considered to express oscillatory sequences if the oscillation score was ≥0.72. This cutoff (0.72) corresponded to the smallest oscillation score within the group with high scores (shown in green in Extended Data Fig. 5d). Note that because the distribution of oscillation scores was bimodal any other choice of threshold between 0.27 and 0.72 would have led to the same results. Using as cutoff 0.72 was also equivalent to asking that at least 8 out of the 11 distributions of $\tau$ conditioned on bin $i$ of $d$, $1 \le i \le 11$, had a significant peak in their PSD, which accounted for the fact that for distances in the PCA sorting that are close to zero, cells exhibit instantaneous co-activity rather than co-activity shifted by some specific time lag, which makes the conditional probability not oscillatory. After applying the cutoff, 15 of 27 calcium imaging sessions in MEC in 5 mice were classified as oscillatory (Extended Data Fig. 5d, shown in green), and among those 15 sessions, 10 were recorded with synchronized behavioural tracking (see 'Self-paced running behaviour under sensory-minimized conditions'). The number of recorded cells in the calcium imaging oscillatory sessions ranged from 207 to 520. In the rest of the calcium imaging data, 0 of 25 PaS sessions in 4 mice were classified as oscillatory, and 0 of 19 VIS sessions in 3 mice were classified as oscillatory.

## Oscillation bin size

The oscillatory sequences progressed at frequencies <0.1 Hz that varied from session to session. The oscillation bin size was a temporal bin size representative of the time scale of the oscillatory sequences in each session. It was used to quantify single-cell and neural population dynamics, for which describing the neural activity at the right time scale was fundamental (for example, see 'Transition probabilities'). For each oscillatory session the period of the oscillatory sequences, denoted by $P_{osc}$, was calculated as the inverse of the frequency $f_{max}$ at which the PSD of the signal $\sin(\varphi(t))$ peaked (see equation (1) and 'Oscillation score'), that is, $P_{osc} = f_{max}^{-1}$. Note that this estimate of the period was reliable when during most of the session the network engaged in the oscillatory sequences, in which case the estimate was equivalent to the length of the session divided by the total number of sequences. However, it became less reliable the more interrupted the oscillatory sequences were.

The oscillation bin size $T_{osc}$ was computed as the period of the oscillatory sequences divided by 10,

$$T_{osc} = \frac{P_{osc}}{10} = \frac{1}{10 \times f_{max}}. \tag{3}$$

This choice of bin size was made so that each sequence would progress across ~10 time points. Across 15 oscillatory sessions, the oscillation bin size ranged from 3 to 17 s (see Extended Data Fig. 9d).

In sessions without oscillatory sequences, there was not a well-defined peak in the PSD of $\sin(\varphi(t))$, and therefore the oscillation bin size was not possible or meaningful to calculate. Yet, to perform the quantifications of network dynamics at temporal scales similar to the ones investigated in oscillatory sessions, the mean oscillation bin size computed across all oscillatory sessions was used (mean oscillation bin size = 8.5 s).

Unless otherwise indicated, the utilized bin size was 129 ms.

## Identification of individual sequences

The characterization of the oscillatory sequences required multiple analyses that relied on identifying individual sequences, for example to quantify the duration of the sequences and their variability. The

procedure for identifying individual sequences was based on finding the time points at which each sequence began (visualized typically at the bottom of the raster plot) and ended (visualized typically at the top of the raster plot, see Extended Data Fig. 6a). Note that the beginning and the end of the sequence are arbitrary because of the periodic boundary conditions in the sequence progression, and therefore a different pair of phases that are $2\pi$ apart could have been used for defining the beginning and the end of the sequence.

One sequence was equivalent to one full turn of the population activity around the ring-shaped manifold—that is, during one sequence the phase of the oscillation traversed $2\pi$ (see 'Phase of the oscillation'). To calculate the phase of the oscillation and determine the time epochs during which it traversed $2\pi$, we smoothed the calcium activity of all cells (bin size = 129 ms) using a gaussian kernel of width equal to the oscillation bin size. Next, the phase of the oscillation was calculated and discretized into 10 bins (that is, the range $[-\pi, \pi)$ was discretized into 10 bins). Time points at which the phase of the oscillation belonged to a bin that was 3 or more bins away from the bin in the previous time point were considered as discontinuity points and were used to define the beginning and the end of putative sequences. Putative sequences were classified as sequences if the phase of the oscillation smoothly traversed the range $[-\pi, \pi)$ rad in an ascending manner. To account for variability, decrements of up to 1 bin of the phase of the oscillation were allowed. This means that there could be fluctuations of up to 0.6 rad in the phase within one individual sequence, and still be considered a sequence. Points of sustained activity were disregarded. Segments of sequences in which the phase of the oscillation covered at least 5 bins (that is, 50% or more of the range $[-\pi, \pi)$ rad) were also identified.

## Sequence duration, sequence frequency and ISI

The duration of individual sequences was defined as the amount of time that it takes the phase of the oscillation to cover the range $[-\pi, \pi)$ in a smooth and increasing manner, which is consistent with the population activity completing one full turn along the ring-shaped manifold. To calculate the sequence duration, the time interval between the beginning and the end of the sequence was determined (see 'Identification of individual sequences').

To quantify the variability in sequence duration within and between sessions, two approaches were adopted. In approach 1 (Extended Data Fig. 6f left), the s.d. of sequence durations was computed for each oscillatory session. To estimate significance, in each of 500 iterations all sequences across 15 oscillatory sessions were pooled (421 sequences in total) and randomly assigned to each session while keeping the original number of sequences per session unchanged. For each iteration the s.d. of the sequence durations randomly assigned to each session was calculated. In approach 2 (Extended Data Fig. 6f, right), for each session $i$, $1 \leq i \leq 15$, where 15 is the total number of oscillatory sessions, we considered all pairs of sequences within session $i$ (within session group) or alternatively all pairs of sequences such that one sequence belongs to session $i$ and the other sequence to session $j$, $j \neq i$ (between session group). For each sequence pair in each group, the ratio between the shortest sequence duration and the longest sequence duration was calculated. The mean was computed over pairs of sequences in each group for each session separately. Notice that the larger this ratio the more similar the sequence durations are.

The sequence frequency was calculated as the total number of identified individual sequences in a session, divided by the total amount of time the network engaged in the oscillatory sequences during the session, which was computed as the length of the temporal window of concatenated sequences.

The ISI was defined as the length of the epoch from the termination of one sequence and the beginning of the next one. In other words, the ISI was calculated as the amount of time that elapsed between the time point at which the phase of the oscillation reached $\pi$ (after completing one turn along the ring-shaped manifold), and the time point at which it is equal to $-\pi$ (prior to initiating the next turn along the ring).

## Mean event rate during segments of the sequences

To determine how population activity varied during individual sequences (Extended Data Fig. 6c), the following approach was adopted. For each oscillatory session (see 'Oscillation score') all individual sequences were identified (see 'Identification of individual sequences'). Each sequence was divided into ten segments of equal length. For each sequence segment, the mean event rate was calculated as the total number of calcium events across cells divided by sequence segment duration and number of cells. For each session the mean event rate per segment was calculated over sequences. Across sessions we found that the percentage rate change from the segment with the minimum event rate to the segment with the maximum rate was no more than 18% (Extended Data Fig. 6c).

## Analysis of Neuropixels data

Neuropixels data was different from the calcium imaging data in that it consisted of spike times and not calcium traces. Despite this fundamental difference, for most of the analyses we applied the same methods to both datasets. When this was not possible (see below), we tried to minimize the differences between the two analyses pipelines.

**Spike matrices.** In order to create arrays that were similar to the matrices of calcium activity, for each recorded unit a spike train was built using a bin size of 120 ms (similar to the bin size used in calcium imaging data, 129 ms). Each time bin contained the number of spikes produced by the recorded unit in that bin. Spike matrices were built by stacking the spike trains of all recorded units (469 units in the example session presented in Fig. 2f, 410 units in the example session shown in Extended Data Fig. 4g).

Calcium traces are temporally correlated due to the slow dynamics of the calcium indicator. In addition, the observed periodic sequences unfolded over a time scale of minutes. To take these two factors into account, we smoothed the spike train of each recorded unit with a Gaussian kernel of width equal to 5 s.

Both the original spike matrix and the smoothed spike matrix were then binarized using, for each spike train, a threshold equal to the mean plus either 1 or 1.5 times the s.d. (1 for smoothed matrices; 1.5 for non-smoothed matrices; as a reference, the threshold for binarization used in calcium data was the mean plus 1.5 times the s.d.; see 'Binary deconvolved calcium activity and matrix of calcium activity').

In the calcium imaging experiment, it took approximately 5 min to initiate the recording after the mouse was positioned on the wheel (mainly due to the time that was needed to find the imaging planes). In the Neuropixels data there was no such delay between positioning the mice on the wheel and starting the data acquisition. In order to make both datasets as comparable as possible, and in order to remove any effects due to arousal, the first 5 min of the Neuropixels sessions were discarded.

**Autocorrelation and spectral analysis.** The autocorrelations were calculated on the spike trains (without smoothing), and the PSD was calculated on the autocorrelations. Methods and parameters used for calculating the autocorrelation and PSDs were the same as in calcium imaging data ('Autocorrelations and spectral analysis of single-cell calcium activity').

**Calculation of oscillation score.** As in the calcium imaging data, in order to quantify the amount of oscillatory activity in the Neuropixels sessions, an oscillation score was computed. Because in the Neuropixels recordings (unlike in the calcium imaging data) there were some long periods of non-sequence activity between bouts of periodic sequences, possibly due to small differences in training protocol, we computed

the oscillation score not on the full spike matrix but on the matrix of concatenated sequences (built by identifying all individual sequences in the smoothed spike matrix and concatenating them as described for the calcium imaging data in 'Identification of individual sequences' and 'Sequence duration, sequence frequency and ISI' above).

**Sorting calculation and raster plot visualization.** Neural population activity was visualized by means of raster plots, for which units were sorted using the PCA method ('Correlation and PCA sorting methods'). The sorting was calculated on the smoothed spike matrix (Fig. 2f and Extended Data Fig. 4g, top), and the obtained sorting was applied also to the non-smoothed spike matrices (Extended Data Fig. 4f,g, bottom).

While the sorting and visualization of neural population activity were performed as we did in calcium imaging data, there was one difference in how the two datasets were analysed. Because in the Neuropixels data the periodic sequences were more salient in some subsets of the sessions than others, for visualization purposes we calculated the sorting on a subset of the smoothed transition matrices. Those subsets are given by [1,200, 1,700] s for the example session of mouse no. 104368 (Fig. 2f) and [1,100, 1,400] s for the example session of mouse no. 102335 (Extended Data Fig. 4g). Note, however, that sequences were identified outside these session subsets too, indicating that the sorting unveils stereotyped sequences also outside the used subsets of data (see 'Sortings are preserved when different portions of data are used for obtaining the sortings').

## Locking to the phase of the oscillation

To calculate the extent to which individual cells in the calcium imaging experiments were tuned to the oscillatory sequences, two quantities were used: the locking degree and the mutual information between the calcium event counts and the phase of the oscillation. For each oscillatory session, the phase of the oscillation $\varphi(t)$ was computed (see equation (1)) and individual sequences were identified (see 'Identification of individual sequences'). Next, the time points that corresponded to all individual sequences in one session were concatenated, which generated a new signal with the phase of the oscillation for all consecutive sequences, and a new matrix of calcium activity in which the network engaged in the oscillatory sequences uninterruptedly.

The locking degree was computed for each cell as the mean resultant vector length over the phases of the oscillatory sequences at which the calcium events occurred (bin size = 129 ms, function circ_r from the Circular Statistics Toolbox for Matlab[66]). The locking degree has a lower bound of 0 and upper bound of 1. It is equal to 1 if all oscillation phases at which the calcium events occurred are the same (that is, perfect locking), and equal to zero if all phases at which the calcium events occurred are evenly distributed (total absence of locking). To estimate significance, for each cell a null distribution of locking degrees was built by temporally shuffling the calcium activity of that cell 1,000 times while the phase of the oscillation remained unchanged, and by computing, for each shuffle realization, the locking degree (shuffling was performed as in 'Sorting of temporally shuffled data'). The 99th percentile of the estimated null distribution was used as a threshold for significance.

In order to assess the robustness of the locking degree, the obtained results were compared with a second measure based on information theory[67]: the mutual information between the counts of calcium events (event counts) and the phase of the oscillation (bin size = 0.52 s). To estimate the reduction in uncertainty about the phase of the oscillation ($P$) given the event counts of the calcium activity ($S$), Shannon's mutual information was computed as follows[68]:

$$\mathrm{MI}(S, P) = \sum_{p,s} \mathrm{Prob}(p, s) \log_2 \frac{\mathrm{Prob}(p, s)}{\mathrm{Prob}(p)\mathrm{Prob}(s)},$$

where $\mathrm{Prob}(p, s)$ is the joint probability of observing a phase of the oscillation $p$ and an event count $s$, $\mathrm{Prob}(s)$ is the marginal probability of event counts and $\mathrm{Prob}(p)$ is the marginal probability of the phase of the oscillation. All probability distributions were estimated from the data using discrete representations of the phase of the oscillation and the event counts. The event counts were partitioned into $s_{\max} + 1$ bins to account for the absence of event counts as well as all possible event counts, where $s_{\max}$ is the maximum number of event counts per cell in a 0.52 s bin, and the phase of the oscillation was discretized into 10 bins of size $\frac{2\pi}{10}$.

The mutual information is a non-negative quantity that is equal to zero only when the two variables are independent—that is, when the joint probability is equal to the product of the marginals $\mathrm{Prob}(p, s) = \mathrm{Prob}(p)\mathrm{Prob}(s)$. However, limited sampling can lead to an overestimation in the mutual information in the form of a bias[69]. In order to correct for this bias, the calcium activity was temporally shuffled (as in 'Sorting of temporally shuffled data') and the mutual information between the event counts of the shuffled calcium activity and the phase of the oscillation, which remained unchanged, was calculated. This procedure, which destroyed the pairing between event counts and phase of the oscillation, was repeated 1,000 times and the average mutual information across the 1,000 iterations was computed and used as an estimation of the bias in the mutual information calculation. In the right panel of Fig. 3a, we report both the mutual information and the bias. In Extended Data Fig. 7a, the corrected mutual information was reported ($\mathrm{MI_c}$), where the bias ($\langle \mathrm{MI_{sh}} \rangle_{\mathrm{iterations}}$) was subtracted out from the Shannon's mutual information (MI): $\mathrm{MI_c} = \mathrm{MI} - \langle \mathrm{MI_{sh}} \rangle_{\mathrm{iterations}}$.

Note that the locking degree and the mutual information between the event counts and the phase of the oscillation yielded consistent results (see Fig. 3a and Extended Data Fig. 7a).

## Tuning of single cells to the phase of the oscillation

The selectivity of each cell to the phase of the oscillation in the calcium imaging data was visualized through tuning curves and quantified through their preferred phase. As in the analysis of 'Locking to the phase of the oscillation', the phase of the oscillation $\varphi(t)$ was computed, individual sequences were identified, and the time points of the phase of the oscillation and the matrix of calcium activity that corresponded to all individual sequences in one session were concatenated.

**Tuning curves.** The range of phases $[-\pi, \pi)$ rad was partitioned into 40 bins of size $\frac{2\pi}{40}$ rad. For each cell the tuning curve in the phase bin $j$, $j = 0, \ldots, 39$, was calculated as the total number of event counts that occurred at phases within the range $\left[-\pi + j\frac{2\pi}{40}, -\pi + (j+1)\frac{2\pi}{40}\right)$ divided by the total number of event counts during the concatenated oscillatory sequences.

**Preferred phases.** The preferred phase of each cell was calculated as the circular mean over the oscillation phases at which the calcium events occurred (function circ_mean from the Circular Statistics Toolbox for Matlab[66]). In most of the analysis the preferred phase was calculated, for each cell, after concatenating all sequences. However, in a subset of analyses (see 'Anatomical distribution of preferred phases'), the preferred phase was also calculated for individual sequences, as the circular mean over the oscillation phases at which the calcium events occurred in each sequence.

Unless otherwise stated, the preferred phase refers to the calculation performed on concatenated sequences (and not on individual sequences).

**Distribution of preferred phases.** To determine the extent to which the preferred phases across locked cells were uniformly distributed in one recorded session, the distribution of the cells' preferred phases, that we shall denote $Q$, was estimated by discretizing the preferred

phases into 10 bins of size $\frac{2\pi}{10}$ rad. The entropy of this distribution $H_Q = -\sum_{x=1}^{10} Q(x)\log_2(Q(x))$ was calculated and used to compute the entropy ratio $H_{ratio}$ which quantifies how much $Q$ departs from a flat distribution:

$$H_{ratio} = \frac{H_Q}{H_{flat}} \qquad (5)$$

where $H_{flat}$ is the entropy of a flat distribution using 10 bins—that is, $H_{flat} = 3.32$ bits. The closer $H_{ratio}$ is to 1 the flatter $Q$ is, and therefore all preferred phases tend to be equally represented. The smaller $H_{ratio}$ is, the more uneven $Q$ is and some preferred phases tend to be more represented than others.

To estimate significance, for each session the procedure for calculating $H_{ratio}$ was repeated for 1,000 iterations of a shuffling procedure where the preferred phase of the cells was calculated after the values of the phase of the oscillation were temporally shuffled. In Extended Data Fig. 7c, both panels, for each session the 1,000 shuffle realizations were averaged.

## Participation index

The Participation Index (PI) quantifies the extent to which a cell's calcium events were distributed across all sequences, or rather concentrated in a few sequences. For neurons that were active only in a few sequences the participation index was small (participation index ~ 0), and for neurons that were reliably active during most of the sequences the participation index was high (participation index ~ 1; Extended Data Fig. 7g shows three example neurons of the session in Fig. 2a).

The participation index was calculated for each cell separately as the fraction of sequences needed to account for 90% of the total number of calcium events. To compute the participation, individual sequences were identified (see 'Identification of individual sequences'), and for each cell the number of calcium events per sequence was calculated and normalized by the total number of calcium events across all concatenated sequences, which yields the fraction of calcium events per sequence. This quantity was sorted in an ascending manner and its cumulative sum was calculated. The participation index is the minimum fraction of the total number of sequences for which the cumulative sum of the fraction of calcium events per sequence ≥0.9 (results remain unchanged when the cumulative sum is required to be ≥0.95).

## Relationship between tuning to the phase of the oscillation and single-cell oscillatory frequency

To determine whether the frequency of oscillation of single-cell calcium activity was correlated with the extent to which the cell was locked and participated in the oscillatory sequences, for each cell the ratio between its oscillatory frequency (see 'Autocorrelations and spectral analysis of single-cell calcium activity') and the sequence frequency (see 'Sequence duration, sequence frequency and ISI') was calculated and denoted relative frequency. Next, for each session cells were divided into two groups: one group had cells with relative frequency ~1 (cells whose oscillatory frequencies were most similar to the sequence frequency), and the other group had cells with relative frequency ≠1 (cells whose oscillatory frequencies were most different from the sequence frequency). The size of each group was the same and was given by a percentage $\alpha$ of the total number of recorded cells in a session. For each group the locking degree (see 'Locking to the phase of the oscillation') and the participation index (see 'Participation index') were compared. For the quantification across all 15 oscillatory sessions, the mean locking degree and participation index were calculated for each group separately and for each session separately, and all 15 sessions were pooled. $\alpha$ varied from 5% to 50%.

## Anatomical distribution of preferred phases

To determine whether the entorhinal oscillatory sequences resembled travelling waves, during which neural population activity moves progressively across anatomical space[20,21,70–74], we took three complimentary approaches.

**Correlation between differences in preferred phase and anatomical distance. Preferred phases calculated using data from the entire session (after concatenating individual sequences).** For each of the 15 oscillatory sessions (across 5 mice) the Pearson correlation between the anatomical distance between cells in the FOV and the difference in their preferred phases (see 'Tuning of single cells to the phase of the oscillation') was calculated. In order not to count the same data twice, each correlation value was calculated using $N \times (N-1)/2$ samples (each sample was a cell pair), where $N$ was the total number of cells recorded in the session. In the presence of travelling waves, a significant correlation between differences in preferred phase and anatomical distance between cells within the FOV is to be expected. To determine statistical significance the cells' preferred phase were shuffled within the FOV 100 times, and for each shuffled realization the correlation values were calculated. Because we were interested in significant correlations, regardless of whether they were positive or negative, both in experimental and shuffled data we took the absolute value of the correlations. Next, the 95th percentile of the shuffled distribution (100 shuffled realizations per session) was used as cutoff for significance and compared with the correlation value in experimental data.

In order to rule out that the small correlation values observed in experimental data could be masking a dependency such that for larger distances the differences in preferred phase increased in absolute value, the same calculations were repeated but now taking the absolute value of the difference in preferred phase. Statistical significance was determined as in the previous paragraph.

**Preferred phases calculated using data from individual sequences.** Travelling waves could still be present if they move in different directions from sequence to sequence. To test for the presence of travelling waves without assuming similar wave directions across successive sequences, the quantification of correlation between the difference in preferred phase as a function of pairwise anatomical distance was repeated for each sequence separately. To calculate the preferred phase of each cell in each sequence (see 'Tuning of single cells to the phase of the oscillation'), the mean phase at which the calcium events occurred in that individual sequence was computed. In each sequence, only cells that had at least 5 calcium events were included in the analysis. This analysis was performed separately on 421 sequences across 15 oscillatory sessions. Similarly to the analysis described above, when sequences were concatenated within a session, the calculations were repeated after taking the absolute value of differences in preferred phase.

Results are presented in Fig. 3f,g. In Fig. 3f, the correlation value was also non-significant when calculated using the absolute value of the differences in preferred phase (correlation = 0.0028, cutoff for significance of the correlation = 0.0146). In Fig. 3g, in the experimental data the absolute value of the correlations ranged from $6.4 \times 10^{-6}$ to 0.147 ($n = 421$). Out of 421 sequences, 27 were classified as significant when compared to the 95th percentile of a shuffled distribution (cutoffs ranged from 0.007 to 0.237, $n = 421$). The fraction 27/421 was slightly above a chance level of 0.05 ($0.05 \times 421 = 21$ sequences), yet for those 27 sequences the correlation values were very low, ranging from 0.008 to 0.137.

**Calculation of local gradients of preferred phase.** Previous studies have investigated the presence of travelling waves by computing local anatomical gradients of the phase of the oscillation, when the phase is calculated through the Hilbert transform applied to the activity of each electrode (for example, ref. 75, Ecog data). In order to perform a similar analysis but applied to each sequence separately, two different approaches were taken.

**Similarity of preferred phases in spatial bins of the FOV.** First, the similarity in preferred phases of all cells within spatial bins of the FOV was used as a proxy for local gradients. The similarity in preferred phases was calculated as the mean vector length (MVL) of the distribution of preferred phases within each bin of the FOV. The analysis was performed for individual sequences separately.

For each of the 15 oscillatory sessions (over 5 mice), the FOV was divided into spatial bins of 100 μm x 100 μm (6 × 6 bins in 10 sessions, 10 × 10 bins in 5 sessions), or 200 μm x 200 μm (3 × 3 bins in 10 sessions, 5 × 5 bins in 5 sessions) (note that for 10 of the 15 oscillatory sessions the FOV was 600 μm x 600 μm, mice no. 60355, no. 60584, no. 60585; while for 5 of the 15 oscillatory sessions the FOV was 1,000 μm × 1,000 μm, mouse no. 59914; mouse no. 59911 did not show the oscillatory sequences). Next, the preferred phase of each cell per sequence was calculated (as we did in 'Correlation between differences in preferred phase and anatomical distance') and for each sequence and every spatial bin of the FOV the MVL was computed (only spatial bins with 10 or more cells were considered). If the MVL was 0, then all preferred phases in that bin were different and homogeneously distributed between −π and π, whereas if the MVL was 1 then all preferred phases were the same. In the presence of a travelling wave, each bin should have a high MVL value compared to chance levels. Statistical significance was determined by repeating the same MVL calculation after shuffling the cells' preferred phases within the FOV 200 times, and using, for each spatial bin, a cutoff for significant of 95th percentile of the shuffled distribution. A non-significant fraction of spatial bins had a MVL value above the cutoff for significance.

**Differences in preferred phase among pairs of cells in small neighbourhoods of the spatial domain.** The analysis presented above is focused on the degree of similarity between preferred phases in spatial bins. In order to avoid small cell sample effects, and effects of adding a threshold number of cells for bins to be included when calculating similarity with the MVL measure above, we decided to also calculate the difference in preferred phases for all pairs of cells that were located within small neighbourhoods in the FOV, expecting that in the presence of travelling waves the differences in preferred phases of cell pairs within small neighbourhoods would be smaller than expected by chance. For each cell in the FOV, all other cells that were located within a circular neighbourhood of radius 50, 100 or 200 μm were identified and the differences in preferred phase between cell pairs within those areas were calculated. Next, for each sequence and each radius separately all phase differences were pooled, and the mean and the median of the obtained distributions were calculated. To determine significance, the preferred phases across all cells were shuffled 200 times and for each shuffled realization a distribution of differences in preferred phase was obtained and used to calculate the mean and median. Because in the presence of travelling waves smaller differences in preferred phases than in the shuffled data were expected, the mean and median calculated on experimental data were compared with the 5th percentile of the distribution of means and medians obtained from shuffled data. This comparison was performed for each sequence and each radius separately.

**Centre-of-mass calculation of the population activity.** To determine whether the population calcium activity was anatomically localized, as expected in the presence of travelling waves, we calculated its centre of mass (COM). First, all individual sequences were identified and the neural data was averaged in time bins of 5 s. We chose bins of 5 s because the sequences are very slow, however, results remain unchanged if bins of 1 s or 2 s are used instead. For each time point (bin size = 5 s) and for each sequence separately the COM of the population activity was calculated as:

$$\text{COM} = \frac{1}{M} \sum_{i=1}^{N} m_i \mathbf{r}_i,$$

where $N$ is the total number of recorded cells in the session, $\mathbf{r}_i$ is the position of neuron $i$ in the FOV, $m_i$ is the total number of calcium events of neuron $i$ within the 5 s time bin, and $M = \sum_{i=1}^{N} m_i$. The COM was visualized for one example sequence both in experimental data, and after randomly shuffling the position of the cells within the FOV (Extended Data Fig. 8d). To quantify the temporal trajectory of the COM across individual sequences, we calculated the cumulative distance travelled by the COM as the sum of the distances travelled by the COM between consecutive time points (bin size = 5 s). The cumulative distance travelled calculated on experimental data was compared with the 5th and 95th percentile of a distribution built by shuffling the positions of the cells in the FOV 500 times.

### Procedure for merging steps

In order to average out the variability observed in single cells at the level of locking degree and participation index while preserving the temporal properties of the oscillatory sequences, an iterative process that defines new variables from combining the calcium activity of cells was implemented for each session separately (Extended Data Fig. 9a). This process is similar to a coarse-graining approach[76].

First, the $N$ recorded cells in one session were sorted according to the PCA method. In the first iteration of the procedure, named merging step one, the calcium activity (see 'Binary deconvolved calcium activity and matrix of calcium activity') of pairs of cells that were positioned next to each other in the PCA sorting were added up (merging step 1 in Extended Data Fig. 9a). This resulted in $\frac{N}{2}$ new variables, which in merging step 2 were grouped together in pairs of adjacent variables by adding up their activity, which yielded $\frac{N}{4}$ new variables. Note that because in the PCA sorting cells whose activity is synchronous are positioned adjacent to each other, the new variables consist of groups of co-active cells.

In general, merging step $j$ generates $\frac{N}{2^j}$ variables by adding up the activity of pairs of $\frac{N}{2^{j-1}}$ variables from merging step $j-1$, $j > 1$, with each new variable defined as:

$$\widetilde{\sigma}_i = \frac{\sigma_{2i-1} + \sigma_{2i}}{2} \qquad i = 1, \dots, \frac{N}{2^j}$$

where $\widetilde{\sigma}_i$ is the $i$th new variable that results from adding $\sigma_{2i-1}$ and $\sigma_{2i}$, which were computed in the previous merging step, $j-1$. In merging step 1, $\sigma_{2i-1}$ and $\sigma_{2i}$ are the calcium activity of cells in the position $2i-1$ and $2i$, $1 \le i \le N$, in the sorting obtained with the PCA method.

This procedure was repeated 6 times until ~10 variables were obtained in each session (the exact number of variables depended on the number of recorded cells, $N$, in each session). If $N$ was an odd number, the last cell in the sorting obtained with the PCA method was discarded and the procedure was applied to the first $N-1$ cells in the sorting. In every merging step the participation index (see 'Participation index') of each new variable was calculated (see Extended Data Fig. 9b).

### Division of cells into ensembles

After 5 merging steps (and for approximately 10 variables), the participation index reached a plateau (Extended Data Fig. 9b). This motivated the decision to split the recorded cells into 10 variables, which we later used to quantify the population dynamics (see 'Analysis of population dynamics using ensembles of co-active cells'). From now on we will refer to those variables as ensembles, to highlight the fact that cells in each ensemble are co-active. The same number of ensembles was used in sessions that did not exhibit oscillatory sequences.

To distribute cells into 10 ensembles, cells were sorted according to the PCA method. If $\frac{N}{10}$ is an integer, where $N$ is the total number of cells in one session, then each ensemble contains $\frac{N}{10}$ cells and the set of cells that belong to ensemble $i$, $1 \le i \le 10$, is $\left\{ (i-1) \times \frac{N}{10} + 1, (i-1) \times \frac{N}{10} + 2, \dots, i \times \frac{N}{10} \right\}$. If $\frac{N}{10}$ is not an integer then ensembles 1 to 9 contain $\left\lfloor \frac{N}{10} \right\rfloor$ cells and ensemble 10 contains $N - 9 \times \left\lfloor \frac{N}{10} \right\rfloor$

cells, where $\lfloor x \rfloor = \max\{m \in \mathbb{N} / m \le x\}$ and $\mathbb{N}$ is the set of natural numbers. In this case the set of cells that belongs to each ensemble is:

$$\left\{ (i-1) \times \left\lfloor \frac{N}{10} \right\rfloor + 1, (i-1) \times \left\lfloor \frac{N}{10} \right\rfloor + 2, ..., i \times \left\lfloor \frac{N}{10} \right\rfloor \right\}, \ 1 \le \text{ensemble} \le 9$$

$$\left\{ 9 \times \left\lfloor \frac{N}{10} \right\rfloor + 1, 9 \times \left\lfloor \frac{N}{10} \right\rfloor + 2, ..., N \right\}, \ \text{ensemble} = 10$$

Note that each cell was assigned to only one ensemble.

After each cell was assigned to one of the ten ensembles, the activity of each ensemble as a function of time was calculated as the mean calcium activity across cells in that ensemble.

Finally, to calculate the oscillation frequency of ensemble activity, the PSD was calculated (Welch's methods, 8.8 min Hamming window with 50% overlap between consecutive windows, pwelch Matlab function). The oscillation frequency was estimated as the frequency at which the PSD peaked. For each session, the oscillation frequency of the activity of the ensembles was compared to the sequence frequency, which was computed as the total number of sequences in the session divided by the amount of time the network engaged in the oscillatory sequences. The latter was calculated as the length of the temporal window of concatenated sequences (see 'Identification of individual sequences').

## Analysis of population dynamics using ensembles of co-active cells

We adopted an ensemble approach to quantify the population dynamics (see 'Procedure for merging steps' and 'Division of cells into ensembles'). With a total of 10 ensembles this approach averaged out the variability observed in single-cell locking degree and participation index while keeping the temporal progression of the oscillatory sequences (Extended Data Fig. 9f). In sessions with oscillatory sequences, all individual sequences were identified (see 'Identification of individual sequences') and the corresponding time bins were concatenated, which yielded a new matrix of calcium activity in which the oscillatory sequences were uninterrupted. Next, cells were divided into ensembles (see 'Division of cells into ensembles') and ensemble activity was downsampled using as bin size the oscillation bin size of the session (see 'Oscillation bin size'). This procedure yielded a matrix, the ensemble matrix, with the activity of each ensemble corresponding to a single row (10 rows in total), and as many columns as time points sampled at the oscillation bin size. In non-oscillatory sessions, the full matrix of calcium activity was used and the temporal downsampling was conducted at the mean oscillation bin size computed across all 15 oscillatory sessions; that is, bin size = 8.5 s (see 'Oscillation bin size' for a description of the bin size used in non-oscillatory sessions). For both types of sessions (with and without oscillations), the activity of the 10 ensembles was described through a vector expressing, at each time point, the ensemble number with the highest activity at that time point (see Extended Data Fig. 9e,f). This vector was used to perform the following analyses: transition probabilities, probability of sequential activation of ensembles, and sequence score.

**Transition probabilities.** The transition probability from ensemble $i$ to ensemble $j$ was quantified as the number of times the transition $i \rightarrow j$ was observed in the data of one session, normalized by the total number of transitions in one session. Transitions were identified from the vector that contained the ensemble number with maximum activity at each time point (transitions to the same ensemble between consecutive time points were disregarded). Transitions were allocated in a matrix of transition probabilities $T$ of size $10 \times 10$, since 10 ensembles were used. In this matrix, the component $(i, j)$ expressed the transition probability from ensemble $i$ to ensemble $j$.

To establish statistical significance of the transition probabilities, the data was shuffled 500 times. In each shuffle realization, each row of the matrix of calcium activity (with concatenated sequences in the case of oscillatory sessions) was temporally shuffled (as in 'Sorting of temporally shuffled data'), and the procedure for calculating the ensemble matrix and transition probabilities was applied to the shuffled data. For each transition, $i \rightarrow j$ the 95th percentile of the shuffled distribution was used to define a cutoff.

**Probability of sequential activation of ensembles.** We calculated the probability of sequential ensemble activation according to the following procedure. From the vector expressing the ensemble number with the highest activity at each time point (sampled at the oscillation bin size), strictly increasing sequences of all possible lengths (from 2 to 10 ensembles) were identified. The number of ensembles in each sequence was the number of ensembles that were active in consecutive time points (epochs of sustained activity were disregarded). While the sequences had to be strictly increasing, they did not have to be continuous. Sequences could skip ensembles, in which case the maximum number of ensembles in one sequence was less than 10. The probability of the sequential activation of $k$ ensembles, $k = 2,...,10$, was next estimated as the number of times a sequence of $k$ ensembles was found, normalized by the total number of identified sequences. Note that all subsequences were also included in this estimation. For example, if the ensembles 1, 2 and 3 were active in consecutive time points, a sequence of three ensembles was identified, as well as three subsequences of two ensembles each: 1, 2, as well as 2, 3 and 1, 3.

In order to test for significance, the shuffled data from 'Transition probabilities' was used. The procedure to compute the probability of sequential activation of ensembles was applied to each of the 500 shuffle realizations performed per session. Shuffled data was compared with recorded data.

**Sequence score.** The sequence score measures how sequential the ensemble activity is. It is calculated from the probability of sequential activation of ensembles as the probability of observing sequences of three or more ensembles. The sequence score was calculated for each session of the dataset separately. To determine if the obtained scores were significant, for each session the 500 shuffle realizations used in 'Probability of sequential activation of ensembles' for assessing significance of the probability of sequential activation of ensembles were used to calculate the sequence score on shuffled data. Those values were used to build a shuffled distribution, and the 99th percentile of this distribution was chosen as the threshold for significance.

## Estimation of number of completed laps on the wheel, speed and acceleration

Features of the mouse's behaviour were used to determine whether the MEC oscillatory sequences were modulated by running.

The wheel had a radius of 8.54 cm (see 'Self-paced running behaviour under sensory-minimized conditions') and a perimeter of 53.66 cm. Therefore mice had to run for ~53.7 cm to complete one lap on the wheel. For each session, we estimated the number of completed laps on the wheel from the position on the wheel recorded as a function of time. The number of completed laps during one sequence (see 'Identification of individual sequences') was calculated as the total distance run during the sequence divided by 53.7 cm.

The speed of the mouse was numerically calculated as the first derivative of the position on the wheel as a function of time (the sampling frequency of the position was 40 Hz for mice 60355 (MEC), 60353, 60354 and 60356 (PaS). The sampling frequency was 50 Hz for mice 60584 and 60585 (MEC), 60961, 92227 and 92229 (VIS). For mice 59911, 59914 (MEC) and 59912 (PaS), the wheel tracking was not synchronized to the ongoing image acquisition; see 'Self-paced running behaviour under sensory-minimized conditions'. The obtained speed signal from the former two groups of mice was interpolated so that the speed values matched the downsampled imaging time points (sampling frequency = 7.73 Hz), and smoothed using a square kernel of 2 s width. A threshold

was applied such that all speed values that were smaller than 2 cm s$^{-1}$ were set to zero and all speed values larger than 2 cm s$^{-1}$ remained unchanged. We decided to threshold for immobility at a non-zero speed value (2 cm s$^{-1}$) in order to avoid classifying as running behaviour frames that only had minor movements of the wheel ('twitches'), which were detected when mice slightly moved on the wheel but did not fully engage in locomotion. The threshold that we used is consistent with the one used in other studies, as in ref. 16.

The speed signal obtained after applying the threshold was used to define immobility (running) bouts as the set of consecutive time points (bin size = 129 ms) for which the speed was equal to (larger than) zero (a similar approach was used in ref. 16). We found that the median of velocities was 0 cm s$^{-1}$ when all velocity values across the 10 MEC oscillatory sessions (over 3 mice) for which we had imaging data synchronized with behavioural data were pooled. This is because for some of the sessions the mice were immobile for most of the session.

When the threshold for immobility (2 cm s$^{-1}$, see above) was discarded (that is, set to 0 cm s$^{-1}$), the median was 1.3 cm s$^{-1}$—that is, still very low. In the absence of a threshold, our main result, which is that the oscillatory sequences traverse epochs of running and immobility, remained the same (median of probability of sequences during running = 0.85; median of probability of sequences during immobility = 0.65; two sample Wilcoxon signed-rank test on the probability of sequences for running versus immobility, $n$ = 10 oscillatory sessions over the 3 mice that had the tracking synchronized to imaging, $P$ = 0.002, $W$ = 55).

The acceleration was numerically calculated as the first derivative of the speed signal. Notice that in this case no interpolation was needed.

Because the available data did not have enough statistical power, it was not possible to compare the behaviour of the mice, for example in terms of its running speed and acceleration, between periods with and without ongoing oscillatory sequences.

Finally, mice that were imaged from the PaS or VIS performed the same minimalistic self-paced running task as the mice that were imaged from the MEC recordings. The range of speed values in PaS or VIS mice across sessions = 0–58.6 cm s$^{-1}$ (PaS) or 0–60.3 cm s$^{-1}$ (VIS); median number of completed laps on rotating wheel in PaS or VIS mice across sessions = 145 (PaS) or 104 (VIS); maximum number of completed laps on rotating wheel in PaS or VIS mice across sessions = 502 (PaS) or 1,743 (VIS). These values are reported for MEC mice in the legend of Extended Data Fig. 2a.

### Estimation of the probability of observing oscillatory sequences
To determine whether the MEC oscillatory sequences were observed during different behavioural states, the probability of observing the oscillatory sequences was calculated conditioned on whether the mouse was running or immobile. For each oscillatory session with behavioural tracking synchronized to the imaging data (10 sessions over 3 mice, see 'Self-paced running behaviour under sensory-minimized conditions' and 'Oscillation score'), all individual sequences were identified (see 'Identification of individual sequences'). The subset of time bins that belonged to individual sequences were extracted and labelled as oscillation (bin size = 129 ms). The fraction of bins labelled as oscillation bins was 0.73 ± 0.07 (mean ± s.e.m., n = 10 sessions). Next, a second label was assigned to the time bins depending on whether they occurred during running or immobility bouts (bins labelled 'running' or 'immobility', respectively, see 'Estimation of number of completed laps on the wheel, speed and acceleration'). The fraction of bins labelled as running = 0.43 ± 0.09, mean ± s.e.m., $n$ = 10 sessions. After applying this procedure, each time bin had two labels, one indicating the running behaviour, and one indicating the presence (or absence) of oscillatory sequences. To estimate the probability of observing the oscillatory sequences conditioned on the mouse's running behaviour, all bins labelled as running or immobility were identified and from each subset, the fraction of bins labelled as oscillation was calculated. These probabilities were computed for each session separately.

### Sequences during immobility bouts of different lengths
The oscillatory sequences occurred both during running and immobility bouts. To quantify the extent to which individual sequences progressed during different lengths of immobility bouts, the following procedure was adopted. First, for each session, all immobility bouts were identified and assigned to bins of different lengths (see 'Estimation of number of completed laps on the wheel, speed and acceleration'; length bins = 0–3 s, 3–5 s, 5–10 s, 10–15 s, 15–20 s, >25 s). Second, all individual sequences were identified (see 'Identification of individual sequences'). Third, for each session and each length bin, the fraction of immobility bouts that were fully occupied by uninterrupted sequences was calculated. To estimate significance, for each session the time bins that belonged to all individual sequences were temporally shuffled. The third step of the procedure described above was performed for 500 shuffle iterations per session. In Fig. 4c, the recorded data has 10 data points per length bin, and the shuffled data has 5,000 data points per length bin, since 500 shuffled realizations per session were pooled.

### Analysis of speed and sequence onset
To determine whether the onset of the MEC oscillatory sequences was modulated by the mouse's running speed, changes in speed before and after sequence onset were investigated. For each session all individual sequences were identified (see 'Identification of individual sequences') and for each sequence the mean speed over windows of 10 s before and after sequence onset was calculated. Because no differences in the mean speed were observed before and after onset (Extended Data Fig. 2f left panel), we next determined whether changes in speed were correlated with the onset of sequence epochs, which were defined as epochs with uninterrupted sequences—that is, epochs with recurring sequences. The same analysis described above was repeated but only for the subset of sequences that were 10 s or more apart—that is, for sequences that belonged to different epochs.

The obtained results remained unchanged when the analysis was performed for 2 s windows before and after sequence onset.

We complemented this analysis by investigating whether new epochs of sequences were more likely to be initiated during running bouts. In each of the 10 oscillatory sessions we first identified all running and immobility bouts that were 20 s long, or longer. We then counted the number of times that a sequence onset occurred in each behavioural state. For this analysis we only considered sequences that were not preceded by other sequences (sequences that were 10 s apart or more). Results were upheld with running and immobility bouts of 40 s or longer, in which case sequence onset was 2.8 times more frequent during running.

### Manifold visualization for example session in VIS and PaS
To visualize whether the topology of the manifold underlying the population activity in example sessions recorded in VIS and PaS was also a ring, PCA was used and a similar procedure to the one described in 'Manifold visualization for MEC sessions' was adopted.

For each example session, one corresponding to VIS and one corresponding to PaS (Fig. 5e,f), PCA was applied to the matrix of calcium activity, which first had each row convolved with a gaussian kernel of width equal to four times 8.5 s, which is the mean oscillation bin size computed across oscillatory sessions (see 'Oscillation bin size'). Neural activity was projected onto the embedding generated by PC1 and PC2. Extended Data Fig. 11d,e shows the absence of a ring-shaped manifold in VIS and PaS example sessions.

### Co-activity and synchronization in PaS and VIS sessions
Sessions recorded in PaS and VIS did not exhibit oscillatory sequences. To further characterize their population activity, synchronization and neural co-activity were calculated.

**Synchronization.** Neural synchronization was calculated as the absolute value of the Pearson correlation between the calcium activity of pairs of cells (bin size = 129 ms). For each session, the Pearson correlation was calculated for all pairs of calcium activity (correlations with the same calcium activity were not considered) and used to build a distribution of synchronization values. In Extended Data Fig. 11j, these distributions were averaged across sessions for each brain area separately.

**Co-activity.** For each time bin in a session (bin size = 129 ms) the co-activity was calculated as the number of cells that had simultaneous calcium events divided by the total number of recorded cells in the session. This number represented the fraction of cells that was active in individual time bins. Using all time bins of the session, a distribution of co-activity values was calculated. In Extended Data Fig. 11k, the distributions were averaged across sessions for each brain area separately.

## Model

To determine whether long sequences act as a template for the formation of given activity patterns in a neural population, we built a simple perceptron model in which 500 units were connected to an output unit (Extended Data Fig. 12a). There was a total of 500 weights in the network, one per input unit. The total simulation time was 120 s, with 3,588 simulation steps and a time step of 33.44 ms (original time step was 129 ms, to mimic the bin size used in calcium data, rescaled so that the length of one of the input sequences was 120 s, similar to the length of the sequences in Fig. 2b). The response of the output unit was given by $R = WX$, where $W$ was the vector of weights, and $X$ the matrix of input activity (each column is a time step, each row is the activity of one input unit). The weights were trained such that the output unit performed one of two target responses (see below). For each target, we trained the model using as input periodic sequences with 5 different lengths (one length per training), covering the range from very slow to very fast as compared to the characteristic time scale of the targets (100 s).

**Inputs.** The activity of input unit $i$ was represented by a Gaussian: $x_i(t) = e^{-\frac{(t-\mu_i)^2}{2\sigma_i^2}}$, $1 \le i \le 500$, $0 \le t \le 240$ s, $\sigma_i = \sigma = 7.6$ s, $\forall i$. Across input units, the means of the Gaussians $\mu_i$ were temporally displaced such that, all together: (1) units fired in a sequence, and (2) the distance between the means of two consecutive cells in the sequence was the same for all pairs of consecutive cells.

This sequence was the slowest of the 5 sequence lengths we considered. Using this sequence as template, in order to build slower and periodic sequences we compressed the template and repeated it periodically by a factor of 2, 3, 4 and 8, to generate faster and periodic sequences of lengths 120, 60, 40 and 30 s respectively.

**Targets.** Two target responses were considered: ramp and Ornstein–Uhlenbeck process.
**Ramp.** The output neuron linearly increased its activity such that it was equal to 0 at time step = 0 (0 s), and to 1 at time step = 2,990 (100 s).

$$F_R(t) = \frac{t}{100}$$

**Ornstein–Uhlenbeck process.** Unlike the first target, which was deterministic, the second target was stochastic and generated by an Ornstein–Uhlenbeck process.

$$\frac{dF_{OU}}{dt} = \frac{\mu_{OU} - F_{OU}(t)}{\tau} + \sigma_{OU}\xi(t)$$

where $\mu_{OU} = 1$ denotes the long-term mean, $\xi$ is a white noise of zero mean and variance $\sigma_{OU} = 0.005$, and $\tau = 25.6$ s denotes the correlation time.

**Training of weights.** The weights between the inputs and the output unit were trained such that the output unit performed one of the two target responses explained above. At the end each of the 1,000 learning iterations, the weights were updated through the perceptron learning rule $\Delta w_i = \eta e x_i$, where $x_i$ was the input from neuron $i$, $1 \le i \le 500$, and $\eta = 1$ was the learning rate. In each learning iteration, the error $e$ was calculated as the sum over time steps $t$ of the difference between the target response and the output response—that is, $e = \sum_t T(t) - WX(t)$, where $T(t)$ is the target response (either the ramp or the Ornstein–Uhlenbeck process) at time point $t$, and $X(t)$ is the vector of input activity at time point $t$. The mean total error plotted in Extended Data Fig. 12d was calculated as the mean error over the last 100 learning iterations.

## Data analysis and statistical analysis

Data analyses were performed with custom-written scripts in Python and Matlab (R2021b). Results were expressed as the mean ± s.e.m. unless indicated otherwise. Statistical analysis was performed using MATLAB and $P$ values are indicated in the figure legends and figures (NS: $P > 0.05$; *$P < 0.05$, **$P < 0.01$, ***$P < 0.001$). For data that displayed no Gaussian distribution and that was unpaired, the Wilcoxon rank-sum test was used. For paired data or one-sampled data, the Wilcoxon signed-rank test was used. Two-tailed tests were used unless otherwise indicated. Correlations were determined using Pearson or Spearman correlations. Friedman tests were used for analyses between groups. The Bonferroni correction was used when multiple comparisons were performed.

Power analysis was not used to determine sample sizes. The study did not involve any experimental subject groups; therefore, random allocation and experimenter blinding did not apply and were not performed.

## Reporting summary

Further information on research design is available in the Nature Portfolio Reporting Summary linked to this article.

## Data availability

The datasets generated during the current study will be available after publication, on EBRAINS (https://doi.org/10.25493/SKKX-4W3). Source data are provided with this paper.

## Code availability

Code for reproducing the analyses in this article are available through this link: https://github.com/soledadgcogno/Ultraslow-oscillatory-sequences.git.

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

**Acknowledgements** The authors thank D. W. Tank for sharing hardware, software and advice when 2-photon imaging was introduced to the lab in 2013–14; A. Tsao, G. B. Keller and T. Bonhoeffer for subsequent help in setting up 2-photon imaging procedures; Ø. Høydal and E. Ranheim Skytøen for recent help with setting up Neuropixels recordings in mice; W. Zong, I. Davidovich, Y. Roudi and E. Kropff for discussion; Y. Burak for discussion and comments on the manuscript; and S. Ball, K. Haugen, E. Holmberg, K. Jenssen, E. Kråkvik, I. Ulsaker-Janke and H. Waade for technical assistance. The work was supported by a Synergy Grant to E.I.M. from the European Research Council ERC ('KILONEURONS', grant agreement no. 951319), FRIPRO grants to E.I.M. (grant no. 286225) and M.-B.M. (grant no. 300394/H10), a Centre of Excellence grant to M.-B.M. and E.I.M. (Centre of Neural Computation, grant no. 223262), and a National Infrastructure grant to E.I.M. and M.-B.M. from the Research Council of Norway (NORBRAIN, grant no. 295721), as well as the Kavli Foundation (M.-B.M. and E.I.M.), a direct contribution to M.-B.M. and E.I.M. from the Ministry of Education and Research of Norway, and an ERC Starting Grant (ERC-ST2019 850769) and an Eccellenza Grant from the Swiss National Science Foundation (PCEGP3_194220) to F.D.

**Author contributions** F.D., H.A.O., M.-B.M. and E.I.M. planned and designed the initial experiments, with later input from S.G.C. Experiments were performed by F.D., R.I.J., S.O.A. and A.L. H.A.O. developed hardware and imaging software, preprocessed the data and performed initial analysis. H.A.O.'s contribution to data preprocessing and A.L.'s contribution to data collection were equal. S.G.C., M.-B.M. and E.I.M. conceptualized and designed analyses, with initial input from F.D. S.G.C. performed analyses of neural activity. S.G.C and C.C. developed the model. F.D. and A.L. performed histological analyses. S.G.C., M.-B.M. and E.I.M. interpreted data, with initial input from F.D. S.G.C. and F.D. visualized data. S.G.C. and E.I.M. wrote the paper, with initial contributions from F.D. and with periodic input from all authors. M.-B.M. and E.I.M. supervised and funded the project.

**Competing interests** The authors declare no competing interests.

**Additional information**
**Correspondence and requests for materials** should be addressed to Soledad Gonzalo Cogno, May-Britt Moser or Edvard I. Moser.

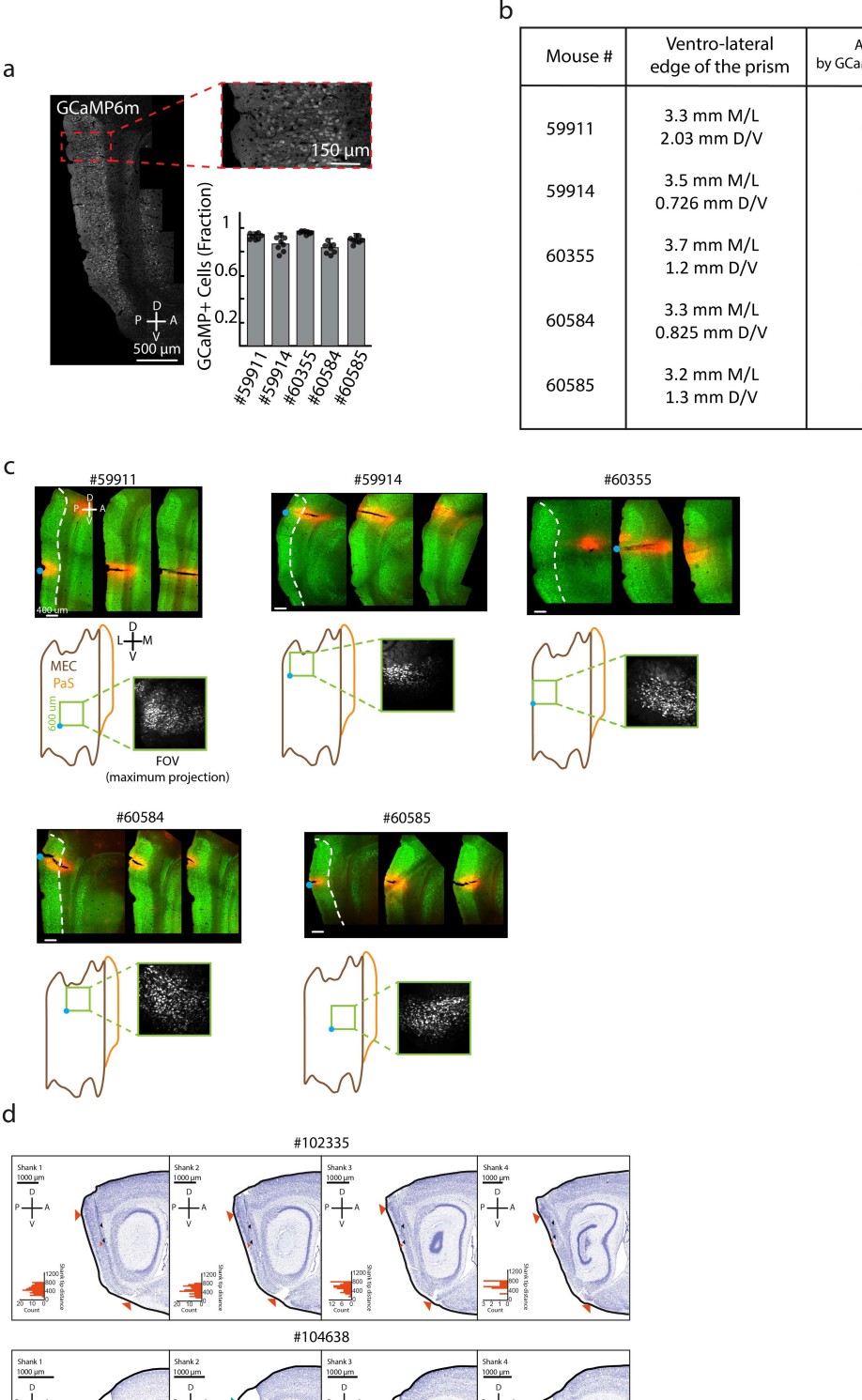

**Extended Data Fig. 1** | See next page for caption.

**Extended Data Fig. 1 | Histology showing imaging locations for each animal in the MEC group. a**. Left: Representative image indicating GCaMP6m expression in the superficial layers of the MEC upon local viral injection at postnatal day P1 (sagittal section). Images were acquired with a 20× objective mounted on a confocal laser scanning microscope LSM 880 (Zeiss), operated by ZEN 3 software (blue edition). Red inset, top right: 60× magnification of the most dorsal portion of the MEC. Bottom right: Fraction of MEC neurons (Nissl +) expressing GCaMP6m; data are shown for all 5 animals with MEC imaging. Data are presented as mean values, error bar indicates the S.D. calculated across multiple (n = 8) adjacent slices. Each dot represents one slice. **b**. Location of the ventro-lateral edge of the prism in stereotactic coordinates, and area of the FoV occupied by cells expressing GCaMP6m. Data are shown for each MEC-imaged animal. Mouse #59911 had no oscillatory sequences. **c**. Prism location in mice that underwent calcium imaging in MEC in the left hemisphere. Top: Maximum of 50 μm thick sagittal brain sections. For each of the 5 mice in (b), 3 sections, shown from lateral (left) to medial (right), were acquired with an LSM 880, 20×. A DiI-coated piano wire pin was inserted at the ventrolateral corner of FoV to enable identification of the FoV on histology sections. Green is GCaMP6m signal, red is DiI signal. Scale bar is 400 μm. The white stippled line encapsulates the superficial layers of MEC. The blue dot adjacent to the leftmost image of the series marks the location of the ventro-lateral corner of the prism. Bottom: estimated location of the FoV for two-photon imaging, projected onto a flat map encompassing MEC (brown outline) and parasubiculum (PaS, orange outline). The blue dot marks the location of the pin used to demarcate the most lateral-ventral border of the prism, while the green square inset is the microscope's FoV. Inset image shows the maximum intensity projections of the FoV. Anteroposterior (AP), Mediolateral (ML), and dorso–ventral (DV) axes are indicated in panels (a) and (c). **d.** Micrographs of Cresylviolet stained sagittal brain sections from all 2 mice implanted with four-shank Neuropixels 2.0 silicon probes in the left hemisphere. Sections are organised from the most laterally placed shank(s) (left) to the most medially placed shank(s) (right). Mouse ID, shank number, and scale bar (1000 μm) are indicated next to each section. The brain of one mouse (#104638) was damaged during extraction, and parts of the MEC and cortex are missing from the section. Coloured arrows indicate MEC borders (dorsal, ventral) and the identified or estimated probe tip in the section. Black arrows indicate estimated dorsoventral range of the probe's active recording sites (as indicated by the insert). For each section, inserts show the number of units recorded at each depth of the probe shank (histogram bin size = 60 μm). Note that the anatomical location of probe shanks can only be approximately estimated, and indicated unit locations are subject to measurement error, e.g., due to the shank tips exiting the cortex, the brain shrinking during perfusion and error in estimating the position of the tip of the probe. Stippled lines indicate borders between brain regions (MEC, medial entorhinal cortex; LEC, lateral entorhinal cortex; PaS, parasubiculum, HF, hippocampal formation; PoR, postrhinal cortex; VISpl, posterolateral visual area; TR, postpiriform transition area; CoA, cortical amygdalar area; PA, posterior amygdalar nucleus). D = dorsal; V = ventral; A = anterior; P = posterior.

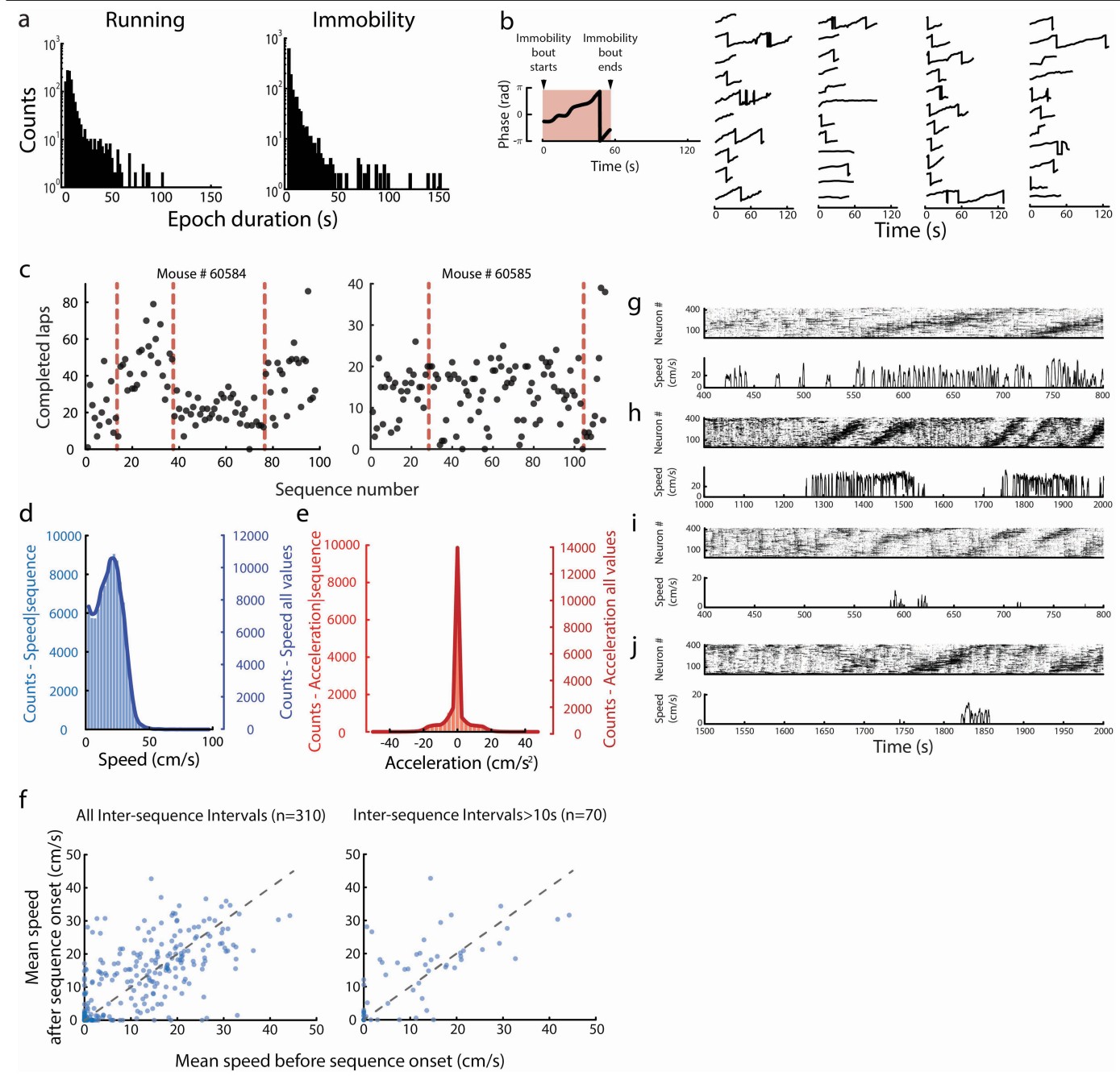

**Extended Data Fig. 2** | See next page for caption.

**Extended Data Fig. 2 | Relationship between the oscillatory sequences and behavior. a**. Quantification of the animals' behavior during head-fixation on the wheel. Duration of epochs of running (speed ≥ 2 cm/s, left) and immobility (speed < 2 cm/s, right) for 10 oscillatory sessions over the 3 animals in which behavioral tracking was synchronized with imaging (1289 running bouts and 1286 immobility bouts in total). Each count is an epoch, and one epoch is obtained by concatenating consecutive time bins with the same behaviour (running or immobility, bin size = 129 ms). For each of the 10 sessions the smallest speed value was always 0 cm/s. The largest speed value ranged from 16.4 to 75.3 cm/s. The median calculated over the entire session ranged from 0 cm/s (in 4 out of 10 sessions) to 18.8 cm/s. Across the 10 sessions, the median of speed values was 0 cm/s (indicating that some of the animals spent much of the session time being immobile, yet those animals exhibited oscillatory sequences too, e.g. animal #60355, Extended Data Fig. 5a; see also Fig. 4a and c). The median speed during running epochs was 7.8 cm/s. The acceleration values ranged from −86.3 to 108.9 cm/s², with a median of 0 cm/s² for all the data as well as the running epochs specifically. **b**. Left: Schematic of the change in phase of the oscillation during immobility epochs that were longer than 25 s and that occurred during the oscillatory sequences. Right: 44 of these epochs from the same 3 mice as in (a). As in the schematic on the left, each line represents the progression of the phase of the oscillation (from −π to π rad) as a function of time. The start of each immobility epoch is aligned at t = 0, and the epoch lasts for as long as the line continues. Different epochs have different lengths, covering a range from 25 s to 258 s. For visualization purposes only the first 120 s are displayed (3 of the epochs were truncated; these had durations of 127.9 (first column, second row), 258.2 (third column, bottom row), 136.1 s (fourth column, second row)). Sudden transitions from π to −π rad reflect the periodic nature of the sequences. **c**. Number of completed laps on the wheel per sequence as a function of the sequence number after pooling sessions (range of completed laps on rotating wheel across 10 sessions = 10 to 1164 laps, median = 624 laps). Sessions are pooled for each animal separately (mouse #60584, 4 sessions; mouse #60585, 3 sessions; the third animal is shown in Fig. 4d). Each dot indicates one individual sequence. The dashed line indicates separation between sessions. A number of laps equal to 1 would indicate an approximate one-to-one mapping between the position on the wheel and the progression of one full sequence. **d**. To determine if sequences are associated with specific running speeds, we extracted all time bins participating in oscillatory sequences and calculated the distribution of observed speed values during those bins (blue bars; n = 167389 time bins concatenated across 314 sequences pooled over 10 oscillatory sessions, over 3 animals, bin size = 129 ms). This distribution was almost identical to the distribution of speed values observed during the full length of the sessions, which also included epochs without the oscillatory sequences (blue solid line, with and without oscillatory sequences; n = 238505 time bins across 10 oscillatory sessions, over 3 animals, bin size = 129 ms). **e**. As in (d) but for the distribution of acceleration values. There is no difference in the range of acceleration values during parts of the session with oscillatory sequences. **f**. Left: To determine whether the oscillatory sequences are modulated by onset of running we calculated the mean running speed during time intervals of 10 s right before and right after the sequence onset (one sample Wilcoxon signed-rank test on the difference between speed before and after sequence onset, n = 310 equence onsets over 10 sessions from 3 animals, p = 0.82, W = 25). Right: Same as left but only for sequences that were 10 s or more apart, i.e. for sequences belonging to different oscillatory epochs (one sample Wilcoxon signed-rank test on the difference between speed before and after sequence onset, n = 70 sequence onsets over 10 sessions from 3 animals, p = 0.12, W = 857). Note that there is no systematic change in speed after onset of sequences. Results remain unchanged if the analysis is repeated for 2 s windows before and after sequence onset (Analysis for all sequences: one sample Wilcoxon signed-rank test on the difference between speed before and after sequence onset, n = 310 equence onsets over 10 sessions from 3 animals, p = 0.82, W = 25; Analysis for all sequences that were 10 s or more apart, one sample Wilcoxon signed-rank test, n = 70 sequence onsets over 10 sessions from 3 animals, p = 1.0, W = 0). **g–j**. Examples of sections of sessions with increased speed after sequence onset (exceptions from the general pattern shown in (f)). Top of each panel: Raster plots, symbols as in Fig. 2a (bin size = 129 ms). Bottom of each panel: Instantaneous speed of the animal during the recording in the top panel. Length of the displayed section was 400, 1000, 400 and 500 s, respectively, for (g–j). Notice that while speed is higher after onset of the sequence in these examples, the increase of speed does not always occur right after sequence onset, but sometimes before (g,h), and sometimes tens of seconds after (i,j). Analyses were restricted to 10 oscillatory sessions in 3 animals, for which the behavioural tracking was synchronized to the imaging (Methods).

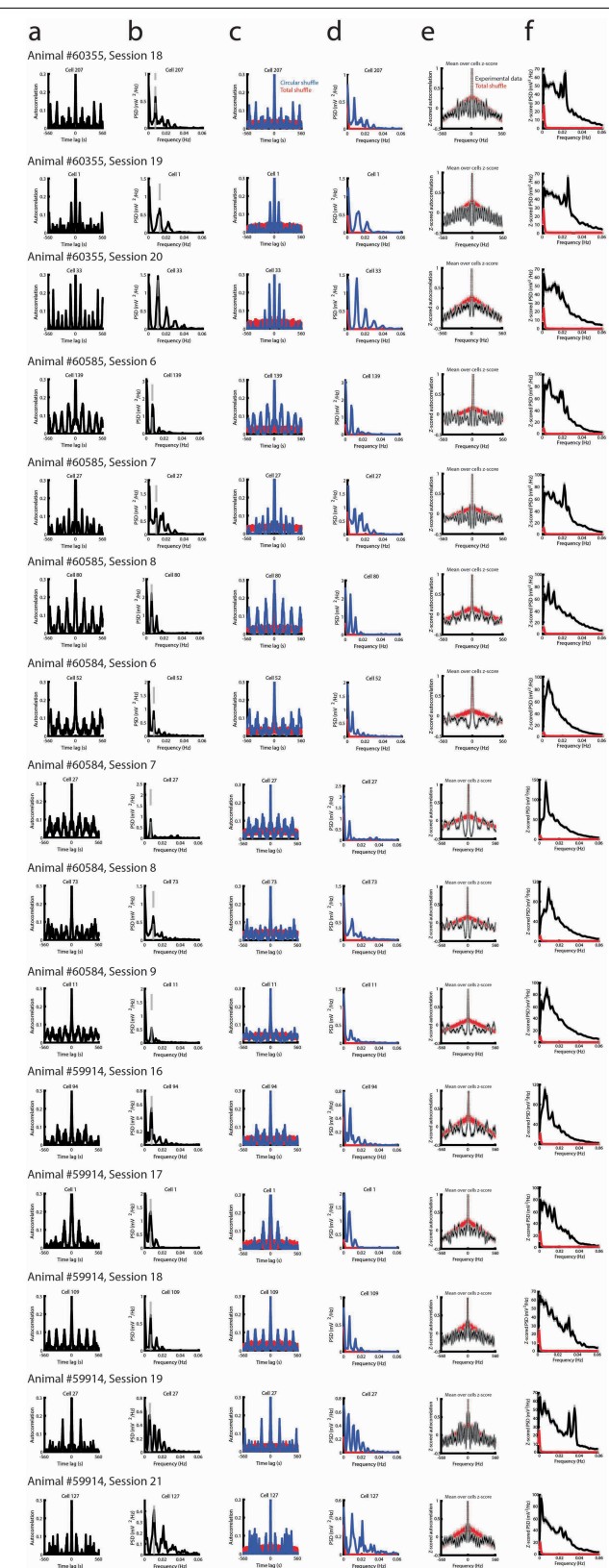

**Extended Data Fig. 3** | See next page for caption.

**Extended Data Fig. 3 | Examples of ultraslow oscillations in single cell calcium activity. a**. Autocorrelation of 15 example cells' calcium activity (one per oscillatory session). **b**. PSD calculated on the autocorrelation of the example cell shown in (a). The dashed line indicates the frequency at which the PSD peaks. Note that the peak is at a frequency <0.1 Hz. **c**. As in (a) but for the signal obtained after the calcium activity was circularly shuffled (blue) or shuffled by destroying the inter calcium event intervals (red). Note that circularly shuffling the calcium activity preserves its periodicity. **d**. PSD calculated on the autocorrelations in (c). Blue indicates circularly shuffled data. Red indicates data that was shuffled by destroying the inter calcium event intervals. **e**. Mean z-scored autocorrelation calculated over all recorded cells in the session. Error bars indicate S.E.M. Black: Experimental data. Red: Shuffled data (obtained by destroying the inter calcium event intervals). **f**. Mean z-scored PSD calculated over all recorded cells in the session. For each cell the PSD was calculated on the autocorrelation of the cell's calcium activity. Error bars indicate S.E.M. Color convention as in (e). Each row shows data from one oscillatory session (15 rows in total, each row corresponds to one oscillatory session). Animal number and session number are indicated at the top.

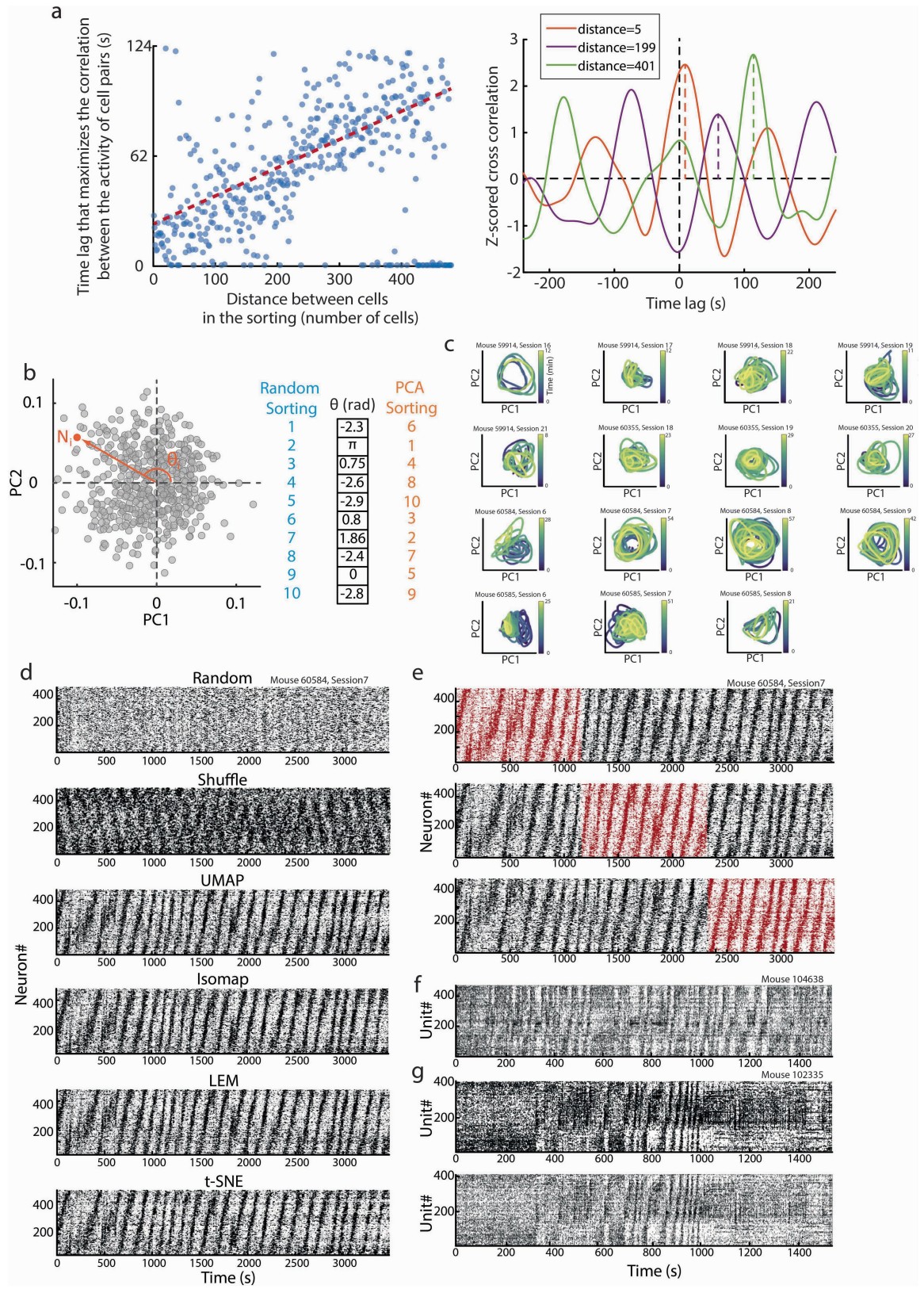

**Extended Data Fig. 4** | See next page for caption.

**Extended Data Fig. 4 | Oscillatory sequences shown by cell sorting based on correlation or dimensionality reduction. a**. Left: Because neural activity progresses sequentially, the time lag that maximizes the correlation between the calcium activity of pairs of cells increases with their distance in the correlation sorting. Sorting is performed as in Fig. 2a. Time lag is expressed in seconds, distance is expressed as the number of cells between the two cells in the sorting. Notice that for large distances (e.g. > 300 cells), the time lag to peak correlation is either larger than 60 s or close to zero. This bimodality is due to the periodicity of the MEC sequences. The dashed line indicates a linear regression ($n = 301$ cell pairs, $R^2 = 0.17$, $p = 2 \times 10^{-14}$, two-sided t-test. The line was fitted to the intermediate samples to avoid the effect of the periodic boundary conditions). Right: The cross correlation between the calcium activity of pairs of cells is oscillatory and temporally shifted. Examples are shown for 3 cell pairs with different distances in the sorting based on correlation values. Orange: cells are 5 cells apart; purple: cells are 199 cells apart; green: cells are 401 cells apart. The dotted line indicates the time lag at which the cross correlation peaks within the first peak. Note that the larger the distance between the cells in the sorting, the larger the time lag that maximizes the cross correlation. **b**. Schematic representation of the "PCA method". Principal component analysis (PCA) was applied to the binarized matrix of deconvolved calcium activity ("matrix of calcium activity") of individual sessions by considering every neuron as a variable, and every time point as an observation. The first two principal components (PC1, PC2) were identified. In the plane defined by PC1 and PC2 (left), the loadings of each neuron defines a vector, which has an associated angle $\theta \in [-\pi, \pi)$ rad with respect to the axis of PC1 (in the schematic, neuron $N_i$ (orange) is characterized by an angle $\theta_i$). Neurons were sorted according to their angles $\theta$ in a descending order (right). Cyan: neuron sorting before application of the PCA method. Orange: neuron sorting after the application of the PCA method. **c**. Projection of neural activity during the oscillatory sequences onto a low-dimensional embedding generated by the first two principal components obtained by applying PCA to the matrix of calcium activity of each session. Each plot shows one session; all 15 oscillatory sessions from the calcium imaging data set are presented. Time is color-coded and shown in minutes, and the temporal range corresponds to all concatenated epochs with oscillatory sequences in the session. Neural trajectories are often circular, with population activity propagating along a ring-shaped manifold. The ring-shaped manifold became even more salient when we applied a non-linear dimensionality reduction method (Laplacian Eigenmaps, LEM) instead of PCA to the data (Fig. 2c, right), suggesting that at least some of the data might lie on a curved surface. **d**. Oscillatory sequences

are not revealed with a random sorting of the cells (first row) or when the PCA sorting method is applied to circularly shuffled data (second row). Oscillatory sequences similar to those of Fig. 2a,b (with correlation sorting or PCA method) are recovered when neurons are sorted according to non-linear dimensionality reduction techniques (UMAP, Isomap, LEM, t-SNE, third to sixth row). Each row of each raster plot is a neuron, whose calcium activity is plotted as a function of time (as in Fig. 2a). Every black dot represents a time bin where a neuron was active (bin size = 129 ms). **e**. Raster plot of calcium activity of the session presented in Fig. 2a. Neurons are sorted according to the PCA method. For calculating the sorting, only the first (top), second (middle) and third (bottom) third of the data was used. The portion of the data used for calculating the sorting is indicated in red. Otherwise, conventions are as in Fig. 2a. This visualization was extended to a quantification for all sessions. For each session we calculated the sortings using (i) all data, (ii) the first half of the data, (iii) the second half of the data. Next we calculated the correlation between the distances in the different sortings. If sortings obtained with different chunks of data preserve the ordering of the neurons, we would expect high correlation values. We compared the obtained correlation values with the 95[th] percentile of a shuffled distribution obtained by shuffling the position of the cells in the sortings. When comparing sorting (i) vs. sorting (ii), (i) vs. (iii), and (ii) vs. (iii), all oscillatory sessions (15 of 15) were above the cutoff of significance (see Methods). The high correlation values obtained in these distance estimates provide support to the fact that using different chunks of data for sorting the cells unveils the same dynamics. **f**. Neuropixels recording showing ultraslow sequences without prior smoothing of the data. Same data as in Fig. 2f. While in Fig. 2f spike trains were first convolved with a Gaussian kernel of width equal to 5 s and next binarized according to the mean plus one standard deviation (Methods), here the spike trains are not convolved with a Gaussian kernel. The bin size is 120 ms. The threshold for binarization of the spike trains is equal to the mean + 1.5 standard deviations. Sorting and conventions as in Fig. 2f. Example session from animal #104638. Sequences are still visible. This session had an oscillation score of 1.0. In this session we identified 12 sequences of durations spanning 18–43 s. **g**. Oscillatory sequences from a Neuropixels recording in a different mouse than in (f) (and Fig. 2f). Top: Similar to Fig. 2f, but from mouse #102335 (n = 410 units). Bottom: Similar to (f), but for the same session as presented in the top panel, without prior smoothing of the data. This session had an oscillation score of 0.91 (see Methods). See comparable example sessions for calcium data in Extended Data Fig. 5a. In this session 9 sequences were identified, with durations ranging from 14 to 69 s.

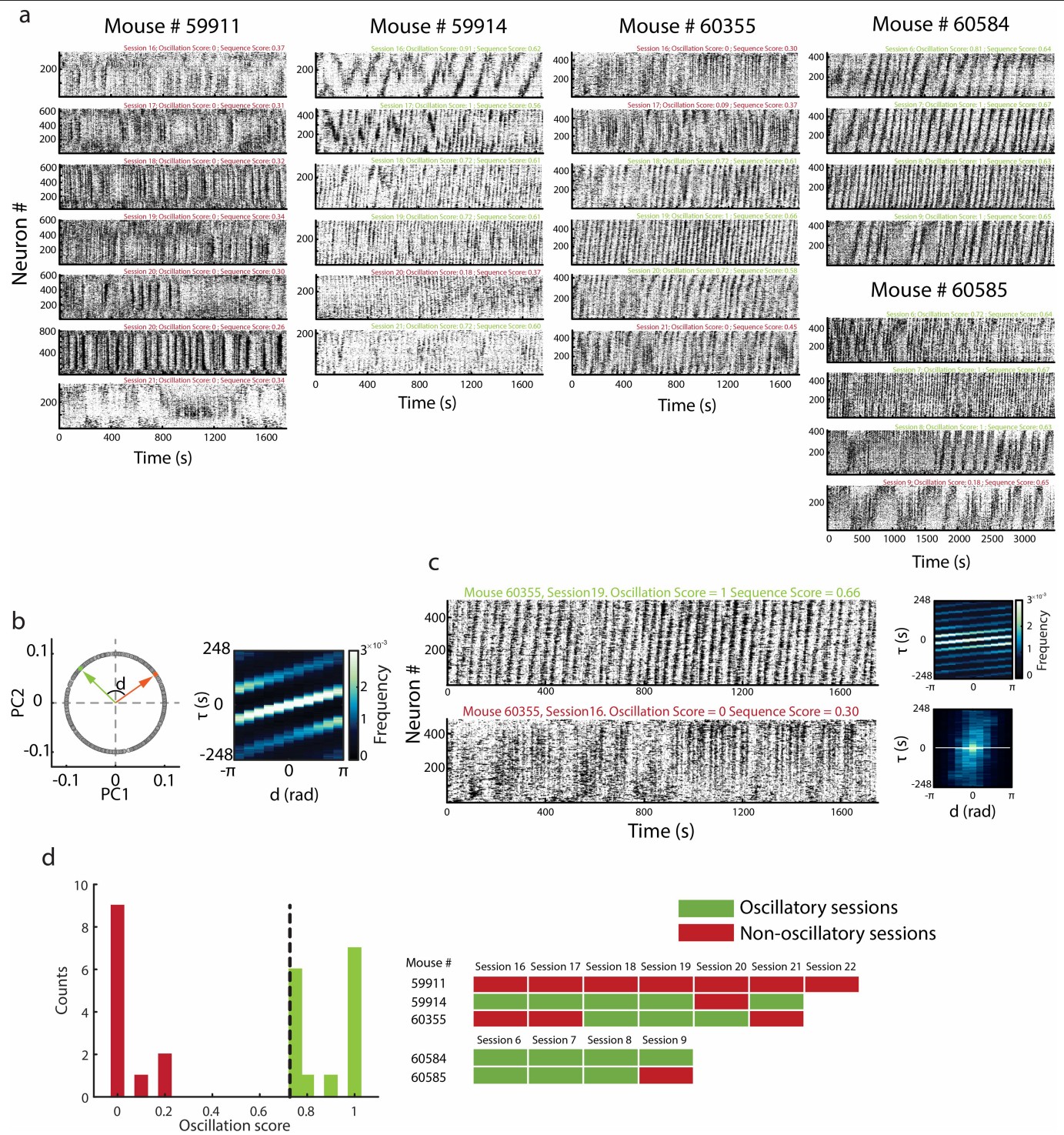

**Extended Data Fig. 5** | See next page for caption.

**Extended Data Fig. 5 | Sorted raster plots for the complete MEC calcium imaging dataset. a**. PCA-sorted raster plots (as in Fig. 2b) for all analysed sessions across the 5 animals in which MEC population activity was recorded, sorted by animals and day of recording. Session numbering starts the first day of habituation on the wheel, with either 5 or 15 habituation sessions. One session was recorded per day, and recordings were conducted on consecutive days. Note that sessions had lengths of approximately 1800 s or 3600 s. Oscillation score and sequence score were calculated for each session separately and are indicated at the top right corner of every plot. Scores colored in green correspond to sessions with oscillatory sequences (see panel d), scores colored in red to sessions without oscillatory sequences. **b**. Left: Distance $d$ between two neurons in the PCA sorting is calculated as the difference between the angles of the vectors defined by the loadings of each neuron on PC1 and PC2 with respect to PC1. The schematic shows the distance between two neurons, one in orange and the other in green. The length of the vectors is disregarded in this quantification. Right: Joint distribution of the time lag τ that maximizes the cross-correlation between the calcium activity of any given pair of neurons and their distance $d$ in the PCA sorting. Color code: normalized frequency, each count is a cell pair. The increasing relationship between τ and $d$ indicates sequential organization of neural activity. **c**. Example sessions with (top) and without (bottom) oscillatory sequences. These sessions were recorded in the same area of the MEC in the same animal, but on different days (Mouse #60355 in panel a). Left: Raster plots of the matrices of calcium activity. Right: Joint distributions of the time lag τ that maximizes the correlation between the calcium activity of any given pair of neurons and their distance $d$ in the PCA sorting (as in panel b). Color code: normalized frequency, each count is a cell pair. Notice the lack of linear pattern in the session without oscillatory sequences. **d**. Left: Distribution of oscillation scores for calcium-imaging sessions recorded in MEC (27 sessions in total over 5 animals). Each count is a session. The oscillation score quantifies the extent to which single cell calcium activity is periodic, and ranges from 0 (no oscillations) to 1 (oscillations). Dashed line: Threshold used for classifying sessions as oscillatory (oscillation score ≥ 0.72) or non-oscillatory sessions (oscillation score <0.72). The threshold was chosen based on the bimodal nature of the distribution (no values between 0.27 and 0.72). 12/27 sessions exhibited scores between 0 and 0.27 (no oscillatory sequences), and 15/27 sessions exhibited scores between 0.72 and 1 ('oscillatory sessions'). Right: List of sessions sorted by animal and number of sessions the animals experienced on the wheel. Session numbering as in (a). Red, sessions classified as not oscillatory; green, session classified as oscillatory.

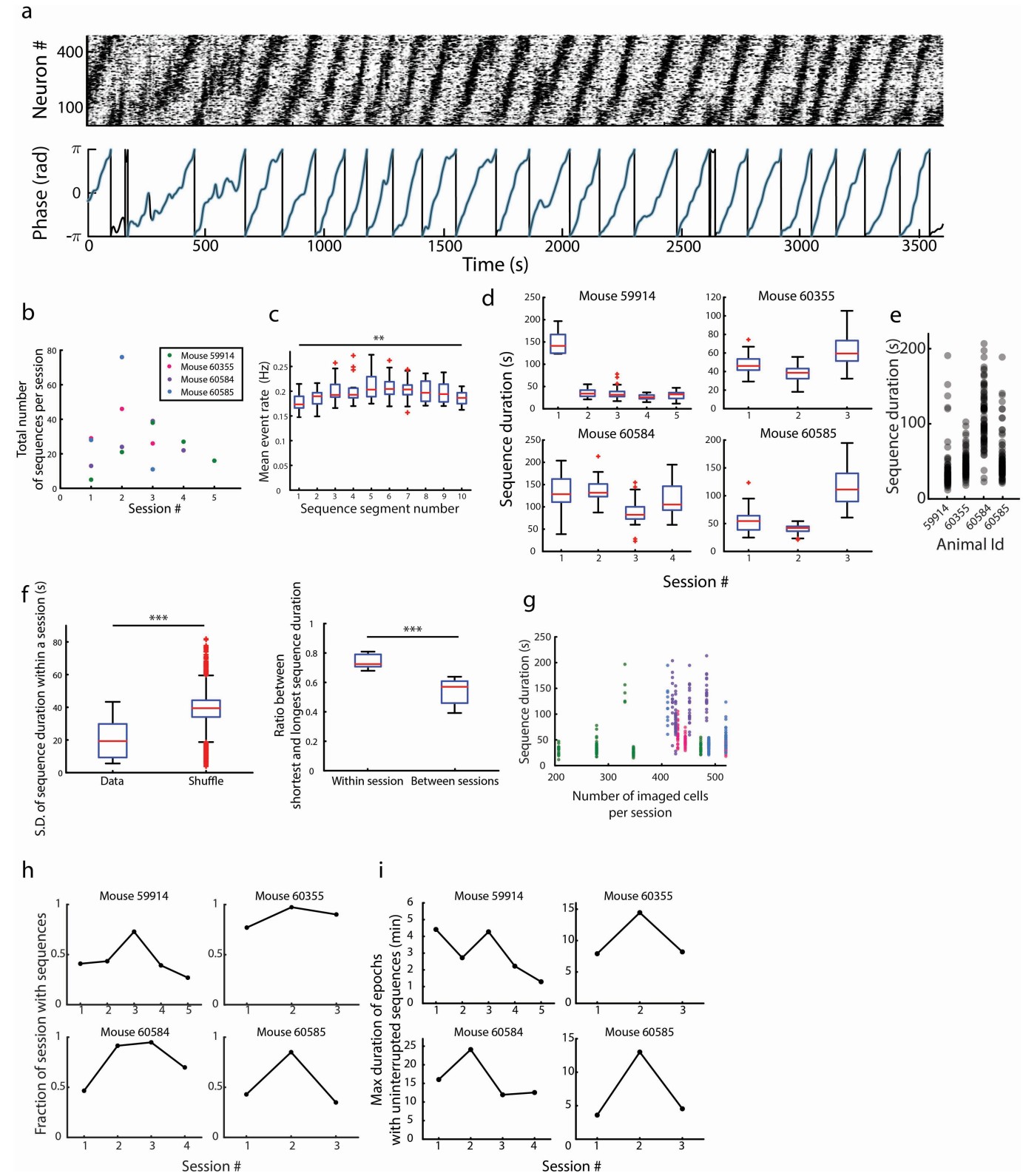

**Extended Data Fig. 6** | See next page for caption.

**Extended Data Fig. 6 | Identification of individual sequences and characterization of the oscillatory sequences. a**. Top: Raster plot of the PCA-sorted matrix of calcium activity of the example session in Fig. 2a. Bottom: Phase of the oscillation calculated on the session presented in the top panel is shown in black, and phase of individual sequences is colored in cyan (bin size = 129 ms). During one sequence the phase of the oscillation traversed smoothly [−π, π) rad. We identified individual sequences by extracting the subset of adjacent time bins where the phase of the oscillation increased smoothly within the range [−π, π) rad. First the phase of the oscillation was calculated across the entire session, second discontinuities in the succession of such phases were identified and used to extract putative sequences and third, putative sequences were classified as sequences if the phase of the oscillation progressed smoothly and in an ascending manner, allowing for the exception of small fluctuations (lower than 10% of 2π, e.g. as in the sequence at 500 s). Points of sustained activity were ignored. Fractions of sequences in which the phase of the oscillation traversed 50% or more of the range [−π, π) rad were also analysed (for example at the beginning of this session). **b**. Total number of individual sequences per session, across 15 oscillatory sessions. Animal number is color-coded. Note that 4 of 5 MEC calcium imaging animals had identifiable oscillatory sequences. **c**. Box plot showing mean event rate as a function of sequence segment for all 15 oscillatory sessions. Each sequence was divided into 10 segments of equal length, and for each sequence segment the mean event rate was calculated as the total number of calcium events across cells divided by the length of the segment and the number of recorded cells. Red lines indicate median across sessions, the bottom and top lines in blue (bounds of box) indicate lower and upper quartiles, respectively. The length of the whiskers indicates 1.5 times the interquartile range. Red crosses show outliers that lie more than 1.5 times outside the interquartile range. The mean event rate remained approximately constant across the length of the sequence. While a non-parametric analysis revealed an overall difference ($n = 15$ oscillatory sessions per segment, $p = 0.0052$, $\chi^2 = 23.5$, Friedman test), the rate change from the segment with minimum to maximum event rate was no more

than 18% and there were no significant differences in the event rate between pairs of segments (Wilcoxon rank-sum test with Bonferroni correction, p > 0.05 for all pairs). *** $p < 0.001$, ** $p < 0.01$, * $p < 0.05$, n.s. $p > 0.05$. **d**. Box plot of sequence duration, for the 15 oscillatory sessions. Note the relatively fixed duration of sequences in individual sessions. Box plot symbols as in (c). **e**. Sequence durations shown separately for each animal with oscillatory sequences (421 sequences in total over 5 animals, only 4 presented sequences). For each animal all oscillatory sessions were pooled. Sequence duration was heterogenous across sessions and animals. **f**. Left: Box plot of the standard deviation of sequence duration within a session, in experimental and shuffled data. The standard deviation of sequence duration is smaller in the experimental data ($n = 15$ oscillatory sessions, 7500 shuffle realizations where sequences were randomly reassigned to the 15 sessions, preserving the original number of sequences per session, $p = 1.8 \times 10^{-7}$, $Z = 5.08$, one-tailed Wilcoxon rank-sum test). Right: Box plot of the ratio between the shortest sequence duration and the longest sequence duration for all pairs of sequences within and between sessions. This fraction is larger for sequence pairs in the within-session group ($n = 15$ oscillatory sessions, the mean fraction per session and group was calculated separately, $p = 1.7 \times 10^{-6}$, $Z = 4.64$, one-tailed Wilcoxon rank-sum test). Notice that for each sequence pair, the larger this ratio, the more similar the length of the sequences are. Symbols as in (c). **g**. Sequence duration is not correlated with the number of recorded cells in the session ($n = 421$ sequences across 15 oscillatory sessions, $\rho = 0.02$, $p = 0.64$, Spearman correlation, two-sided t-test). Each dot is a sequence. Animal number is color-coded as in (b). **h**. Fraction of the session in which the MEC population engaged in the oscillatory sequences. Session length was 30 min for mice 59914 and 60355, and 60 min for mice 60584 and 60585. The fraction of session time with oscillatory sequences varied within and across animals. **i**. Duration of the longest epoch with uninterrupted oscillatory sequences. Only epochs that met the strict criterion of no separation between sequences were considered. Sequences could progress uninterruptedly for minutes in each of the animals and span up to 23 consecutive sequences.

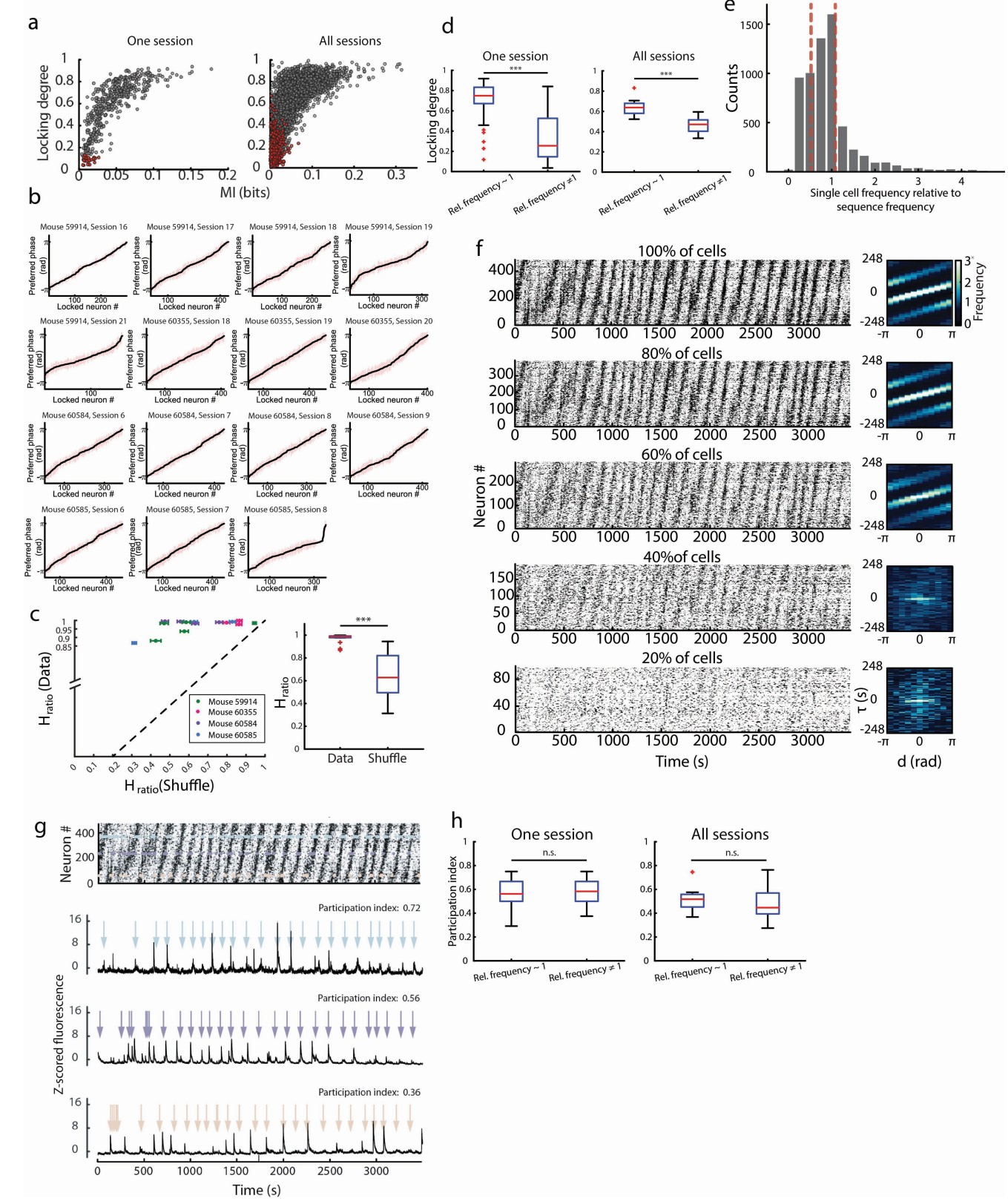

**Extended Data Fig. 7** | See next page for caption.

**Extended Data Fig. 7 | Characterization of locking degree and participation index. a**. Consistency between two measures of phase locking for individual neurons. The locking degree was calculated for each cell as the length of the mean vector over the distribution of oscillation phases ($[-\pi,\pi]$ rad) at which the calcium events occurred (bin size = 129 ms). The locking degree was consistent with the mutual information between the calcium event counts and the phase of the oscillation (bin size = 0.52 s). Scatter plots show the relation between the two measures, with each dot representing one neuron. Left: Data from the example session in Fig. 2a (n = 484 cells). Right: All neurons from all 15 oscillatory sessions are pooled (n = 6231 cells over 5 animals). Red dots indicate neurons that did not meet criteria for locking. The consistency between the two measures strengthens the conclusion that the vast majority of the neurons in MEC are locked to the oscillatory sequences. **b**. Distribution of preferred phases (the mean phase at which the calcium events occurred) in the population of locked neurons for all 15 oscillatory sessions. Black line indicates the preferred phases; red intervals indicate one standard deviation (calculated over the oscillation phases at which the calcium events of an individual cell occurred). Neurons are sorted according to their preferred phase in an ascending manner. Across the 15 oscillatory sessions, the smallest preferred phases ranged from −3.14 to −3.11 rad, and the largest preferred phase ranged from 3.08 to 3.14 rad, suggesting that the entire range of phases was covered. **c**. Phase preferences are distributed evenly across the MEC cell population. Left: The nearly-flat nature of the phase distribution is illustrated by comparing the entropy of the distribution of preferred phases in recorded (y axis) and shuffled data (x axis). $H_{ratio}$ is the entropy of the distribution of preferred phases (calculated as in (b)) estimated from the data and divided by the entropy of a flat distribution ($H_{ratio}$ = 1 if the distribution of preferred phases is perfectly flat, $H_{ratio}$ = 0 if all neurons have the same preferred phase). Each point in the scatterplot indicates one session (15 sessions). Horizontal error bars indicate one S.D. across shuffled realizations, and are centered around the mean across shuffled realizations. The black dashed line indicates identical values for recorded and shuffled data. Animal number if color-coded. Notice the discontinuity in the y axis between 0 and 0.85. $H_{ratio}$ is substantially larger for recorded data than for shuffled data. Right: Box plot of $H_{ratio}$ for recorded and shuffled data. For each session the 1000 shuffled realizations were averaged (n = 15 oscillatory sessions, $p = 6 \times 10^{-6}$, $Z = 4.52$, two-sided Wilcoxon rank-sum test). Red lines indicate median across sessions, the bottom and top lines in blue (bounds of box) indicate lower and upper quartiles, respectively. The length of the whiskers indicates 1.5 times the interquartile range. Red crosses show outliers that lie more than 1.5 times outside the interquartile range. **d**. Left: Box plot comparing locking degree for cells with an oscillatory frequency that was similar (relative frequency ~ 1) or different (relative frequency ≠ 1) from the sequence frequency in the example session in Fig. 2a (n = 48 cells in each group from a total of 484 cells in the recorded session, $p = 3.4 \times 10^{-11}$, $Z = 6.63$, two-sided Wilcoxon rank-sum test). Right: As left panel but for the locking degree across all 15 oscillatory sessions, including the example in the left panel

(n = 15 sessions over 5 animals, $p = 2.8 \times 10^{-5}$, $Z = 4.19$, two-sided Wilcoxon rank-sum test). Ten per cent of the total number of cells was used to define each of the groups with similar (relative frequency ~ 1) and different (relative frequency ≠ 1) oscillatory frequency as compared to the sequence frequency. Relative frequency was calculated for each cell as the oscillatory frequency of the cell's calcium activity divided by the sequence frequency in the session. Box plot symbols as in (c). Note that cells with relative frequency similar to 1 are more locked to the phase of the oscillation. For all percentages considered to define similar and different groups (5, 10, 20, 30, 40, and 50%) the p-values were significant. **e**. Histogram showing the distribution of single-cell oscillatory frequency divided by the sequence frequency of the session (n = 6231 cells pooled across 15 oscillatory sessions). A value of 1.0 indicates that single-cell and sequence frequency coincide. The left and right dashed lines indicate 25th (0.52) and 75th (1.08) percentiles respectively. Note that for approximately half of the data the oscillatory frequency is very similar at single-cell and population level. **f**. The oscillatory sequences remain visible after excluding increasing fractions of neurons and keeping only those with the lowest locking degree. Each row shows a PCA-sorted raster plot (left, rasterplot conventions as in Fig. 2b) and the corresponding joint distributions of the time lag $\tau$ that maximizes the correlation between the calcium activity of neuron pairs and their distance $d$ in the PCA sorting (right, symbols as in Extended Data Fig. 5b). The fraction of included neurons is indicated on top of the raster plot. For building the raster plots, neurons were sorted according to their locking degree value and neurons with the highest locking degrees were removed. **g**. Examples of different participation degrees in 3 example neurons from the session in Fig. 2a. Top: PCA sorted raster plot of the calcium matrix shown in Fig. 2a. Calcium events from the neuron with high participation index (PI, 0.72) are highlighted in light blue, from the neuron with intermediate PI (0.56) in purple, and from the neuron with low PI (0.36) in orange. Bottom three panels: Z-scored fluorescence calcium signals as a function of time from the above neurons with high (top), intermediate (middle), and low (bottom) PIs. Colored arrows represent the time points at which the oscillatory sequences are at the neuron's preferred phase. Notice how the neuron with high PI tends to exhibit a peak in the calcium signal for most of the sequences. Neurons with intermediate and low PIs demonstrate the same but to a lesser extent, with the calcium signal not peaking in each sequence. **h**. Similar to (d), but for the participation index. Box plot symbols as in (c). Left: Data from the example session shown in Fig. 2a (n = 48 cells in each group, $p = 0.51$, $Z = 0.66$, two-sided Wilcoxon rank-sum test). Right: As left panel but for data pooled across 15 oscillatory sessions. The mean participation index was calculated for each group ("relative frequency ~ 1" and "relative frequency ≠ 1") and each session separately and the data was then pooled across sessions (n = 15 sessions, $p = 0.56$, $Z = 0.58$, two-sided Wilcoxon rank-sum test). For all percentages considered to define the similar and different groups (5, 10, 20, 30, 40, and 50%) the p-values were non-significant. *** $p < 0.001$, n.s. $p > 0.05$.

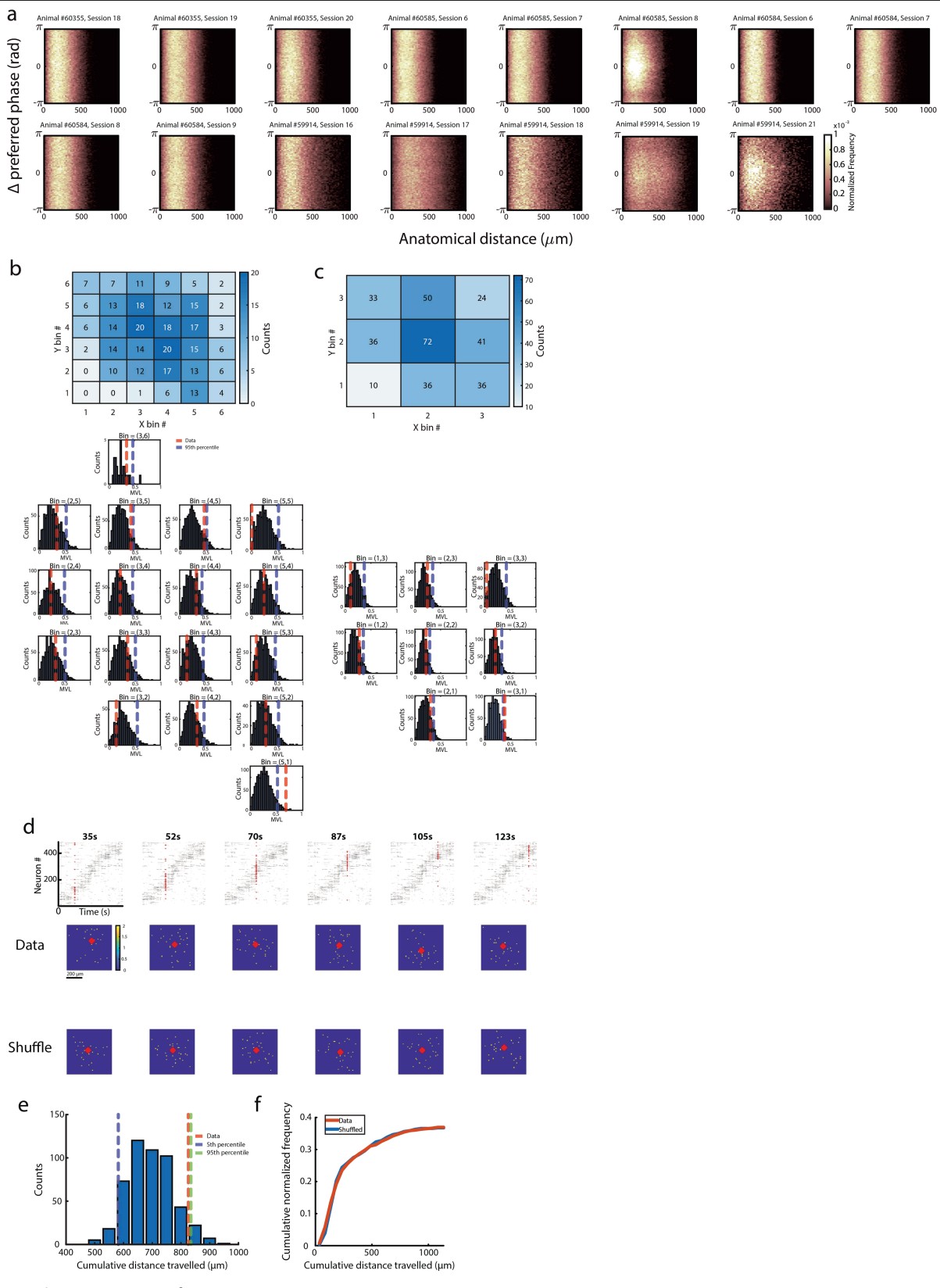

**Extended Data Fig. 8** | See next page for caption.

**Extended Data Fig. 8 | The oscillatory sequences are not topographically organized. a**. 2D histograms of differences in preferred phase between pairs of neurons and their anatomical distance in the FoV for all 15 oscillatory sessions (5 animals, of which 4 had oscillatory sequences). Preferred phases were calculated as the mean oscillation phase at which the calcium events occurred (after pooling all sequences in a session and not on each sequence separately; see Fig. 3f, g for one individual sequence). Each histogram was built using $N*(N-1)/2$ samples, where N is the total number of recorded cells in the session. One count is a cell pair, the color bar indicates normalized frequency. The absolute Pearson correlation values were calculated for each session, and ranged from $8.5 \times 10^{-5}$ to 0.015. Only session 6 from animal #60585 (first row, fourth column) had a correlation value above the 95th percentile of a shuffled distribution built by shuffling the preferred phases in the FoV (1/15, probability = 0.37, binomial probability distribution; not statistically significant at a chance level of 5%). For the participation index (not shown) the correlation values were also very small and ranged from $9.3 \times 10^{-4}$ to 0.040. Out of 15 oscillatory sessions, 2 sessions (sessions 6 and 8 from animal #60584, correlation = 0.033 and 0.040 respectively) were classified as significant (2/15 sessions, probability = 0.13, binomial distribution, not statistically significant at a chance level of 5%). **b**. Analysis of similarity of preferred phases within spatial bins for one single example sequence (number 19) of the session presented in Fig. 2a. Similarity was calculated as the mean vector length (MVL) of the distribution of preferred phases in the spatial bin. In the presence of travelling waves, large MVL values in every bin are expected. Top: The FoV is binned into 6×6 bins, each of size 100 um x 100 um. The heat map shows the number of cells located within each spatial bin. Counts are color coded. Bottom: Each panel indicates a spatial bin in the FoV, and shows the shuffled distribution of MVL values obtained after shuffling the preferred phases in the FoV (histogram), the 95th percentile of the shuffled distribution (dotted blue line), and the MVL calculated on experimental data (dotted red line). To have good statistics only spatial bins that had more than 10 neurons were included in the quantifications. The plots that are missing are for bins with 10 or fewer cells, as indicated in the heat map. When using 100 μm x 100 μm bins, only 17 bins had more than 10 cells. From the 17 bins, one was classified as having similar phases (1/17, probability = 0.37, binomial distribution, not statistically significant at a chance level of 5%); when using 200 μm x 200 μm, only one bin out of eight with more than 10 cells was classified as having cells with similar phases (1/8, probability = 0.28, binomial distribution, not statistically significant at a chance level of 5%). When all sequences across all calcium imaging sessions are considered (n = 421, 15 oscillatory sessions over 5 animals), the MVL values calculated on experimental data ranged from 0.0082 to 0.98 (the 95th percentile MVL value was 0.3399, i.e. small), and were larger than the cutoff for significance in 121 out of 2448 spatial bins (121/2448, smaller than expected at a chance level of 0.05: 122/2448). This analysis was focused on the degree of similarity between preferred phases in spatial bins. In order to avoid small cell sample effects, we performed a second analysis based on the difference in preferred phases for all pairs of cells that were located within small

neighborhoods in the FoV (Methods). We expected that in the presence of travelling waves the mean and median of the distributions of differences in preferred phases of cell pairs within small neighborhoods would be smaller than expected by chance. For neighborhoods of 50 μm, only 16 out of 421 sequences had a mean below the cutoff for significance (16/421, smaller than expected at a chance level of 0.05: 21/421), and 16 out of 421 sequences a median below the cutoff for significance (16/421, smaller than expected at a chance level of 0.05: 21/421). For neighborhoods of 100 μm, 16 and 19 sequences (out of 421) were below the cutoff for the mean and median, respectively (16/421 and 19/451, both below a chance level of 0.05: 21/421). For neighborhoods of 200 μm, 25 sequences were slightly above the cutoff for the mean and 18 were below the cutoff for the median (chance level of 0.05: 21/421). **c**. Similar to (b), but with spatial bins of 200 μm x 200 μm. For all sequences, the MVL values calculated on experimental data ranged from 0.0037 to 0.975 (the median of MVL values was 0.3105, i.e. small), and were larger than the cutoff for significance in 115 spatial bins out of 2392 (115/2392, smaller than expected at a chance level of 0.05: 120/2392). The lack of similarity in preferred phases within spatial bins is inconsistent with a coherent oscillation in that spatial bin, and therefore inconsistent with the presence of travelling waves. **d**. Top: Rasterplots showing one example sequence from the session in Fig. 2a (sequence #19). Y axis: Neuron #. X axis: Time (s). Each panel shows the same sequence, and a total of 150 s (the length of the illustrated sequence). Neurons that were active in one particular time bin are indicated in red. The visualized time bin is indicated at the top of each panel (bin size = 1 s). Middle: Anatomical distribution of the population activity in each of the time bins in the top panel (bin size is now 5 s). The FoV (600 μm x 600 μm) was divided into 50×50 square spatial bins. The total number of calcium events across cells in one spatial bin is color coded (yellow indicates high activity, purple no activity). The big red dots indicate the position of the center-of-mass (COM) of the population activity in that time bin. Bottom: Similar to the middle panel, but for one shuffle realization in which the position of the cells was randomly shuffled within the FoV. **e**. Quantification of the flow of the COM for the example sequence shown in (d). Cumulative distance travelled, quantified as the sum of the distances travelled by the COM between consecutive time points (bin size = 5 s), in experimental data (dotted red line), in shuffled data (blue histogram, built by shuffling the positions of the cells in the FoV 500 times), and the 5th and 95th percentile of the shuffled distribution (dotted blue and green lines, respectively). The data shows no significant difference from cumulative distances expected by chance. **f**. Quantification of the flow of the COM for all sequences. Cumulative normalized frequency of the cumulative distance travelled in experimental data (n = 421 sequences, orange) and the median of the shuffled distributions (n = 421 sequences, blue). Out of 421 sequences, 21 were below the cutoff for significance (21/421, at the chance level of 0.05: 21/421, bin size = 5 s). The results are similar when changing the temporal bin size used for the quantifications (23/421 for bin size = 1 s, 23/421 for bin size = 2 s, chance level of 0.05: 21/421).

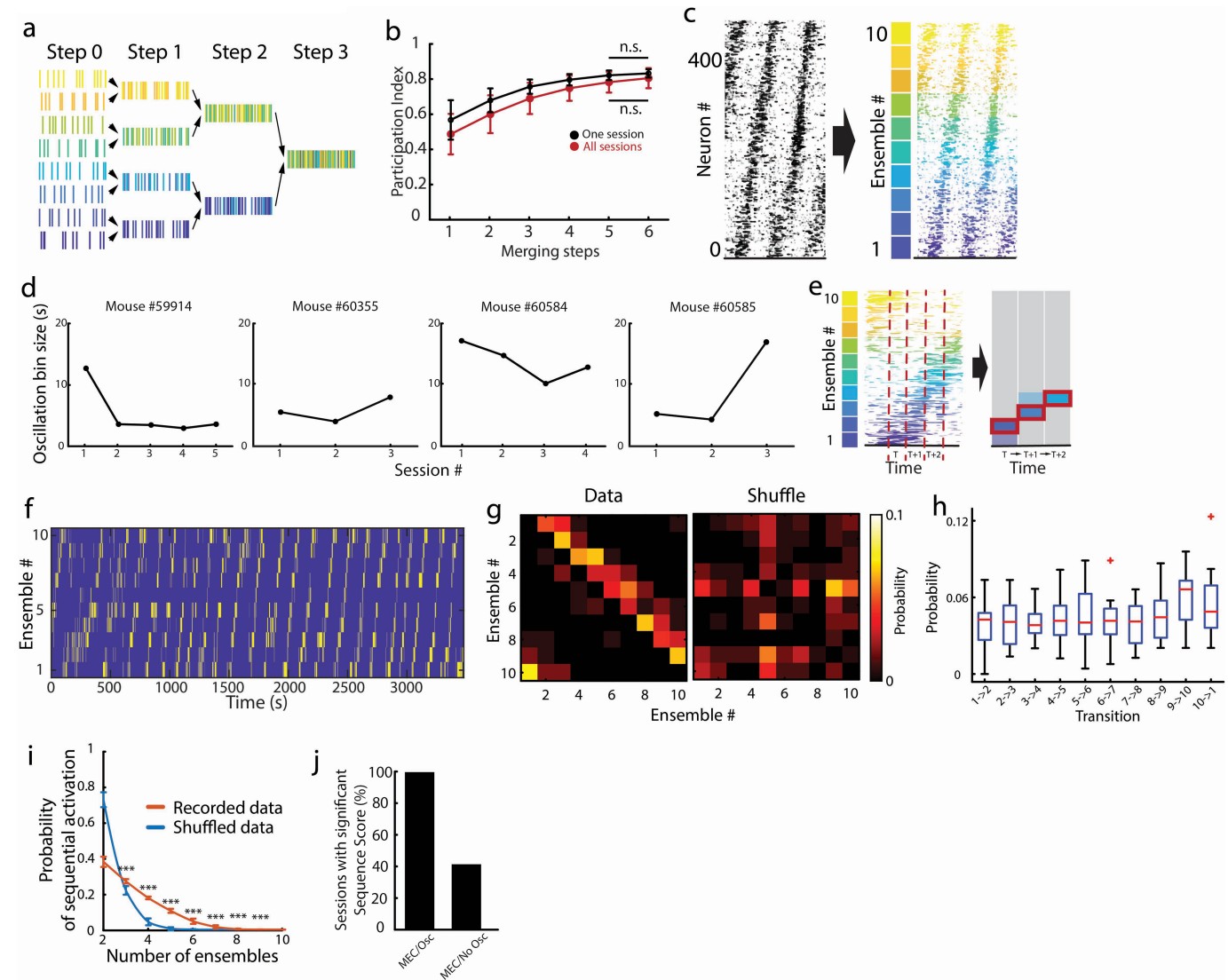

**Extended Data Fig. 9 |** See next page for caption.

**Extended Data Fig. 9 | Analysis of ensemble activation during the oscillatory sequences. a**. Schematic of calcium activity merging steps (data are not included in this panel). We began by sorting the neurons according to the PCA method. Next, in successive iterations, or merging steps, we added up the calcium activity of pairs of consecutive neurons (merging step = 1) or consecutive ensembles (merging step > 1). **b**. Participation index (PI) as a function of merging step (mean ± S.D.). Black trace, example session in Fig. 2a; red trace, all 15 oscillatory sessions. The more neurons per ensemble, the higher the participation index of the ensemble. Note that the participation index plateaus after 5 merging steps, which corresponds to approximately 10 ensembles in most of the sessions (two-sided Wilcoxon rank-sum test to compare the participation indexes in merging steps 5 and 6; Black trace: $n = 30$ PIs in merging step 5, $n = 15$ PIs in merging step 6, $p = 0.23$, $Z = 1.20$; Red trace: $n = 15$ PIs in merging step 5 and 6, PIs of each merging step were averaged for each session separately, $p = 0.14$, $Z = 1.49$). **c**. Schematic of the process for splitting neurons into ensembles of co-active cells. Neurons sorted according to the PCA method are allocated to 10 equally sized ensembles (color-coded). Note that the participation index plateaued after 5 merging iterations, consisting of approximately 10 ensembles depending on the session (panel b). **d**. To quantify the temporal progression of the population activity at the time scale at which the oscillatory sequences evolved, we calculated, for each session, an oscillation bin size. This bin size is proportional to the inverse of the peak frequency of the PSD calculated on the phase of the oscillation, and hence captures the time scale at which the sequences progress. The oscillation bin size is shown for each of the 15 oscillatory sessions (4 out of 5 animals, those that had oscillatory sequences). **e**. Schematic of the method used for quantifying temporal dynamics of ensemble activity. For each session and each ensemble we calculated the mean ensemble activity at each time bin (oscillation bin size). Only the ensemble with the highest activity within each time bin (red rectangle) was considered. The number of transitions between ensembles in adjacent time bins divided by the total number of transitions was used to calculate the transition matrices in (g). **f**. The ensemble with the highest activity in each time bin, indicated in yellow and calculated as in (e), plotted as a function of time for the example session in Fig. 2a. All other ensembles are indicated in purple. Notice that the transformation in (e) preserves the oscillatory sequences. **g**. Left: Matrix of transition probabilities between pairs of ensembles at consecutive time points. Rows indicate the ensemble at time point $t$, columns indicate the ensemble at time point $t + 1$. Data are from the example session in Fig. 2a (bin size = 15.12 s). Right: Same as left panel but for one shuffle realization. Transition probabilities are color coded. In the left diagram, note the higher probability of transitions between consecutive ensembles (increased probabilities near the diagonal), the directionality of transitions (increased probabilities above diagonal) and the periodic boundary conditions in ensemble activation (presence of transitions from ensemble 10 to ensemble 1). **h**. Box plot showing transition probabilities between consecutive ensembles for all 15 oscillatory sessions. The probabilities remain approximately constant across transitions between ensemble pairs ($n = 15$ oscillatory sessions per transition, $p = 0.56$, $\chi^2 = 7.77$, Friedman test), and there were no significant differences between pairs of transitions (two-sided Wilcoxon rank-sum test with Bonferroni correction, $p > 0.05$ for all transitions). Transitions from ensemble 10 to ensemble 1 were equally frequent as transitions between consecutive ensembles, as expected from the periodic nature of the sequences. Red lines indicate median across sessions, the bottom and top lines in blue (bounds of box) indicate lower and upper quartiles, respectively. The length of the whiskers indicates 1.5 times the interquartile range. Red crosses show outliers that lie more than 1.5 times outside the interquartile range. **i**. Probability of sequential ensemble activation as a function of the number of ensembles that are sequentially activated (mean ± S.D.; For 3–9 ensembles: $n = 15$ oscillatory sessions over 5 animals, 7500 shuffle realizations; $p = 5.4 \times 10^{-11}$, $1.0 \times 10^{-11}$, $5.9 \times 10^{-13}$, $4.5 \times 10^{-49}$, 0, 0, $9.0 \times 10^{-220}$ respectively, range of $Z$ values: 6.45 to 59.18, one-tailed Wilcoxon rank-sum test). Orange, recorded data; blue, shuffled data. For each session, the probability of sequential ensemble activation was calculated over 500 shuffled realizations, and shuffled realizations were pooled across sessions. The recorded data contained significantly longer sequences than the shuffled control. Probability of sequential activation of ≥ 3 ensembles in recorded data = 0.62; probability of sequential activation of ≥ 3 ensembles in shuffled data = 0.27. **j**. Percentage of sessions with significant sequence score in sessions classified as oscillatory vs non-oscillatory. In MEC sessions with oscillatory sequences, 100% (15 of 15) of the sessions showed significant sequence scores, while in MEC sessions without oscillations, 41% (5 of 12) of the sessions demonstrated significant sequence scores. For corresponding raster plots, see Extended Data Fig. 5a. \*\*\*$p < 0.01$, ns $p > 0.05$.

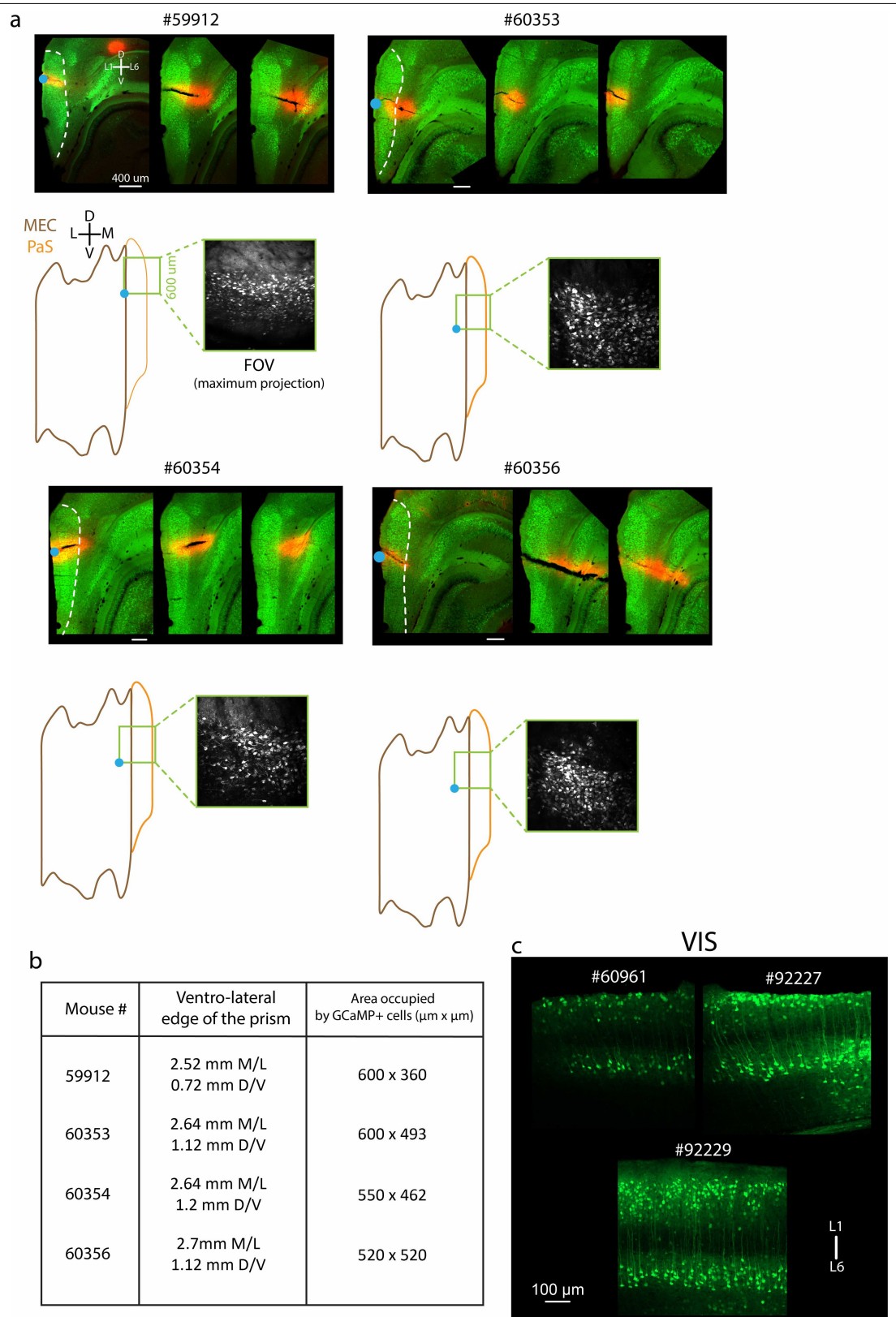

**Extended Data Fig. 10** | See next page for caption.

**Extended Data Fig. 10 | Histology showing imaging location in animals with FoVs in parasubiculum and visual cortex. a**. Histological determination of prism location in mice that were implanted more medially, touching parasubiculum more than MEC. Top: Maximum intensity projection of 50 μm thick sagittal brain sections (sections acquired with an LSM 880, 20x). Three consecutive sections from the same mouse are shown, from the most lateral (left) to the most medial (right). Green is GCaMP6m signal, while red is DiI signal (used to demarcate ventrolateral corner of the prism, as in Extended Data Fig. 1). Scale bar is 400 μm. The white stippled line encapsulates the superficial layers of the parasubiculum (PaS). Dorsal PaS on top, layer 1 on the left. Bottom: Estimated location of the field of view (FoV) on a flat map encompassing MEC (brown outline) and PaS (yellow outline). The blue dot marks the location of the pin used to demarcate the most lateral-ventral border of the prism, while the green square inset shows the microscope FoV. Inset images show maximum intensity projections of the FoV. Dorsoventral (DV), and mediolateral (ML) axes are indicated. **b**. Location of the ventro-lateral edge of the prism in stereotactic coordinates, and area of the FoV occupied by cells expressing GCaMP6m for each PaS-imaged animal. **c**. Histological determination of imaging location in the visual cortex (VIS) of three mice that underwent calcium imaging. Green is GCaMP6m signal. Images are taken from coronal slices, and zoomed in on visual cortex (Scale bar is 100 μm; L1 at the top, L6 at the bottom). Dorsal pole of the brain is on top. Maximum intensity projection, LSM 880, 20x.

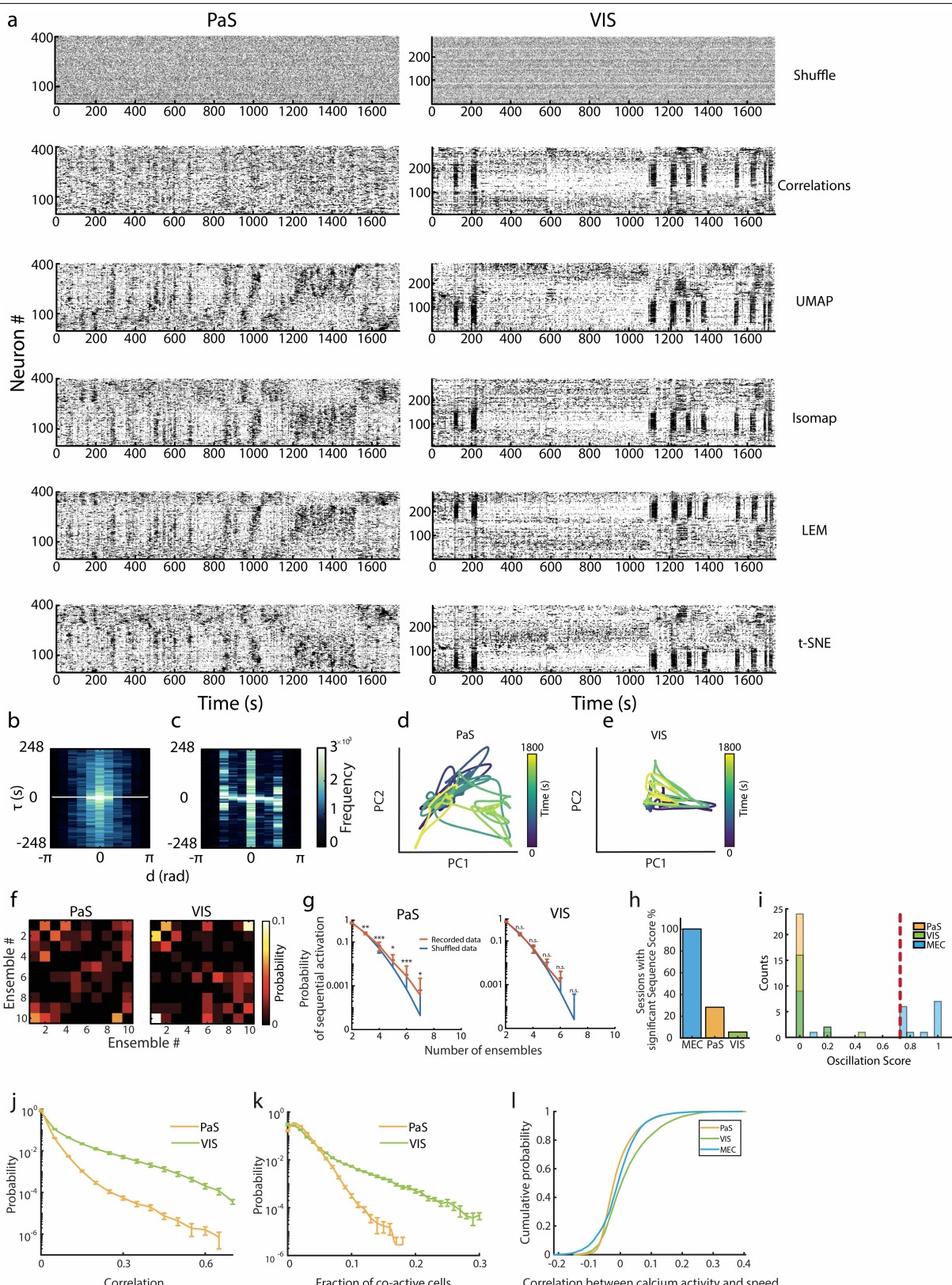

**Extended Data Fig. 11** | See next page for caption.

**Extended Data Fig. 11 | Lack of oscillatory sequences in parasubiculum and visual cortex. a**. Alternative sorting methods, as in Extended Data Fig. 4d, but applied to sessions recorded in the PaS (left) or VIS (right). The PCA sorting method applied to temporally shuffled data did not unveil oscillatory sequences (first row). No oscillatory sequences were recovered when neurons were sorted according to their correlation values (second row), or according to different dimensionality reduction techniques (UMAP, Isomap, LEM, t-SNE). Each row of each raster plot shows the calcium activity of a single neuron, with activity plotted as a function of time, as in Fig 2a. Every dot indicates that one neuron was active at one specific time bin (bin size = 129 ms). Sequence scores and oscillation scores are presented in Fig. 5e,f. **b**,**c**. Joint distributions of time lag τ that maximizes the cross-correlation between any given pair of neurons and their distance $d$ in the PCA sorting (as in Extended Data Fig. 5b), applied to the recordings in Fig. 5e (PaS) and Fig. 5f (VIS). Normalized frequency is color-coded. Notice lack of linear relationship between $d$ and τ, in contrast to Extended Data Fig. 5b. **d**,**e**. Projection of the neural activity onto the low-dimensional embedding defined by the first two principal components obtained from applying PCA to the matrix of calcium activity of the PaS session (d) and the VIS session (e) shown in Fig. 5e, f. Bin size = 8.5 s. Note lack of obvious ring topology. Time is color-coded. **f**. Transition probabilities between ensembles across consecutive time bins (bin size - 8.5 s, Methods) for the PaS example session in Fig. 5e (left) and the VIS example session in Fig. 5f (right). **g**. Probability of sequential ensemble activation as a function of the number of ensembles that are sequentially activated in PaS (left) and VIS (right) (mean ± S.D.). Orange, recorded data (25 PaS sessions; 19 VIS sessions); blue, shuffled data. For each session, the probability of sequential ensemble activation was calculated over 500 shuffled realizations, and shuffled realizations were pooled across sessions for each brain area separately. Probability is shown on a log-scale. In PaS the probability of long sequences was significantly larger in experimental data than in shuffled data ($n$ = 25 PaS sessions, 12500 shuffled realizations; For 2 ensembles: $p$ = 0.998, $Z$ = −2.90; For 3–7 ensembles: range of $p$ values: 5.7 × 10$^{-4}$ to 0.036, range of $Z$ values: 1.80 to 3.25, one-tailed Wilcoxon

rank-sum test). This was not the case in VIS ($n$ = 19 VIS sessions, 9500 shuffled realizations; For 2 ensembles: $p$ = 0.106, $Z$ = 1.25; For 3–6 ensembles: range of $p$ values: 0.087 to 0.999, range of $Z$ values: −3.34 to 1.36, one-tailed Wilcoxon rank-sum test). **h**. Percentage of sessions with significant sequence score (MEC oscillatory sessions: 15 of 15, PaS: 7 of 25; VIS: 1 of 19). The sequence score quantifies the probability of observing sequential activation of 3 or more ensembles. **i**. Distribution of oscillation scores for the entire calcium imaging data set, as in Extended Data Fig. 5d (19 VIS sessions over 3 animals, 25 PaS sessions over 4 animals, 27 MEC sessions of which 15 were classified as oscillatory, over 5 animals). Dashed line indicates threshold for classifying sessions as oscillatory with reference to the MEC data. Note that the bars for different brain regions sometimes overlap, and that bars are colored with transparency for visualization purposes (e.g. for sessions in PaS with oscillation score 0, the count is 24). **j**. Normalized distribution of the Pearson correlation values (absolute value) between the activity of cell pairs in VIS (green) and in PaS (yellow). Each dot indicates the mean across sessions (25 PaS sessions, 19 VIS sessions; all sessions in the data set were used, not only those with behavioural tracking synchronized to imaging), error bars indicate S.E.M. Probability is shown on a log-scale. **k**. Same as (j) but for the distribution of values of coactivity for all sessions recorded in PaS (yellow) and VIS (green). Coactivity was estimated for each session separately as the fraction of the recorded cells that was simultaneously active in 129 ms bins. Probability is shown on a log-scale. **l**. Cumulative probability of correlation values calculated between the calcium activity of one cell and the speed of the animal in that session for MEC ($n$ = 4595 cells from 10 sessions, 3 animals), PaS ($n$ = 6851 cells from 18 sessions, 3 animals), VIS ($n$ = 6037 cells from 19 sessions, 3 animals). Only sessions for which the imaging data was synchronized to behavioural data were used (VIS-PaS: $p$ = 3.15 × 10$^{-169}$, $Z$ = 27.7; VIS-MEC: $p$ = 1.05 × 10$^{-85}$, $Z$ = 19.6, MEC-PaS: $p$ = 5.16 × 10$^{-12}$, $Z$ = 6.80, one tailed Wilcoxon rank-sum test). Calcium activity was more correlated with the speed of the animal in visual cortex than in MEC and PaS.

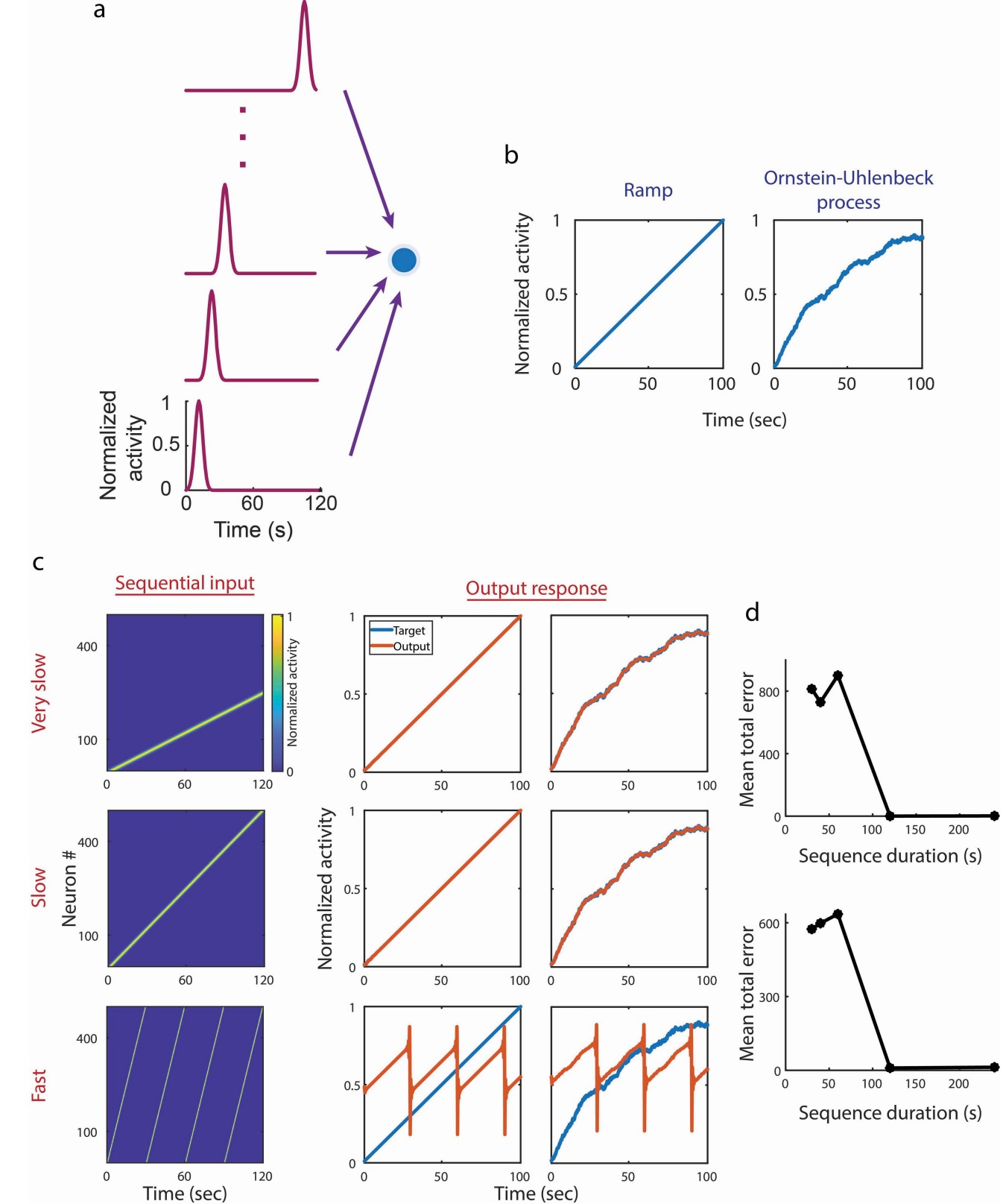

**Extended Data Fig. 12 |** See next page for caption.

**Extended Data Fig. 12 | Ultraslow oscillatory sequences might serve as template for generating new activity patterns. a**. Schematic of the model. 500 input units (depicted in purple, on the left) are connected to an output unit (blue dot, on the right). Each arrow represents a connection, and there are 500 connections in total. The activity of each input unit is represented by a periodic Gaussian bump. The means of the Gaussians are temporally displaced such that, all together, input units fire in a sequence (the 'template'). **b**. The weights between the input units and the output unit were trained such that the output unit reproduced a target activity. Two targets were considered: a ramp of activity, which is deterministic (left), and an Ornstein-Uhlenbeck process, which is stochastic (right). Both targets had a characteristic time scale of 100 s. **c**. Input (left) and output (right) activity for three different sequence lengths: sequences are very slow (top row), slow (middle) or fast (bottom) as compared to the targets. Left: Heat map of the activity of the input units as a function of time, in seconds. Blue indicates no activity, yellow indicates maximal activity. Top: Sequences are 400 s long. Middle: Sequences are 120 s long. Bottom: Sequences are 30 s long. Right: Output response corresponding to the three sequences regimes: very slow, slow and fast sequences. Target response is shown in blue, obtained response after training the networks using the sequences as input is shown in orange. Note that when the sequences have a time scale that is similar (middle) or slower (top) than the targets, the output unit can reproduce the desired target. **d**. Mean total error, calculated as the difference between the target and the obtained response after training, as a function of the input sequence length. Top: Target is the ramp of activity. Bottom: Target is the Ornstein-Uhlenbeck process.

|  | Soledad Gonzalo Cogno |
| --- | --- |
|  | May-Britt Moser |

# Reporting Summary

## Statistics

For all statistical analyses, confirm that the following items are present in the figure legend, table legend, main text, or Methods section.

| n/a | Confirmed | |
| --- | --- | --- |
| ☐ | ☒ | The exact sample size (*n*) for each experimental group/condition, given as a discrete number and unit of measurement |
| ☐ | ☒ | A statement on whether measurements were taken from distinct samples or whether the same sample was measured repeatedly |
| ☐ | ☒ | The statistical test(s) used AND whether they are one- or two-sided<br>*Only common tests should be described solely by name; describe more complex techniques in the Methods section.* |
| ☐ | ☒ | A description of all covariates tested |
| ☐ | ☒ | A description of any assumptions or corrections, such as tests of normality and adjustment for multiple comparisons |
| ☐ | ☒ | A full description of the statistical parameters including central tendency (e.g. means) or other basic estimates (e.g. regression coefficient) AND variation (e.g. standard deviation) or associated estimates of uncertainty (e.g. confidence intervals) |
| ☐ | ☒ | For null hypothesis testing, the test statistic (e.g. *F*, *t*, *r*) with confidence intervals, effect sizes, degrees of freedom and *P* value noted<br>*Give P values as exact values whenever suitable.* |
| ☒ | ☐ | For Bayesian analysis, information on the choice of priors and Markov chain Monte Carlo settings |
| ☒ | ☐ | For hierarchical and complex designs, identification of the appropriate level for tests and full reporting of outcomes |
| ☐ | ☒ | Estimates of effect sizes (e.g. Cohen's *d*, Pearson's *r*), indicating how they were calculated |

*Our web collection on statistics for biologists contains articles on many of the points above.*

## Software and code

Policy information about availability of computer code

| Data collection | MESc, versions 3.3 and 3.5, Femtonics, Hungary (2p Imaging); ZEN, Version 3 (blue edition), Carl ZEISS, Germany (confocal microscopy), SpikeGLX (https://billkarsh.github.io/SpikeGLX) version 20201103, Imec phase30 version 3.31, Motive (OptiTrack) version 2.2.0, MATLAB (MathWorks) version r2020a |
| --- | --- |
| Data analysis | Commercial software: MATLAB (MathWorks) versions r2019b, r2020a and r2021b, Python version 3.7, Imaris versions 9.8.0 and 9.8.2, Bitplane (processing of histological images).<br><br>Open-source Python codes:<br>- Suite2P: https://github.com/MouseLand/suite2p<br>- Kilosort (version 2.5): https://github.com/MouseLand/Kilosort<br><br>Open-source Matlab codes:<br>- UMAP version 1.3.4: https://se.mathworks.com/matlabcentral/fileexchange/71902-uniform-manifold-approximation-and-projection-umap<br>- Toolbox for Dimensionality Reduction, available in: https://lvdmaaten.github.io/drtoolbox/<br>- Circular Statistics Toolbox version 1.21.0.0: https://se.mathworks.com/matlabcentral/fileexchange/10676-circular-statistics-toolbox-directional-statistics<br>Paper: https://www.jstatsoft.org/article/view/v031i10 |

For manuscripts utilizing custom algorithms or software that are central to the research but not yet described in published literature, software must be made available to editors and reviewers. We strongly encourage code deposition in a community repository (e.g. GitHub). See the Nature Portfolio guidelines for submitting code & software for further information.

## Data

Policy information about availability of data

All manuscripts must include a data availability statement. This statement should provide the following information, where applicable:
- Accession codes, unique identifiers, or web links for publicly available datasets
- A description of any restrictions on data availability
- For clinical datasets or third party data, please ensure that the statement adheres to our policy

The datasets generated during the current study will be available after publication, on EBRAINS.

# Field-specific reporting

Please select the one below that is the best fit for your research. If you are not sure, read the appropriate sections before making your selection.

☒ Life sciences     ☐ Behavioural & social sciences     ☐ Ecological, evolutionary & environmental sciences

For a reference copy of the document with all sections, see nature.com/documents/nr-reporting-summary-flat.pdf

# Life sciences study design

All studies must disclose on these points even when the disclosure is negative.

| | |
|---|---|
| Sample size | Samples included all available cells. |
| Data exclusions | Cells with very low signal-to-noise ratio (below 4) were excluded because of their unsuitability for the performed analyses. |
| Replication | For all animals included in the study, in the results text we indicate for each animal either the experimental sessions or the fraction of experimental sessions in which the effect was found. From 5 MEC calcium imaging animals, ultraslow oscillatory sequences were observed in 4 animals. From 2 MEC Neuropixels animals, ultraslow oscillatory sequences were observed in the 2 animals. |
| Randomization | The study did not involve any experimental subject groups; therefore, random allocation did not apply and was not performed. |
| Blinding | The study did not involve any experimental subject groups; therefore, experimenter blinding did not apply and was not performed. |

# Reporting for specific materials, systems and methods

We require information from authors about some types of materials, experimental systems and methods used in many studies. Here, indicate whether each material, system or method listed is relevant to your study. If you are not sure if a list item applies to your research, read the appropriate section before selecting a response.

### Materials & experimental systems

| n/a | Involved in the study |
|---|---|
| ☒ | ☐ Antibodies |
| ☒ | ☐ Eukaryotic cell lines |
| ☒ | ☐ Palaeontology and archaeology |
| ☐ | ☒ Animals and other organisms |
| ☒ | ☐ Human research participants |
| ☒ | ☐ Clinical data |
| ☒ | ☐ Dual use research of concern |

### Methods

| n/a | Involved in the study |
|---|---|
| ☒ | ☐ ChIP-seq |
| ☒ | ☐ Flow cytometry |
| ☒ | ☐ MRI-based neuroimaging |

## Animals and other organisms

Policy information about studies involving animals; ARRIVE guidelines recommended for reporting animal research

| | |
|---|---|
| Laboratory animals | C57/Bl6 mice, male. Age: 3-5 months for one group (22-36g during the recordings), 0-2 months for another group (15-20g during the recordings). |
| Wild animals | None |
| Field-collected samples | None |
| Ethics oversight | Protocols approves by Norwegian Animal Welfare Act and the European Convention for the Protection of Vertebrate Animals used |

Ethics oversight | for Experimental and Other Scientific Purposes, Permit numbers 18011 and 29893.

Note that full information on the approval of the study protocol must also be provided in the manuscript.

