## [Peer Review File · Nature]

Manuscript Title: Minute-scale oscillatory sequences in medial entorhinal cortex

Reviewer Comments & Author Rebuttals

Reviewer Reports on the Initial Version:

Referees' comments:

Referee #1 (Remarks to the Author):

This intriguing and really interesting article by Gonzalo Cogno et al describes calcium signals recorded with 2Ph imaging in head fixed mice walking on a wheel in the dark, about a month after injection of AAV1-Syn-GCaMP6m in the medial entorhinal cortex. The paper's main result is that such recordings reveal very slow and non-stochastic dynamics in the neurons of MEC with two principal characteristics: population activity is periodic with long periods (up to 100s), thus within the range of what is often described as infra-slow oscillations; activity is sequential, such that distributed groups of co-active neurons succeed each other in reliable order, closing a loop that typically starts over again after one full period, and typically many times over in each recording session. This activity is not a wave in a spatial sense, and it involves nearly all the imaged MEC neurons, suggesting to the authors that it must involve all known functional cell types in MEC indiscriminately. Plotted in a low-d phase space using a variety of cross-validating dimensionality reduction techniques, these patterns trace closed loops. Much of the manuscript is devoted to technical issues related to the processing of the primary imaging data and to the analysis and interpretation of the so-processed datasets. The authors carry out similar recordings and analyses from the visual cortex and para-subiculum, and report an absence of infra-slow phenomena as seen in MEC.

This paper is technically very elaborate and very interesting in that it describes an ordered global phenomenon, apparently specific to the MEC, that has (to my knowledge) never been reported before. Infra-slow oscillations have been described before but never, I believe, as masking such ordered yet non-propagating population dynamics. The paper is intriguing (intriguing in a positive sense of the word) for a number of functional and mechanistic reasons. I will raise some of them below.

1. Issues related to technical aspects of the experiments.

I. A first issue concerns the imaging. This technique is (I think) new in this lab and one wonders whether a round of calibration/sanity checks using familiar electrophysiological techniques was carried out. I am sure that the first thing the authors must have wondered is whether their observations were artifactual, and linked to some technical imaging/data processing issue. Has this been tested and how (see below)?

II. Were the parameters of the timeseries analyses (deconvolution kernel, thresholds, sampling rates etc...) explored or based on prior knowledge and factual data? In essence, what do the binary "calcium event" signals mean: subthreshold potentials, spiking, bursting, all of the above?

III. In the same vein as II, could astrocytes or other glia be responsible for or involved in these phenomena?

IV. Following this, were tetrode or N-pixel recordings carried out in the same conditions (head-fixed mouse on a wheel, darkness, etc...), to check some aspects at least of the results, such as the existence of this phenomenon in such conditions, with its long and variable periodicity? (This might also allow one to identify nested theta and gamma that cannot be extracted from the imaging data due to the low sampling there.)

V. Is this a result of head fixation? An e-phys signature would enable one to test this in freely behaving animals.

VI. Does this phenomenon disappear in lit conditions, in sleep, or with closed loop visual feedback?

VII. In the 2-step procedure leading to the ranking of neurons, I seem to understand that each entire dataset was used to compute pairwise correlations. What happens if only a fraction of the time series (eg first third) is used for the ranking and then applied to the entire data? This test would be to check that the apparent sequence reliability is not an artifact of the ranking procedure.

VIII. I did not understand the classification of running vs immobility with a threshold for immobility at non-zero speed ($<2\text{cm/s}$). Why this choice and how do the results look when true immobility is used to partition the data? I also did not understand the median of velocities at 0cm/s .

2. Issues related to the periodicity and sequences

I. One of the most interesting and astonishing aspects of this study is the slow periodicity of this activity but also the variations of its period—sometimes within a recording epoch, more often across recording episodes (eg EDF 4a,b) and finally, across animals. In one case (mouse 59914, episodes 1, 2, 5), it even seems as if activity runs in reverse (I assume episodes 4 and 5 plot the same neurons). In others, as with mouse 59911, or mouse 60355 session 16, activity is classified as non-periodic but it is clearly far from random and highly correlated across the population (presumably the rank ordering of the neurons rests on these pairwise correlations?). All this richness of ordered (simultaneous or sequential, with tunable speed) but non-spatial-wave population activity suggests something highly non-trivial (and thus very interesting) if it is not linked to some external (technical) cause, or a sensory signal. I could not help but wonder if the search for a correlation with animal motion (apparently non-existent, or at least not clear) is sufficient to eliminate all external or behavioral correlates.

II. The correlation times series seem to indicate occasional discontinuities (eg fig 2b, second and third sequences). Likewise, when one zooms out on the displays, one occasionally perceives other structures, such as a constricted zone around cell 200 in fig 2a, very slow periodicity (EDF4 mouse 59914, last row), etc. Are these real, or artifacts of the ranking are displayed (see 1-VI)?

III. One of the most astonishing aspects (to me) is the sequential but non-topological activation of these populations and the fact that they appear to form closed loops. That they form closed loops suggests that the authors' sample (over a small fraction of the MEC) of neurons is sufficient to describe the entire system's path. The putative ensembles that succeed each other to close a loop would therefore occupy the entire MEC, and be intermingled throughout. Is this a correct understanding?

IV. If so, what mechanisms could account for this spatially widespread coordination of widely distributed but more or less synchronized ensembles, plus a very slow propagation of activity across ensembles that live in the adjacent physical space?

V. The E and MB Moser lab nicely showed a while back that grid (spatial) periodicity varies in discrete steps along EC. Would matching discontinuities (here, in dynamics) have been detected in the present data, given the sampling constraints on these imaging experiments?

VI. An interesting feature of sequences might be their initiation from a period of inactivity. Can these initiating groups be identified and compared across successive sequences separated by inactivity?

Referee #2 (Remarks to the Author):

This work identified slow oscillatory activity in the medial entorhinal cortex (MEC) through calcium imaging of many neurons. The period of the oscillation was about 50-60 seconds and the majority of MEC neurons participated in this oscillation but different neurons fired at different oscillatory phases. As a result MEC neurons tend to fire in a sequential manner: their firing rate modulation followed a slow sequential order. This oscillatory activity was present more frequently during movement as compared to immobility periods. Finally, this kind of sequential network activity associated with oscillations was not seen in the visual cortex and parasubiculum. The finding is surprising and required simultaneous imaging of many neurons to demonstrate the periodic sequential activity. I liked the dimensionality reduction approach to identify oscillatory phases, especially since summing up the activity of all cells would not be able to identify this sort of oscillatory pattern. Overall the manuscript used very original data analysis approaches. I have two questions only:

1. It looks from the shown examples that a subgroup of neurons in the visual cortex engaged transiently in periodic activity as well, albeit it was not sequential but up-down state-like but at a much lower frequency than cortical slow wave oscillations in sleep. Could this be quantified in some way and acknowledge that slow oscillations of a different form can exist outside the MEC?
2. The group has been recently published work in the same journal in which they recorded from many MEC cells using neuropixel probes. The reviewer cannot help being curious whether similar oscillations could be detected when real action potentials are detected, not calcium signal and animals engage in more natural behavior.

Jozsef Csicsvari

Referee #3 (Remarks to the Author):

In this manuscript, Cogno et al. provide evidence for ultra-slow oscillations that create slow minute-scale sequences in medial entorhinal cortex (MEC), which entrain large neuronal populations in MEC. These sequences were found using two-photon imaging in head-fixed mice that were running on a running-wheel in darkness, without a particular task or rewards. These slow oscillations/sequences transcended borders between epochs of running and immobility, and were found in MEC but not in

parasubiculum (PaS) or visual cortex.

While this is an interesting phenomenon, I have two central concerns that diminish my enthusiasm about these results. First, I was not convinced that these sequential activations do not simply reflect traveling waves, which have been found in many brain areas. Second, I could not understand in concrete terms what could be the functional significance of this phenomenon, namely what it might be useful for.

Traveling waves: I remained unconvinced after reading the analyses described by the authors in lines 241-251, Fig. 3, and lines 1960-1993, which were aimed to convince the reader that these ultraslow oscillations cannot be a reflection of traveling waves on the surface of MEC. The problem of course is that if these are traveling waves then, although this is the first time (to my knowledge) that such slow traveling waves were found in MEC, this finding is by no means novel, because traveling waves were described in many other brain regions, and hence this would be “just another brain area” where the same was found; this would strongly weaken the claim for novelty in this work. The reasons I am not convinced that these are not simply traveling waves are as follows. One reason why Fig. 3e,f,g,h are not enough, in my mind, to rule out traveling waves is that the authors should systematically plot a scatter-plot of delta preferred phase vs. pairwise anatomical distance, and delta participation index vs. pairwise anatomical distance, rather than just separating them into two subgroups. Moreover, in Fig. 3e it looks to my eye that there is in fact an over-representation of cells with preferred phase = $-\pi$ (blue), in the north-east corner of the imaging window (dorso-medial corner); I also see in the imaging window in Fig. 3e some stripes that go from west-northwest to east-southeast; so there seems to be an anatomical organization, after all: the authors need to do an analysis to check this, for example to compute local gradients in preferred phase, and compare to shuffles of preferred phase across all cells. Furthermore, in Movie 1 it seems to me that in some of the oscillation cycles there is in fact a traveling wave across the cortical surface; the authors should analyze it like people usually analyze traveling waves, for example by calculating center-of-mass of the population-level calcium activation in each time bin, and use it to test for topographical flow of population activity across time bins. Of course this analysis must be done for each oscillation cycle separately. This leads me to the final, and biggest problem related to the analyses conducted by the authors regarding traveling waves, namely, that as they describe in the Methods in lines 1960-1993, they pooled the data across the entire session to examine the topography. This pooling would reveal topography if at each oscillation cycle the wave travels in the same direction, but it would fail to reveal traveling waves if at each cycle the wave travels at a different (random) direction, as was described in many previous studies in other brain regions (some of which the authors cited). Therefore, all the analyses performed by the authors regarding traveling waves, as well as the additional analyses that I proposed above, must be repeated separately on individual oscillation cycles. I realize that if the directions of the traveling waves would have been completely random, then the authors would likely not obtain repeated activation sequences across tens of minutes; but perhaps the directions of the waves on the cortex are variable but not entirely random, i.e. span a certain sector of angles, which could still yield the sequences they see, yet smear their analyses of topography. In fact, it is possible that those sessions where they found nice oscillations were sessions in which traveling waves traveled at a narrow set of directions (narrow sector), while those sessions where they could not detect oscillations were sessions in which traveling waves traveled at a wide (random) set of directions – and would in fact be revealed as traveling waves if the analysis

was done for each oscillation-cycle separately. In short, all the topography analyses should be repeated separately per oscillation-cycle, in addition to the analyses pooled across the entire session.

Functional significance: The authors proposed in the abstract and in the discussion that these minute-scale sequences can be used as a “scaffold” for processes such as navigation and episodic memory formation. I don’t quite understand what a “scaffold” is, and therefore I could not quite imagine what is the possible usefulness of such ultraslow oscillations, in mechanistic terms. Likewise, in the discussion (line 449 onward) the authors propose that another function of these ultraslow oscillation could be to prevent different cell classes from drifting apart; I could not understand drifting apart from what, and why does it need to be prevented; so again, it was hard for me to understand concretely what exactly is the proposed function. Likewise, I was unable to understand the other functional proposals in the discussion. I suggest that the authors add a modeling chapter to their manuscript, where they would show using concrete simulations of some mathematical mechanistic model, e.g. a neural-network model, what could be the potential functional usefulness of such ultraslow waves/sequences. Without such concrete modeling effort, I find it truly difficult to understand the potential function of these oscillations.

Additional major comments:

Related to the comment above: On lines 450-451 the authors propose that these ultraslow oscillations synchronize/modulate faster oscillations such as theta or gamma. This is an empirical question, which the authors can and should analyze directly: Do these ultraslow oscillations modulate the amplitude of theta or gamma oscillations in MEC?

Because there is no specific task in these experiments, perhaps these oscillations reflect slow oscillations of attention, or some other cognitive variables? This should be discussed.

Threshold used for the oscillation score was 0.72. I could not quite understand why 0.72. Please show that the results in the paper (in all figures) are robust to other choices of the threshold.

Line 1640-1646: Shuffling: Why did the authors randomly shuffled the time-bins for each cell? This destroys the normal physiological temporal structure of activity in each cell. It seems to me that a much better shuffling procedure (and much more common in the field) would be to rigidly and circularly shift the activity of each cell, making sure to shift by a different random amount for each cell.

Fig. 1b, right: Given the wide heterogeneity of oscillation frequencies for different neurons (seen in the right panel), how is it possible that a coherent population oscillation emerges? This panel has left me quite perplexed, as it seems to contradict the rest of the paper.

Fig. 1c,d,e: The authors show only 3 examples in the entire paper, and should add many more examples in the Extended Data Figures. In addition, the authors must add shuffles in these examples (and other examples they will add), so that the reader can see if the data is indeed more periodic

than shuffles. On a related note: What is the percentage of cells with spectral peak that protrudes above 95% or 99% of the shuffles (after Bonferroni-correcting for the number of frequency bins)?

Extended Data Fig. 3d: Even though these examples are supposed to show the 15 oscillatory sessions, I see that about half of these “oscillatory” sessions do not show clear ring structure in the plot of PC1 vs. PC2. For example, a ring is missing in Mouse 59914 session 17, Mouse 60355 session 18, Mouse 60585 session 17, etc. Why is that the case? Also: if there is no ring structure here, indicating lack of an oscillation, then how did the authors extract the ‘phase’ or angle of the oscillation, which they used for some of their analyses and indices?

Minor comments:

Lines 250-251: “Extended Data Figs 6j,6k,6l”: this should probably be Extended Data Figs 7j,7k,7l.

Line 286: “three of more ensembles”: should be “or”.

Line 809, one sample t-test: Why t-test? You plot it as median (boxplot), so a Wilcoxon test on the median seems more appropriate.

Lines 832-833 (figure legend of Fig. 3f): why the tildes (\sim), in ~ 0 rad and $\sim \pi$ rad? Why approximately? The authors should provide the exact definition, not approximate definition. The same goes for the figure legend of Fig. 3h: what is the exact definition?

Line 867: “Time bins colored in blue”: this seems more like aquamarine or some other greenish color, not blue.

Line 915: “Probability is shown on a log-scale”: looking at the figure, the y-axis seems actually to be on a lin-scale.

Line 1048, “lower than 10% of 2π ”: I can see actually examples of the phase going down by more than 10%, e.g. at $t=250$ s.

Line 1122: “The preferred phases cover the entire range of phases...”: This was not the case in 4 of the sessions, in particular all the 3 sessions in the rightmost column of this panel, as well as Mouse 60585 session 18. What does this mean?

Line 1436: twice “with with”.

Lines 1510-1528: The authors write that they used DiI to mark the corner of the prism. But in other places in the paper they wrote they used a different substance, DiL. Please clarify across the paper which substance was used, DiI or DiL.

Line 1938: “the number calcium events”: should be “the number of calcium events”.

Fig. 3f,h: Fonts are too small on the x-axis: please increase the font size.

Fig. 6i, left panel: Is there really a significant difference between the red and blue lines in the first three error bars (n = 2, 3 and 4 ensembles)? It says '*', '**' and '***', but looking at the error bars, the difference does not seem significant to me. Am I missing something?

To all Reviewers:

Thanks to all Reviewers for highly insightful comments and thanks to the Editors for helpful dialogue and advice on priorities. We appreciate the detailed feedback and suggestions, which we tried to thoroughly address. The changes we have made in response to these comments represent, in our view, substantial improvements to the manuscript and result in a paper of better quality. Below we provide a point-to-point response to the Reviewer and Editor comments (comments in *italics*).

General comment from Editors:

Your manuscript entitled "Minute-scale oscillatory sequences in medial entorhinal cortex" has now been seen by 3 referees, whose comments are attached below. While they find your work of potential interest, as do we, they have raised important concerns that in our view need to be addressed before we can consider publication in Nature. Should further experimental data allow you to address these criticisms, we would be happy to consider a revised manuscript (unless something similar has been accepted at Nature or appeared elsewhere in the meantime).

Thank you for the opportunity to submit a revised manuscript. As you will see, we have been able to address the Reviewers' comments with new experiments, analyses, and modelling work. Correspondence with the Editors (email from Noah Gray, July 6, 2022) confirmed the need for Neuropixels data to validate the observation of ultraslow oscillations and periodic sequences in the calcium imaging experiments. The dialogue with the Editors also clarified that other suggested experiments should be left for future studies (for example, nested oscillations in the gamma, theta and ultraslow bands; prevalence of the periodic sequences during free foraging, sleep and sensory enriched conditions; tracking of multiple aspects of behaviour, etc), considering the amount of work that each of these would take, and the extensive space that would be needed to present them. We therefore decided, in agreement with the Editor, to replicate the finding with Neuropixels data, but to leave other suggested experiments for follow-up projects.

We also appreciate the Editors' willingness to wait a year until we had collected the new data. Adding the new experiments was a bigger effort than the Reviewers might have imagined, as Neuropixels recordings, at the time, were established in our lab only for rats, not for mice. New procedures had to be developed to target the probes to the corresponding brain regions, using different access routes than with calcium imaging in mice, and those of Neuropixels recordings in rats. In particular, efforts were made to maximize sampling from medial entorhinal cortex layer 2 cells in mice, from which we recorded almost exclusively in the calcium imaging study (Extended data Fig. 1a-c). In the Neuropixels study, probes had to be inserted from the top of the brain, with recording sites organized linearly along shanks that upon insertion mostly penetrated entorhinal layer 3, with only the bottommost sites reaching layer 2 (see histology in Extended data Fig. 1d). Due to these geometrical constraints, the probes inevitably targeted a larger mix of cell populations than the calcium imaging recordings. Yet, despite the differences in cell samples, we found ultraslow oscillations (Fig. 1e,f) and corresponding ultraslow oscillatory sequences (Fig. 2d, Extended data Fig. 4e,f) also in the Neuropixels data. The presence of ultraslow sequences was significant (quantifications are presented below) and visible by eye in the spike raster plots. The replication of ultraslow oscillations and sequences in Neuropixels data demonstrates that the phenomena manifest also in direct measurements of spiking activity and

opens the door to future characterizations of the minute-scale oscillatory sequences using both technologies.

Please note that we made a change of terminology in the manuscript, to reflect more transparently the sequential nature of the population dynamics. We now write *oscillatory sequences* instead of population oscillation, and *sequences* instead of cycles. In order to address the comments from the Reviewers in an accurate manner, the change in terminology is not adopted in the letter. Although we hope the current terminology will be helpful to readers, we will be happy to further adjust the terminology according to the Reviewers and Editor suggestions.

Reviewer comments:

REVIEWER #1

This intriguing and really interesting article by Gonzalo Cogno et al describes calcium signals recorded with 2Ph imaging in head fixed mice walking on a wheel in the dark, about a month after injection of AAV1-Syn-GCaMP6m in the medial entorhinal cortex. The paper's main result is that such recordings reveal very slow and non-stochastic dynamics in the neurons of MEC with two principal characteristics: population activity is periodic with long periods (up to 100s), thus within the range of what is often described as infra-slow oscillations; activity is sequential, such that distributed groups of co-active neurons succeed each other in reliable order, closing a loop that typically starts over again after one full period, and typically many times over in each recording session. This activity is not a wave in a spatial sense, and it involves nearly all the imaged MEC neurons, suggesting to the authors that it must involve all known functional cell types in MEC indiscriminately.

Plotted in a low-d phase space using a variety of cross-validating dimensionality reduction techniques, these patterns trace closed loops. Much of the manuscript is devoted to technical issues related to the processing of the primary imaging data and to the analysis and interpretation of the so-processed datasets. The authors carry out similar recordings and analyses from the visual cortex and para-subiculum, and report an absence of infra-slow phenomena as seen in MEC.

This paper is technically very elaborate and very interesting in that it describes an ordered global phenomenon, apparently specific to the MEC, that has (to my knowledge) never been reported before. Infra-slow oscillations have been described before but never, I believe, as masking such ordered yet non-propagating population dynamics. The paper is intriguing (intriguing in a positive sense of the word) for a number of functional and mechanistic reasons. I will raise some of them below.

We thank the Reviewer for his or her enthusiasm about our findings and the positive feedback, as well as for the careful assessment of our manuscript.

1. Issues related to technical aspects of the experiments.

I. A first issue concerns the imaging. This technique is (I think) new in this lab and one wonders whether a round of calibration/sanity checks using familiar electrophysiological

techniques was carried out. I am sure that the first thing the authors must have wondered is whether their observations were artifactual, and linked to some technical imaging/data processing issue. Has this been tested and how (see below)?

In order to test whether our findings were linked to the recording method (calcium imaging) or processing pipeline we used, we decided – after consultation with the Editors – to perform new experiments in which we used Neuropixels probes to record neural activity in the same behavioural protocol as in the imaging study. We reasoned that if we could replicate our demonstration of ultraslow oscillations and periodic sequences using electrophysiology, we could rule out any lingering concerns about artefacts in our calcium imaging experiments and data analysis. The new experiments, based on recordings from two mice with 469 and 410 units respectively, confirm that ultraslow periodic sequences can be captured at minute-scale time scales with Neuropixels probes too. Ultraslow *oscillations* from Neuropixels experiments are shown in two new panels in Fig. 1 (e,f; examples with periods of 60 s – frequency = 0.016 Hz and 0.015 Hz respectively). Ultraslow *sequences* in Neuropixels recordings are shown in a new panel in Fig. 2 (d; example with cycle lengths of 18-43 s). In this example, we identified 12 cycles using the same algorithm as designed for the calcium imaging data (oscillation score = 1.0, above the threshold of 0.72 used in calcium data for classifying a session as oscillatory). Another example, from a different mouse, is shown in a new panel in Extended data Fig. 4f. Here 9 cycles were identified, with a cycle length ranging from 14 to 69 s (oscillation score = 0.91, session classified as oscillatory according to the threshold of 0.72 used for the calcium imaging data). Please refer to Extended data Fig. 5d for the distribution of oscillation scores in the calcium data and Extended data Fig. 5a for comparable example sessions in the calcium data. Implantation sites for the Neuropixels experiments are shown in Extended data Fig. 1d. The new Neuropixels data shown in Fig. 1 and 2 and Extended data Fig. 4e,f are of course also described in the main text (lines 83 to 88, 128 to 135 and 170 to 171).

The replication of the finding using Neuropixels probes shows that the detection of this phenomenon does not depend on the employed recording methodology. The sequences are somewhat noisier than in the calcium imaging data, as expected when sampling from a more mixed cell population (Neuropixels probes targeted multiple layers, were more ventral, and were better able to capture fast dynamics of interneurons). In Neuropixels recordings, there were also some longer periods of non-sequence activity between bouts of sequences, possibly due to small differences in training protocol (see Methods). However, importantly, the quantifications developed for sequence identification confirm the presence of periodic sequences also in the Neuropixels data, despite these differences in cell samples and behavioural testing: The oscillation score for the two example sessions in Fig. 2d and Extended data Fig. 4e,f are 1.0 and 0.91 respectively (the number of identified periodic sequences is presented above). Both scores are well above the cutoff for classifying a session as oscillatory in the calcium data (cutoff = 0.72, see Methods sections “Oscillation score” and “Analysis of Neuropixels data”, as well as Extended data Fig. 5d).

We would like to add two remarks in our response to the Reviewer comment. First, we emphasize that in the calcium imaging dataset we did not see the sequences in all the recorded sessions (Extended data Fig. 5a for recordings in MEC and Extended data Fig. 12 for recordings in visual cortex and parasubiculum), which suggests that the phenomenon we report is not an artefact of the technique or pre-processing of the data, which was common for all sessions. Second, the present work is not (as claimed) the first calcium imaging paper from our lab; calcium imaging was used also in Obenaus et al (PNAS, 2022, ref. 42), Zong et al (Cell, 2022), and Jacobsen et al (Cell Reports Methods, 2022) and the methods we used to gather and pre-process data are the same as in those papers. The fact that those papers report functional cell classes, such as grid cells and object-vector cells, with

pristine firing patterns and tuning properties on par with those reported in electrophysiological recordings from our lab, lends further support to the validity of our methods.

II. Were the parameters of the timeseries analyses (deconvolution kernel, thresholds, sampling rates etc...) explored or based on prior knowledge and factual data? In essence, what do the binary “calcium event” signals mean: subthreshold potentials, spiking, bursting, all of the above?

For the initial analysis of extracted calcium imaging traces, we used non-negative deconvolution (NND) (ref. 37), which is implemented as part of the Suite2p package (ref. 38), one of the most used analysis packages for two-photon imaging data. The algorithm itself has been validated on a large set of ground truth data from various cortical areas (Pachitariu et al., 2018; Berens et al., 2018). In comparison to other, more complex, algorithms, it was found that NND is robust to assumptions about the shape of calcium responses (single action potentials or bursts), “such that a simple decaying exponential kernel performs better than more biologically accurate kernels that include a rising time segment, and even performs better than kernels estimated directly from ground truth data” (Pachitariu et al., 2018).

For selection of appropriate values of tau, we followed recommendations of Suite2p for our analysis. Due to the absence of ground truth data for our indicator x region x imaging conditions, we used a decay tau that was at the lower end of biologically plausible values (tau = 1s), which allowed even short and low amplitude spiking responses to be picked up by the analysis and therefore did not bias our analysis towards large amplitude calcium transients (presumed bursting responses). We considered this to be particularly important for our recording of stellate and pyramidal cells in layer 2/3 of medial entorhinal cortex, which are known to have low intrinsic firing rates (2.1-2.3 Hz, Ray et al., 2014; Tang et al., 2014). This parameter is chosen globally in Suite2p (i.e., not based on the signal of each cell, but fixed for all cells in one recording).

Finally, we noticed that due to the sensitivity and the “one fit all” nature of the approach, multiple small and low amplitude calcium events were picked up. Many of these signal deflections did not resemble physiologically plausible spiking responses, i.e., did not follow the expected fast rise / exponential decay time courses of calcium imaging traces and we therefore chose to threshold based on the signal statistics (standard deviation over mean) for each cell individually (see Methods). This additional filtering step cleaned up, for each cell, deconvolution results that were either too small or too fast to be considered biologically meaningful. We have now added a brief explanation for our rationale of selecting the value for *tau* to the Methods section (line 1861 to 1865). The thresholding step is explained in line 1886 to 1889.

Because the ultraslow oscillations and periodic sequences unfolded at time scales of seconds to minutes, using a sampling rate of 7.73 Hz (bin size = 129 ms) or 30.95 Hz (original sampling frequency, bin size = 32 ms) didn't yield any differences in the obtained results, i.e. ultraslow oscillations and the population oscillation were also detected when we used the original sampling frequency. However, using a smaller sampling rate allowed us to work with smaller arrays (i.e. the number of neurons didn't change but the number of time points decreased by a factor of 4), which in some of the analyses reduced the computing time. We have commented on this in lines 1880-1884 of the methods section.

Finally, for the reasons described above, the recorded calcium traces likely picked up both (groups of) single spikes and bursts. The sensitivity of our imaging approach in combination with the calcium indicator we used was most likely not high enough, however, to distinguish between the two types of firing or to detect subthreshold potentials. As a reminder, in all analyses the calcium events were

obtained by binarizing the deconvolved calcium traces according to a threshold equal to the mean plus 1.5 times the standard deviation (all values above this threshold were set to 1, see Methods). The calcium events therefore reflect, most likely, a combination of single spikes and bursting activity. Note that the binarization is not needed to observe the periodic sequences (Fig. 2c). We have now added a sentence commenting on the recording of bursts and groups of single spikes in the Methods, line 1894 to 1896.

III. In the same vein as II, could astrocytes or other glia be responsible for or involved in these phenomena?

We thank the Reviewer for raising this interesting possibility. Because brain oscillations can be modulated by glia-related mechanisms (ref. 65), we don't rule out a role for glia (astrocytes) also in ultraslow oscillations. For example, at fast time scales, the vesicular release of astrocytes is thought to be necessary for maintaining gamma oscillations (Lee et al., 2014). At the time scale of minutes, wave-like propagation of calcium events through networks of astrocytes in the hippocampus were found to underlie reduction in the power of ultraslow oscillations in the local field potential (ref. 77). Moreover, brain oscillations can be generated by the combined action of neuronal and glial networks such that synaptic connectivity generates the rhythm and the rhythm is modulated by astrocyte-dependent synaptic regulation. One mechanism by which astrocytes may modulate slow cortical rhythms (frequency < 1 Hz) is by exciting synapses through regulation of NMDA receptors and suppressing A1 (adenosine) receptor mediated tonic inhibition (ref. 76). We now comment on this in lines 470 to 472, in the Discussion.

While we are open to a role for glia in modulation of the oscillations, we clearly rule out, however, that the reported periodic sequences are direct reflections of glial activity. First, we used a neuron specific promoter (hsyn) in our GCaMP constructs, and, in addition, glial cells look morphologically very different from the neurons we recorded. Second, wave-like processes expressed in glial activity, for example during spreading depression (Baird-Daniel et al., 2017), during which neuronal activity is gradually depressed throughout anatomical space, are typically topographically organized, reflecting the anatomical organization of astrocytes. The lack of topographical organization of the oscillatory sequences (Fig. 3e-g, Extended data Fig. 8,9, see responses to Reviewer 3, below) speaks against the idea that the reported sequences express glial activity.

IV. Following this, were tetrode or N-pixel recordings carried out in the same conditions (head-fixed mouse on a wheel, darkness, etc...), to check some aspects at least of the results, such as the existence of this phenomenon in such conditions, with its long and variable periodicity? (This might also allow one to identify nested theta and gamma that cannot be extracted from the imaging data due to the low sampling there.)

As we mention in response to comment 1i, we have performed Neuropixels recordings in head-fixed mice running on a wheel in darkness, as requested, and as explained above, we identified ultraslow sequences also in the Neuropixels data. Using the same methods that we developed for the calcium dataset, we found that the length of the sequences ranged from 18 to 43 s in the session reported in Fig. 2d (and from 14 to 69 s in the session reported in Extended data Fig. 4f).

We agree with the Reviewer that Neuropixels recordings enable the study of nested rhythms. In agreement with the Editor, we decided to leave the question of whether and how fast and ultraslow

rhythms are nested for follow-up projects and publications, considering that this would expand the paper even further. Yet, to give the Reviewers a feeling of what the data look like, we have repeated our autocorrelation and power spectral density (PSD) analyses in a small sample of single cell data collected with Neuropixels probes. In Reviewer Letter (RL) Fig. 1 (below), we show the same two example cells we present in Fig. 1e,f, but now visualized in the theta band. These two example cells show that theta modulation (RL Fig. 1) can be identified in cells that are also modulated by ultraslow rhythms (Fig. 1e,f). This result hints at the possibility that the well characterized theta and gamma rhythms interact and coexist with ultraslow rhythms in the medial entorhinal cortex. However, we decided (in agreement with the Editor) not to include this data in the paper, because a fuller characterization of nesting between the rhythms would require a larger sample as well as much more extensive analysis, beyond the scope of a revision and beyond the space that we have available for the paper.

RL Figure 1: a, PSD (left) calculated on the autocorrelation (right) of one example cell's spike train (same cell as in Fig. 1e). b, Similar to panel a but for the example cell shown in Fig. 1f. The dashed line indicates the frequency at which the PSD peaked (7.42 Hz in both example cells).

V. Is this a result of head fixation? An e-phys signature would enable one to test this in freely behaving animals.

We thank the Reviewer for posing this question. For the revised manuscript we decided, together with the Editor, to focus on replicating the finding using Neuropixels probes (to show that the results are not dependent on the recording technique) but not on further characterizing ultraslow sequence activity during free navigation in open field environments or other tasks, as this would go beyond the scope of the current paper and could take additional years of experiments and analyses. However, we agree with the Reviewer that this is an interesting and important question, and we have started working on it in follow-up projects. The data are extremely preliminary and not ready for inclusion in the paper. Yet, in RL Fig. 2 we present one example open field session recorded with one Neuropixels probe implanted in MEC where we can see clear traces of periodic sequences during the first part of the session (first 400 s), despite the more extensive changes in location, behaviour and sensory inputs compared to head-fixed stationary running. This very preliminary data suggests that sequences are a robust phenomenon not restricted to the head-fixed condition, although it will take time to determine how, when and for how long, and in which cells, they are expressed during free navigation. We hope that by publishing the letter along with the paper, we will leave with an impression of what is to be expected in free-foraging conditions.

RL Figure 2: Example session recorded in MEC during free foraging in an open field arena. Note the presence of recurring sequences during the first 400 seconds of the session.

While we chose not to include the free-foraging data in the manuscript, we have added in the main text some discussion as to what to expect in freely behaving animals. Our preliminary observations would be in line with previous work (Carrillo-Reid et al., 2015, ref. 36) showing that spontaneous neural activity in the visual cortex, recorded in the absence of sensory stimulation, can be organized according to sequences of activity, and that these sequences of activity may recur during sensory evoked experience. Our working hypothesis is that sequences are similarly present under minimalistic conditions in entorhinal activity (although here they are periodic) and that, during foraging in an open field, snippets of those sequences can be preserved, or re-activated. This was hinted at in the original manuscript in our discussion of a ‘scaffolding’ function for sequences but we realize that the idea needed better unpacking. In the revised manuscript we have rewritten this section of the Discussion, using different concepts and supporting it by a computational model that we introduced in response to a request from Reviewer 3 (lines 493 to 500 in the Discussion). We imply what we expect to see during free behaviour (frequent resets, snippets of sequences), although the collection and analysis of such data will likely take another 2-3 years.

VI. Does this phenomenon disappear in lit conditions, in sleep, or with closed loop visual feedback?

We thank the Reviewer for raising this interesting question, which is similar to the previous one. As we mention in our response to Comment V, we plan to investigate these questions in future projects, in which the goal will be to determine the extent to which the periodic sequences span different behavioural conditions as well as how such sequences are integrated into ongoing activity reflecting, for example, the animal’s changing position in the environment. In agreement with the Editor, we decided not to collect such data for the current manuscript, as explained above.

However, while we do not add such data, we now comment – in the last paragraph of the Discussion – on what could be expected during more natural behaviors, e.g. intermittent sequences, frequent resets, and segregation between neuronal subpopulations (lines 502-506). The addition of how the observed sequences may give rise to smaller snippets under more enriched circumstances, in lines 493-496 (response to the previous comment), is also relevant to the present question.

VII. In the 2-step procedure leading to the ranking of neurons, I seem to understand that each entire dataset was used to compute pairwise correlations. What happens if only a fraction of the time series (eg first third) is used for the ranking and then applied to the entire data? This test would be to check that the apparent sequence reliability is not an artifact of the ranking procedure.

The Reviewer is correct: the entire dataset was used to obtain the sorting using pairwise correlations (Fig. 2a), principal component analysis (Fig. 2b) and non-linear dimensionality reduction methods (Extended data Fig. 4c). To address the Reviewer’s comment, in Extended data Fig. 4d we now show

one example session where we sorted the data using three different chunks of data and applied the obtained sortings to the entire session. Regardless of which chunk of data we used, the global dynamics was the same. The same result is always recovered because the sequences are very stereotyped and unfold throughout the entire session.

To extend this quantification to all sessions, we reasoned that if the same sequences are present when different subsets of data are used to calculate the sortings, then the sortings have to be very similar between themselves. To calculate the similarity between sortings, we proceeded as follows. For each oscillatory session we sorted the neurons according to three different chunks of data: (i) using all data, as we do in the paper, (ii) using the first half of the data, (iii) using the second half of the data. For each cell pair in a session, we next calculated the distance between the two cells in each of the three sortings. We illustrate this calculation with a toy example: If 5 neurons were recorded, and sorting (i) was: (1,4,5,2,3), the distance between cells 1 and 5 was 2, because those two cells were 2 positions apart in the sorting. The distance between cells 1 and 3 was 1 and not 4, however, because in the calculation of distances we took into account that the sorting mirrors the position of the cells in the ring, which has periodic boundary conditions.

We next calculated the correlation between the distances in sorting (i) vs. sorting (ii), sorting (i) vs. sorting (iii) and sorting (ii) vs. sorting (iii). If sortings obtained with different chunks of data preserve the ordering of the neurons, we would expect high correlation values. We compared the obtained correlation values with the 95th percentile of a shuffled distribution obtained by shuffling the position of the cells in the sortings.

- Sorting (i) vs. sorting (ii) : 15 of 15 sessions were above the cutoff of significance. Correlation values in experimental data ranged from 0.38 to 0.85. The 95th percentile of shuffled data ranged from 0.004 to 0.015 ($n = 15$ in both experimental and shuffled data).
- sorting (i) vs. sorting (iii) : 15 of 15 sessions were above the cutoff of significance. Correlation values in experimental data ranged from 0.52 to 0.86. The 95th percentile of shuffled data ranged from 0.005 to 0.013 ($n = 15$ in both experimental and shuffled data).
- sorting (ii) vs. sorting (iii) : 15 of 15 sessions were above the cutoff of significance. Correlation values in experimental data ranged from 0.17 to 0.53. The 95th percentile of shuffled data ranged from 0.005 to 0.013 ($n = 15$ in both experimental and shuffled data).

The high correlation values obtained in these distance estimates provide support for what we illustrate in Extended data Fig. 4d: using different chunks of data for sorting the cells unveils the same dynamics. We summarize the results of the distance comparisons in the revised Methods, line 2018, as well as in the legend of Extended data Fig. 4d.

We also want to mention that the sequence reliability can be observed not only from the rasterplots (which require the cells to be sorted), but also from the phase of the oscillation (Fig. 2f and Extended data Fig. 6a). Because we equate sequence with a full turn around the ring shape manifold, one sequence is equivalent to the phase of the oscillation traversing, smoothly, the range $[-\pi, \pi)$. Hence, the reliability of the sequences in the raster plot can also be observed by the smooth periodic repetition of the phase of the oscillation (Fig. 2e,f and Extended data Fig. 6a). We comment on this because the phase of the oscillation does not depend on the ranking of the neurons. We added a comment in the manuscript in lines 147 to 149.

VIII. I did not understand the classification of running vs immobility with a threshold for immobility at non-zero speed (<2cm/s). Why this choice and how do the results look when

true immobility is used to partition the data? I also did not understand the median of velocities at 0cm/s.

We decided to threshold for immobility at a non-zero speed value (2 cm/s) in order to avoid classifying as running behaviour frames that only had minor movements of the wheel, which are detected when animals slightly move on the wheel but do not fully engage in locomotion (“twitches”). The threshold that we used is consistent with the one used in other studies, e.g. Villette et al., 2015 (ref. 35).

If the threshold is not used, our main result, which is that the population oscillation traverses epochs of running and immobility, remains the same (RL Fig. 3). We have added this information in the main text (line 340), and in the Methods (lines 2728 to 2734) where the speed threshold is reported.

RL. Figure 3: Probability of observing the population oscillation given that the animal was running (left) or immobile (right) (immobility here defined as 0 cm/s; median of probability of oscillations during running = 0.85; median of probability of oscillations during immobility = 0.65; two sample Wilcoxon signed-rank test on the probability of oscillation for running vs. immobility, $n = 10$ oscillatory sessions over the 3 animals that had the tracking synchronized to imaging, $p = 0.002$, $W = 55$). Box-plot symbols as in Fig. 3b.

The median of velocity values was 0 cm/s. We built the distribution of velocity values by pooling data across the 10 sessions (over 3 animals) for which behaviour and imaging data were synchronized. We obtained this value for the median because in some of the sessions the animals were immobile for most of the session (RL Fig 4a - we have now added a comment in the legends of Extended data Fig. 2a). Specifically, for each of the 10 oscillatory sessions, the smallest speed value was always 0 cm/s, and the largest and median (calculated over the entire session) were, respectively:

- Mouse 60355 – Session 18 = 75.3 ; 0 cm/s
- Mouse 60355 – Session 19 = 31.2 ; 0 cm/s
- Mouse 60355 – Session 20 = 16.4 ; 0 cm/s
- Mouse 60585– Session 16 = 62 ; 2.9 cm/s
- Mouse 60585– Session 17 = 55.9 ; 16.7 cm/s
- Mouse 60585– Session 18 = 53.4 ; 0 cm/s
- Mouse 60584– Session 16 = 44.7 ; 4.4 cm/s
- Mouse 60584– Session 17 7= 39.7 ; 18.8 cm/s

Mouse 60584– Session 18 = 39 ; 11.5 cm/s

Mouse 60584– Session 19 = 44.2 ; 10.1 cm/s

Note that there were no systematic changes in the smallest speed across consecutive sessions (which could have happened due to habituation or training). We have now included this information in the legend of Extended data Fig. 2a.

When we do not apply the threshold, the median is 1.3 cm/s (RL Fig. 4b), i.e. still very low. We have now added this information in the Methods section, lines 2728 and 2729.

RL Figure 4: Histograms of speed values across 10 oscillatory sessions with behavioural information synchronized to imaging data (3 animals). Each histogram has a total of 238495 counts, equivalent to all frames pooled across sessions (bin size = 129 ms). a, Speed distribution when a threshold for immobility of 2 cm/s is applied and all values below the threshold are set to zero. Note the gap between speed values of 0 cm/s and 2 cm/s. Median = 0 cm/s, as reported in the article. b, Similar to a, but without applying any threshold. In this case the median is 1.3 cm/s.

2. Issues related to the periodicity and sequences

1. One of the most interesting and astonishing aspects of this study is the slow periodicity of this activity but also the variations of its period—sometimes within a recording epoch, more often across recording episodes (eg EDF 4a,b) and finally, across animals. In one case (mouse 59914, episodes 1, 2, 5), it even seems as if activity runs in reverse (I assume episodes 4 and 5 plot the same neurons). In others, as with mouse 59911, or mouse 60355 session 16, activity is classified as non-periodic but it is clearly far from random and highly correlated across the population (presumably the rank ordering of the neurons rests on these pairwise correlations?). All this richness of ordered (simultaneous or sequential, with tunable speed) but non-spatial-wave population activity suggests something highly non-trivial (and thus very interesting) if it is not linked to some external (technical) cause, or a sensory signal. I could not help but wonder if the search for a correlation with animal motion (apparently non-existent, or at least not clear) is sufficient to eliminate all external or behavioral correlates.

We thank the Reviewer for raising this question. First, we agree with the Reviewer that bouts of population activity that were classified as non-periodic still display interesting structure, even if it is not in the form of ultraslow periodic sequences, and that when sequences are present, they may exhibit a variety of dynamics, including the expression of small snippets of sequence activity and even occasional reversals of sequences. We expect these dynamics to be much richer when data are recorded in non-minimalistic environments, such as during foraging or navigation, and during encoding of experience. As explained above, experiments on sequence activity under such behavioural conditions have started and we will know a lot more in 2-3 years. After consultation with the Editors, we have chosen not to include such data and instead focus the paper on the characterization of the ultraslow periodic sequences under sensory minimized conditions. However, because this is an interesting question and many readers may ask the same, we have now elaborated further on the rich dynamics – and what we would expect to see under more natural conditions – in the Discussion (lines 504 to 508). See also our response to Comments 1.V and 1.VI above.

We further agree with the Reviewer that searching for a correlation between the periodic oscillation and animal motion does not rule out other behavioural correlates. One such other behavioural correlate that has come to our mind is behavioural arousal. With our experimental protocol we only had access to movement information, through the position of the animal on the wheel, from which we calculated the animal's speed and acceleration, and identified epochs of locomotion. Acquiring other kinds of behavioural data would require new experiments that would be beyond the scope of this manuscript, as discussed above (see first page of the letter).

Yet, we agree that the comment on behavioural correlates is interesting. In the revised Discussion (line 467 to 470), we have thus added a comment on previous work suggesting that ultraslow oscillations in neural population activity (such as shown in Fig. 1) may be brain-wide and related to fluctuations in arousal functions mediated by neuromodulatory inputs from the brain stem (e.g. ref. 74).

II. The correlation times series seem to indicate occasional discontinuities (eg fig 2b, second and third sequences). Likewise, when one zooms out on the displays, one occasionally perceives other structures, such as a constricted zone around cell 200 in fig 2a, very slow periodicity (EDF4 mouse 59914, last row), etc. Are these real, or artifacts of the ranking are displayed (see 1-VI)?

Those discontinuities are indeed real. Discontinuities can be observed both in the raster plots, as the Reviewer indicates, as well as when plotting the phase of the oscillation as a function of time, as in Extended data Fig. 6. In the presence of discontinuities there are disruptions in the smooth progression of the phase (as a reminder, the calculation of the phase does not rely on sorting the cells). We allowed for small discontinuities in our cycle-identification algorithm under the rationale that while sequences could sometimes progress very neatly, at other times they might be noisier and yet be consistent with the population activity traversing one full turn around the ring manifold (see legend of Extended data Fig. 6). Specifically, when we required the phase of the oscillation to smoothly cover the range $[-\pi, \pi)$ rad, we allowed for small fluctuations in the phase. We have added a comment in the Methods (lines 2246 to 2247) to indicate that the tolerance for fluctuations in the phase of the oscillation within one cycle was 0.6 rad.

III. One of the most astonishing aspects (to me) is the sequential but non-topological activation of these populations and the fact that they appear to form closed loops. That they

form closed loops suggests that the authors' sample (over a small fraction of the MEC) of neurons is sufficient to describe the entire system's path. The putative ensembles that succeed each other to close a loop would therefore occupy the entire MEC, and be intermingled throughout. Is this a correct understanding?

Thank you for the interesting comment. We agree with the Reviewer's interpretation. The fact that the ultraslow sequences are periodic and not topographically organized (see section in the manuscript about traveling waves, as well as the responses to the comments from Reviewer 3) suggests that within our field of view (FoV) we sampled enough neurons to see the periodic sequences.

Our data also raise the possibility that other MEC neurons, outside the FoV, may be intermingled within the same sequences. This idea is illustrated in Extended data Fig. 7d. In that figure we downsampled the number of cells in one example session, and we found that when we only kept 80% or 60% of the total number of recorded cells, the sequences looked the same as when all cells are included. This shows that cells that are not included in the visualization and quantification of the sequences could still participate in the observed dynamics. We now comment on this in lines 220 to 222 of the manuscript.

IV. If so, what mechanisms could account for this spatially widespread coordination of widely distributed but more or less synchronized ensembles, plus a very slow propagation of activity across ensembles that live in the adjacent physical space?

We thank the Reviewer for raising this question. Oscillations at time scales of tens of seconds to minutes have been reported in individual cells of multiple dispersed brain areas (references 20 to 26). These oscillations can be correlated with arousal indicators such as pupil area and blood flow, suggesting that ascending neuromodulatory systems could play a role in controlling the periodic activity (ref. 74; see response to Reviewer 3 comment, below). Oscillations could also be modulated via astrocyte-dependent synaptic regulation (refs. 76,77; see response to section 1, Comment III). These mechanisms suggest that the brain is capable of generating single-cell ultraslow periodic activity across a widespread network of brain areas.

However, while our data support the notion that ultraslow oscillations are widespread (appearing in visual cortex, parasubiculum and MEC), those oscillations were only in the MEC organized at the network level into recurring activity sequences, pointing to MEC as having mechanisms for sequential coordination of single cell oscillations that are not present in parasubiculum or visual cortex. As we note in the Discussion (lines 474 to 482), the population oscillation could be consistent with dynamics expected in a one-dimensional continuous attractor network (references 78 to 80), where cells are conceptualized as lying on a functional ring with positions determined by the cells' position in the sequence. It has not been determined yet whether such ring-like connectivity exists among different classes of MEC neurons and, in case there is such connectivity, which signal is responsible for moving the bump of activity along the ring. Sequential activity could be generated by various types of structured connectivity, for example in recurrently connected networks (references 81 and 82) and in feedforward networks in which sequences may arise through synfire chains or rate propagation (reference 83 to 87). More work has to be done to build biologically plausible models that shed light on the mechanisms underlying ultraslow repeating sequences with the variability at the single-cell level that we here observed in the MEC population oscillation.

The third-last paragraph of the Discussion is dedicated to these questions. We have revised it extensively, summarizing the above ideas and adding new text.

V. The E and MB Moser lab nicely showed a while back that grid (spatial) periodicity varies in discrete steps along EC. Would matching discontinuities (here, in dynamics) have been detected in the present data, given the sampling constraints on these imaging experiments?

Yes, we think such discontinuities for grid cells would have been detected in our data too. Our field of view was slightly larger than the one used in two other calcium-imaging papers from the lab: Obenhaus et al., 2022 (ref. 42) and Zong et al., 2022. In those papers, with a FoV of 500 μm x 500 μm , we identified two modules of grid cells when recording in open field sessions using a box of 80 cm x 80 cm. Because the field of view in the present study was larger (600 μm x 600 μm in 3 MEC animals, 1000 μm x 1000 μm in 2 MEC animals) and covered the area included in the Obenhaus and Zong papers (Extended Data Fig. 1), we expect to have grid cells in our cell sample, and those grid cells would be expected to be recruited from at least 2 modules. However, we don't have a direct demonstration for the existence of grid cells from multiple modules in our data because we didn't record any open field sessions. We comment on this in line 224: 'Because the oscillatory sequences involve the vast majority of MEC neurons in the recording region (94%, median calculated over 15 oscillatory sessions across 5 animals), and given that multiple cell types can be recorded within the field of view (references 6, 41 and 42), the sequences most likely include a mixture of functional cell types such as grid, head-direction, and object-vector cells, with grid cells spanning more than one module'.

The participation index was not topographically organized in anatomical space (see legend of Extended data Fig. 8a, as well as analysis on traveling waves suggested by Reviewer 3, in particular RL Fig. 9,10), suggesting that cells that were ventrally located did not participate in the sequences less than cells that were dorsally located. The frequency of single cell calcium activity did not appear in discrete steps (Extended data Fig. 7c; see also RL Fig. 12 and response to Reviewer 3 comment, below), suggesting that at the ultraslow time scale, cells tend to share the same periodicity.

We share the Reviewer's curiosity and plan to explicitly address the relationship to grid cells and grid-cell modules in future Neuropixels studies of animals freely foraging in an open field arena (as in RL Fig. 2). As indicated above, in agreement with the Editor (see first page of the letter), we decided not to include these new experiments in the present study.

As we mention above, we comment on this topic in line 224 and in the legend of Extended data Fig. 8a.

VI. An interesting feature of sequences might be their initiation from a period of inactivity. Can these initiating groups be identified and compared across successive sequences separated by inactivity?

We thank the Reviewer for posing this interesting question. We started by determining whether there is an increase in the neural population activity after sequence onset (we assume this is what the Reviewer implies by 'initiation from a period of inactivity'). We calculated the mean calcium activity rate in a time window right before and right after the sequence onset. Because the cycle length varied from session to session (Extended data Fig. 6d), in order to compare the calcium activity across sessions, we set the length of the time window to one oscillation bin size (Extended data Fig. 10I, see Methods line 2205). As a reminder, the oscillation bin size is different for each session and chosen

such that the sequences unfold in approximately 10 oscillation bins in the session. We next divided the neural population into 10 ensembles of cells (Fig. 4a) and for each ensemble repeated the calculation of mean activity rate before and after sequence onset. We didn't find significant changes in the neural population activity (calculated as: activity *after* sequence onset – activity *before* sequence onset) (RL Fig. 5a). When we analysed ensembles separately, we found that ensembles that got active early in the sequence (1 to 4 - typically depicted at the bottom of the raster plot), showed a slight increase in their mean calcium activity rate compared to the pre-sequence time window. Ensembles that were located in the middle of the sequence (5 to 7) did not present significant changes in their activity and ensembles that fired late in the sequence (8 to 10) showed a slight decrease in their mean calcium activity rate after sequence onset (RL Fig. 5b).

These results would be consistent with ensemble 1 being the initiation group the Reviewer refers to, since those are the cells that fire right after sequence onset. However, we want to remark that while the rasterplots also seem to indicate that the sequences always start at ensemble 1 (Extended data Fig. 6a and Extended data Fig. 10n), in the presence of brief interruptions, the activity does not reset, but continues its progression along the sequence (e.g. see disruptions in the sequence progression in the first 500 s of Extended data Fig. 6a).

Because we didn't find changes at the population level, and the changes for the ensembles were very small, suggesting that any effect is very weak, we decided not to include these results in the manuscript and only present them here (we assume that the letter will be published along with the manuscript). We would of course include these analyses in the paper if the Reviewer and Editors consider them important.

RL Figure 5: Change in the mean calcium event rate right after sequence onset, compared to the activity before sequence onset. The time window used for the calculation was one oscillation bin. a, The mean activity is calculated over all recorded neurons. b, The mean activity is calculated separately for each of the 10 ensembles. n=71 sequence onsets for each group (all neurons, or each ensemble).

REVIEWER #2

This work identified slow oscillatory activity in the medial entorhinal cortex (MEC) through calcium imaging of many neurons. The period of the oscillation was about 50-60 seconds and the majority of MEC neurons participated in this oscillation but different neurons fired at

different oscillatory phases. As a result MEC neurons tend to fire in a sequential manner: their firing rate modulation followed a slow sequential order. This oscillatory activity was present more frequently during movement as compared to immobility periods. Finally, this kind of sequential network activity associated with oscillations was not seen in the visual cortex and parasubiculum. The finding is surprising and required simultaneous imaging of many neurons to demonstrate the periodic sequential activity. I liked the dimensionality reduction approach to identify oscillatory phases, especially since summing up the activity of all cells would not be able to identify this sort of oscillatory pattern.

We thank the Reviewer for his positive feedback and enthusiasm about our findings.

Overall the manuscript used very original data analysis approaches. I have two questions only:

1. It looks from the shown examples that a subgroup of neurons in the visual cortex engaged transiently in periodic activity as well, albeit it was not sequential but up-down state-like but at a much lower frequency than cortical slow wave oscillations in sleep. Could this be quantified in some way and acknowledge that slow oscillations of a different form can exist outside the MEC?

We thank the Reviewer for noticing this interesting aspect of our results. Indeed, a subgroup of cells in the visual cortex engaged in ultraslow periodic activity (Fig. 6b, new panels with visualization of ultraslow oscillations in two example cells; Fig. 6d), despite the absence of periodic sequences at the neural population level (Fig. 6f, same session as Fig. 6b). In order to quantify the fraction of cells displaying ultraslow oscillations, we proceeded the same way we did for the MEC (line 79 in the Results section and from line 1918 on in the Methods section; please also see our response to Reviewer 3's comment related to the fraction of MEC cells with power spectrum peak protruding the 95th percentile of shuffled data). Briefly, for each of the 6037 visual cortex cells recorded across 19 sessions, we divided the calcium activity into consecutive epochs of length 20 s. We next shuffled those epochs (and preserved the ordering of the time bins within each epoch, bin size = 129 ms). Out of the 6037 cells, 4487 displayed a peak in the power spectral density that was above the 95th percentile of shuffled data (74%). However, these single-cell oscillations were not organized into periodic sequences of neural activity (Fig. 6f). We believe that it is the intrinsic connectivity of MEC that plays a key role in orchestrating single cell oscillations into periodic sequences, whereas ultraslow oscillations not organized in sequences may exist in much more widespread brain regions, consistent with previous findings cited in the Introduction and at the beginning of 4th paragraph of the Discussion (see also response to Comment 2.IV from Reviewer 1).

The periodic activity identified in the visual cortex seemed to be very much locked to oscillations in the running behaviour of the animals (RL Fig. 6), in direct contrast with our finding in the MEC (Fig. 5a). In order to determine whether the ultraslow oscillations identified in the visual cortex were correlated with the animal's running behaviour, for each cell in a session we have now calculated the correlation between its calcium activity and the speed of the animal in that session. For completeness, we performed this analysis for all sessions in parasubiculum, visual cortex and MEC for which the behavioural data was synchronized to imaging data (in MEC we only considered oscillatory sessions, but separate analyses showed that results remain exactly the same if all sessions are considered). We found that neuronal activity in the visual cortex was more correlated with the animal speed than

sessions in the MEC or in PaS (Extended data Fig. 13j). This new result suggests that the periodic activity we observed in medial entorhinal cortex and visual cortex might be different in nature, with the former being more independent of ongoing behavior, and the latter more behaviourally tuned. The correlation between neural activity and running in the visual cortex may result in patterns of activity that resemble the alternation of up and down states during sleep, as the Reviewer noted.

RL Figure 6: Snippet of 500 s of the visual cortex session presented in Fig. 6d. Top: raster plot. Time bins colored in aquamarine indicate that the animal ran faster than 2 cm/s. Middle: Instantaneous speed of the animal. Bottom: Position of the animal on the wheel, expressed relative to an arbitrary point on the wheel.

2. The group has been recently published work in the same journal in which they recorded from many MEC cells using neuropixel probes. The reviewer cannot help being curious whether similar oscillations could be detected when real action potentials are detected, not calcium signal and animals engage in more natural behavior.

Jozsef Csicsvari

We appreciate the Reviewer's curiosity. In response to Reviewer 1 and 2's comments on validation by Neuropixels recordings, we decided to perform new experiments using the same behavioural protocol with Neuropixels probes (see our general comments on page 1, and our response to Comment 1 from Reviewer 1, for a detailed summary of the new experiments and analyses). As noted in our responses in these earlier sections of the letter, we were indeed able to detect ultraslow oscillations both at the single cell (Fig. 1e,f) and population levels (Fig. 2d and Extended data Fig. 4e,f) using Neuropixels 2.0 silicon probes.

After consultation with the Editors, we chose to target all new Neuropixels experiments towards reproducing the results obtained with calcium imaging in the previous version, not adding new behavioural protocols. Yet, as explained in our responses to Reviewer 1, we already have some preliminary figures that suggest that snippets of the ultraslow periodic sequences are also present during more natural behaviour, such as free foraging in an open field while animals search for cookie crumbs (RL Fig. 2). These data are somewhat anecdotal, however, and in agreement with the Editor's advice, we prefer to perform a larger number of recordings and analyses before we publish them.

However, we expect that the present letter will be published along with the paper, allowing similarly curious readers to get a glimpse into what might come in future work.

REVIEWER #3

In this manuscript, Cogno et al. provide evidence for ultra-slow oscillations that create slow minute-scale sequences in medial entorhinal cortex (MEC), which entrain large neuronal populations in MEC. These sequences were found using two-photon imaging in head-fixed mice that were running on a running-wheel in darkness, without a particular task or rewards. These slow oscillations/sequences transcended borders between epochs of running and immobility, and were found in MEC but not in parasubiculum (PaS) or visual cortex.

While this is an interesting phenomenon, I have two central concerns that diminish my enthusiasm about these results. First, I was not convinced that these sequential activations do not simply reflect traveling waves, which have been found in many brain areas. Second, I could not understand in concrete terms what could be the functional significance of this phenomenon, namely what it might be useful for.

We thank the Reviewer for the detailed reading and assessment of our article. We agree that our previous analyses on travelling waves were not exhaustive enough and did not account for the possibility that travelling waves may change their direction of propagation across successive cycles (although this possibility would raise the question of how differently directed waves are integrated into the same continuous oscillation, across multiple cycles). The revised version of the manuscript addresses this possibility. We have implemented all the analyses suggested by the Reviewer. Results are presented in Fig. 3, in two new extended data figures (number 8 and 9), and in a new paragraph in the main text of the paper. Following the Reviewer's advice, we also developed a model that illustrates one possible functional significance of the population oscillation. We have devoted one new extended data figure to the model (number 14), as well as a whole new paragraph in the main text of the paper. Please find below detailed responses to the comments.

Traveling waves: I remained unconvinced after reading the analyses described by the authors in lines 241-251, Fig. 3, and lines 1960-1993, which were aimed to convince the reader that these ultraslow oscillations cannot be a reflection of traveling waves on the surface of MEC. The problem of course is that if these are traveling waves then, although this is the first time (to my knowledge) that such slow traveling waves were found in MEC, this finding is by no means novel, because traveling waves were described in many other brain regions, and hence this would be "just another brain area" where the same was found; this would strongly weaken the claim for novelty in this work. The reasons I am not convinced that these are not simply traveling waves are as follows. One reason why Fig. 3e,f,g,h are not enough, in my mind, to rule out traveling waves is that the authors should systematically plot a scatter-plot of delta preferred phase vs. pairwise anatomical distance, and delta participation index vs. pairwise anatomical distance, rather than just separating them into two subgroups.

Thank you for these suggestions. We agree that the separation into two groups did not represent a thorough enough approach for the quantification of travelling waves. We have removed that analysis from the paper and replaced it by the analyses suggested by the Reviewer. Instead of comparing the extremes of the delta preferred phase (or delta participation) distribution, we now provide, for each oscillatory session, a 2D histogram of anatomical distance (x) versus phase or participation difference (y). 2D histograms are shown for each of the 15 oscillatory sessions in the entire dataset (Extended Data Fig. 8a). Frequency is colour-coded. We also made scatter plots, as the Reviewer suggested (RL Fig. 7). However, the large number of datapoints made the plots very difficult to interpret. For this reason, and to allow readers to see any differences in density of data points, in the paper we decided to include the histograms and not the scatter plots.

RL Figure 7: Difference in the preferred phase of two cells as a function of the distance between their positions in the field of view. Each dot is a cell pair. The plot shows data from example session 17, animal #60584, presented in the second row and second column of Extended data Fig. 8a. The density of dots may mask relationships between anatomical distance and differences in phase; therefore, we have chosen to illustrate the relationship with colour-coded 2D histograms instead (Extended data Fig. 8a).

Distribution of phase:

The histograms in Extended Data Fig. 8a show the difference in preferred phase between pairs of cells as a function of the cell pair's anatomical distance. In order not to count the same data twice, each plot was made using $N*(N-1)/2$ samples (each sample is a cell pair), where N is the total number of cells recorded in the session. It can be seen that there is no accumulation of points along the diagonal, which would have been expected with travelling waves. In the presence of travelling waves we would expect a significant correlation between differences in preferred phase and anatomical distance between cells within the field of view (FoV). We tested for the correlation between the two variables by calculating the absolute value of the Pearson correlation (we were interested in a significant correlation, regardless of whether it was positive or negative). The correlations between phase difference and anatomical distance ranged from 8.5×10^{-5} to 0.015, i.e. they were extremely small. To determine statistical significance we used as cutoff the 95th percentile of a distribution built with the absolute values of the correlations obtained in shuffled data, where the preferred phases were shuffled in the FoV. We then compared the absolute value of the correlation in experimental data with this cutoff (95th percentile values ranged from 0.005 to 0.037 in shuffled data, $n = 15$). Only one session

(session 16, animal #60585) of the experimental data was classified as having a significant correlation (1/15, probability = 0.37, binomial probability distribution; not statistically significant to a chance level of 5%). The correlation value of this session was very low (0.007).

We next tested whether the absolute value of the difference in preferred phase was correlated with the anatomical distance, as this dependency could explain the small correlation values presented in the paragraph above. In this case, the correlations ranged from 7.5×10^{-4} to 0.028. Only session 19 (animal #59914) was classified as significant (1/15 sessions, probability = 0.37, binomial distribution; not significant when compared to a chance level of 5%), yet again the correlation was very low (0.028; 95th percentile values across sessions ranged from 0.005 to 0.053). Taken together, these results show that the preferred phases are not correlated with how far apart cells are in the FoV.

Due to the high number of data points in the scatter plots, and in order to show the distributions of differences in preferred phase, we also made box plots that show the median and the tails of the differences in preferred phase for successive spatial bins of 150 μm (RL Fig. 8). Across bins the distributions appear qualitatively the same and there was no bias towards smaller differences in preferred phase for smaller bins. We decided not to include these plots in the paper because they do not provide additional information as compared to the 2D histograms. The histograms are preferable, in our opinion, due to differences in sample size across bins, which could influence the comparisons.

The lack of correlation between the differences in preferred phase and the anatomical distance as shown in the box plots as well as the scatter plot and histograms suggest the absence of travelling waves.

RL Figure 8: Box plots of the difference between preferred phases, plotted from $-\pi$ to π , as a function of the distance between the positions in the field of view, indicated in bins. Each bin contains anatomical distances within a range of 150 μm , with bin 1 having distances in the range $[0, 150) \mu\text{m}$, bin 2 $[150, 300) \mu\text{m}$, bin 3 $[300, 450) \mu\text{m}$ and so on. Note that in some sessions the field of view was larger than in others, yielding larger distances between cells, and hence more bins. Each plot shows the data of one oscillatory session across 5 animals. The preferred phase of each cell was calculated as the mean phase at which the calcium events occurred after concatenating all cycles.

Distribution of participation index:

For completeness we repeated the same analysis for the participation index (PI), as suggested by the Reviewer. We plotted the 2D histograms of the difference between the participation index (PI) of two cells and their pairwise anatomical distance (RL Fig. 9) and made corresponding box plots (RL Fig. 10). Because for each cell the PI is calculated as the fraction of cycles needed to explain 90% of the total number of deconvolved calcium events (Methods), the PI is a rational number and therefore the subtraction between two PIs is also a rational number, which is why the differences in PI for cell pairs appear as a set of discrete values, with the number of values depending on how many cycles were identified in the session. The correlation values were also here very small and ranged from -9.3×10^{-4} to 0.040. When we shuffled the PI values in the cell sample and tested for significance with the same procedure used for the preferred phases, we found that the correlation of 2 sessions (sessions 16 and 18, animal #60584, correlation = 0.033 and 0.040 respectively) were higher than the 95th percentile value of the shuffled data (which ranged from 0.025 to 0.054, $n = 15$). However, this fraction was not significant when compared to a chance level of 0.05 (2/15 sessions, probability = 0.13, binomial distribution). Because we are not aware of any clear hypothesis on how these two quantities, PI and anatomical distance, should correlate in presence of travelling waves, we decided to only include figures for preferred phases. These figures, shown in Extended Data Fig. 8a, are described in text between lines 260 to 262. For PI, we only include a text line (in the Extended data Fig. 8a legend), in the absence of an obvious hypothesis; however, we would of course give these analyses a more prominent place in the paper if the Reviewer and Editor consider it relevant.

RL Figure 9: Absolute value of the difference between the participation index of two cells as a function of the distance between their positions in the field of view. Each dot is a cell pair. Each plot shows the data of one oscillatory session (across 5 animals). The participation index is calculated as the fraction of cycles needed to explain 95% of the total number of deconvolved calcium events. Note that that data appears as discrete (most clearly on trials with few sequences) because the participation index is a rational number and therefore the difference between two participation indexes is also a rational number. Session number and animal ID are organized as in Extended data Fig. 8a.

RL Figure 10: Box plots of the absolute value of the difference between the participation index of two cells as a function of the distance between their positions in the field of view. Each plot shows the data of one oscillatory session across 5 animals. The participation index is calculated as the fraction of cycles needed to explain 90% of the total number of deconvolved calcium events. Conventions as in Fig. 3b.

Distribution of phase for single cycles:

The lack of correlations presented in Extended data Fig. 8a and in RL Fig. 7,8 points to a possible absence of traveling waves during the MEC population oscillation. But those analyses were performed by calculating the global preferred phase and participation index of each cell after pooling all cycles in a session. The Reviewer suggests that travelling waves could still be present if they move in a different direction in each cycle (in which case they would cancel out after averaging). In the revised manuscript, to test for the presence of travelling waves without assuming similar wave directions across successive cycles, we repeated the quantification of correlation between the difference in preferred phase as a function of pairwise anatomical distance but now for each cycle separately (see manuscript, between lines 266 and 273). We calculated the preferred phase of each cell in each cycle by computing the mean phase at which the calcium events occurred in that individual cycle. In Fig. 3f we show data for a single example cycle from session 17, animal #60584, same as in Fig. 2a (see also Extended Data Fig. 8a). The correlation between the difference in preferred phase and the anatomical distance was 0.0026, which (unsurprisingly) did not pass the cutoff for significance. We performed the same analysis on each of the 421 sequences recorded across the 15 experimental sessions. In the experimental data the correlation values ranged from 6.4×10^{-6} to 0.147 ($n = 421$). We found that 27 cycles were classified as significant when comparing to shuffled data (95th percentile of shuffled distribution ranged from 0.007 to 0.237, $n = 421$). While this number (27 out of 421) was slightly above a chance level of 0.05 ($0.05 \times 421 = 21$ sequences), for those 27 cycles the correlation values were all very low, ranging from 0.008 to 0.137 (Fig. 3g).

When taking the absolute value of differences in preferred phase, in experimental data the correlation values ranged from 5.5×10^{-6} to 0.137. Out of the 421 cycles, 23 had correlation values that were above the 95th percentile of the shuffled distribution, yet this fraction was at a chance level of 5%

(chance level of 0.05 = 21 sequences out of 421; 95th percentile of the shuffled distribution ranged from 0.006 to 0.246).

In conclusion, all correlations between anatomical distance and phase difference were extremely low and contained within the values expected by chance. Please note that we couldn't repeat this analysis for the participation index, because this quantity is defined as the fraction of cycles that are needed to capture 90% of the calcium events, and therefore cannot be extended to a cycle-by-cycle estimation.

Moreover, in Fig. 3e it looks to my eye that there is in fact an over-representation of cells with preferred phase = $-\pi$ (blue), in the north-east corner of the imaging window (dorso-medial corner); I also see in the imaging window in Fig. 3e some stripes that go from west-northwest to east-southeast; so there seems to be an anatomical organization, after all: the authors need to do an analysis to check this, for example to compute local gradients in preferred phase, and compare to shuffles of preferred phase across all cells.

We thank the Reviewer for this suggestion. In order to implement this analysis, we first looked at other papers that compute local gradients, typically in the presence of traveling waves. In many of these papers, e.g. reference 117, the phase of the oscillation is calculated through the Hilbert transform applied to the activity of each electrode. This procedure, however, requires concatenating cycles. In order to perform the calculation of local gradients for each cycle separately, we took two different approaches, which are described in the revised manuscript between lines 274 and 284. The results are shown in Extended data Fig. 8b,c and Extended data Fig. 9a,b.

First, we used as a proxy for local gradients the similarity in preferred phases of all cells within each spatial bin of the FoV. We estimated the similarity in preferred phases by calculating the mean vector length (MVL) of the distribution of preferred phases within each bin of the FoV. The analysis was performed for individual cycles, in agreement with the Reviewer's previous comment.

For each session we binned the FoV using spatial bins of 100 μm x 100 μm (6x6 bins in total for sessions 1 to 10, 10x10 bins in total for sessions 11 to 15), or 200 μm x 200 μm (3x3 bins in total for sessions 1 to 10, 5x5 bins in total for sessions 11 to 15) (note that for 10 of the 15 oscillatory sessions the FoV was 600 μm x 600 μm , animals #60355, #60584 and #60585; while for 5 of the 15 oscillatory sessions the FoV was 1000 μm x 1000 μm , animal #59914). Next, we calculated the preferred phase of each cell per cycle as described above, and for each cycle and every spatial bin of the FoV we calculated the MVL (we only considered spatial bins with more than 10 cells). If the MVL was 0, then all preferred phases in that bin were different and homogeneously distributed between $-\pi$ and π , whereas if the MVL was 1 then all preferred phases were the same. In the presence of a travelling wave, in each bin we would expect a high MVL value compared to chance levels.

In Extended data Fig. 8b,c we present the results for one example cycle of one session (the same as depicted in Fig. 3f). In this session the FoV was 600 μm x 600 μm . When using 100 μm x 100 μm bins, there were cells in most of the bins, while when using 200 μm x 200 μm bins there were cells in all bins (Extended data Fig. 8b,c top). For each spatial bin we compared the MVL with the 95th percentile of a shuffled distribution created by shuffling the preferred phases across the FoV before calculating the MVL (a total of 1000 shuffled realizations). In the example cycle, when using 100 μm x 100 μm bins, only 17 bins had more than 10 cells. From the 17 bins, only one was classified as having more similar phases than expected by chance (1/17, probability = 0.37, binomial distribution); when using 200 μm x 200 μm , only one bin out of eight with more than 10 cells was classified as having cells with similar phases (1/8, probability = 0.28, binomial distribution).

We repeated the same analysis across all cycles and sessions, and for each spatial bin in each cycle of each session we compared the 95th percentile of the shuffled distribution with the MVL calculated on experimental data. The MVL calculated on experimental data was larger than the cutoff for significance in 115 spatial bins out of 2392 when using 200 μm x 200 μm bins (115/2392, not larger than expected at a chance level of 0.05: 120/2392), and for 121 out of 2448 when using 100 μm x 100 μm bins (121/2448, not larger than expected at a chance level of 0.05: 122/2448). The lack of similarity in preferred phases within spatial bins is inconsistent with a coherent oscillation in that spatial bin, and therefore inconsistent with the presence of travelling waves. The new analyses shown in Extended data Fig. 8b,c are reported on lines 274-279 in the main text, in addition to text in the figure legend and in the Methods (line 2483-2501).

The analysis presented above is focused on the degree of similarity between preferred phases. In order to avoid small cell sample effects, and effects of adding a threshold number of cells for bins to be included when calculating similarity with the MVL measure above, we decided to also calculate the difference in preferred phases for pairs of cells. The calculation was performed on cell pairs located within small neighborhoods in the FoV, expecting that in the presence of travelling waves the differences in preferred phases of cell pairs within such small neighborhoods would be smaller than expected by chance. For each cell in the FoV, all other cells that were located within a radius of 50, 100 or 200 μm were identified and differences in preferred phase of cell pairs within those areas were calculated. We pooled all the phase differences in one cycle for each radius separately and computed the mean and the median of the obtained distributions. For comparison, we shuffled the phases across all cells 200 times and for each shuffled realization we created a distribution of differences in preferred phases and calculated its mean and median. Because in the presence of travelling waves we would expect smaller differences in preferred phases of neighbouring cells than in the shuffled data (as in the MVL analysis above), we compared the mean and median calculated on experimental data with the 5th percentile of the distribution of means and medians obtained from shuffled data. In Extended data Fig. 9a we show results for one example cycle (same as in Fig. 3f). For neighborhoods of 50, 100 and 200 μm , both the mean and the median were above the cutoff for significance, consistent with a lack of traveling wave in this individual cycle. We next extended this analysis to all cycles across all sessions (Extended data Fig. 9b). The differences in phase of neighboring cell pairs were no different from the median of the shuffled distribution and they were rarely lower than the 5th percentile of the shuffled distribution. When considering neighborhoods of 50 μm , 16 out of 421 cycles had a mean below the cutoff for significance (16/421, not larger than expected at a chance level of 0.05: 21/421), and 16 out of 421 cycles a median below the cutoff for significance (16/421, not larger than expected at a chance level of 0.05: 21/421). For neighborhoods of 100 μm , 16 and 19 cycles (out of 421) were below the cutoff for the mean and median respectively (16/421 and 19/421, neither fraction was larger than the chance level, which was 21/421 for $p=0.05$). For neighborhoods of 200 μm , the count was slightly above the cutoff for the mean, and slightly below for the median (25/421 and 18/421 respectively, chance level of 0.05: 21/421) (Extended data Fig. 9b). These new analyses, shown in Extended data Fig. 9a,b, are reported on line 279-281 in the main text, in addition to text in the figure legend and in the Methods (line 2502-2518).

All in all, the two sets of neighbourhood analyses (for MVLs and cell pairs) converge in showing that neighbouring cells are no more similar in phase than dispersed cells, or shuffled cells, in contrast to what would be expected if the sequences were travelling waves.

Furthermore, in Movie 1 it seems to me that in some of the oscillation cycles there is in fact a traveling wave across the cortical surface; the authors should analyze it like people usually analyze traveling waves, for example by calculating center-of-mass of the population-level calcium activation in each time bin, and use it to test for topographical flow of population activity across time bins. Of course this analysis must be done for each oscillation cycle separately.

Thanks for this suggestion. In order to calculate the flow of the center-of-mass (COM) we started by isolating all individual cycles and we temporally downsampled the neural data using a bin size of 5 s. We then calculated the COM for each time point and for each cycle separately (see Methods, line 2519). We visualized the topographical propagation of activity for one example cycle in Extended data Fig. 9c (temporal bin size 5s, spatial bin size 12 μm x 12 μm), following for example Zhang et al. (2018) (ref. 117). We plotted the activity on the FoV for 6 equidistant time points during the cycle, and overlaid the COM, indicated in red. We repeated this visualization when the position of the neurons was randomly shuffled. The experimental data looked very similar to the shuffled data, in that in both cases the neural activity spread all over the FoV, with successive COMs being no closer than temporally separated COMs. The COM did not describe any clear trajectory in either case. We note, however, that a quantification of COM when the activity is not localized, as in this case, might not reflect meaningful properties of the data.

To quantify the flow of COM, we calculated the cumulative distance travelled, quantified as the sum of the distances travelled by the COM between consecutive time points. Cumulative distance travelled in experimental data was compared to the tails of a distribution built by shuffling the positions of the cells in the FoV 500 times for each cycle. In Extended data Fig. 9d we show the results obtained for one example cycle, for which the distance calculated on experimental data was between the 5th and 95th percentile of the shuffled distribution. When all 421 cycles were considered, cumulative distances were similar to those obtained in the shuffled data (Extended data Fig. 9e). Only 21 had a cumulative travelled distance below the 5th percentile (21/421, at the chance level of 0.05, equal to 21/421). The results are similar when changing the temporal bin size used for the quantifications (23/421 for bin size of 1 s, 23/421 for bin size of 2 s, chance level of 0.05: 21/421). Conversely, only 19 cycles had a cumulative travelled distance above the 95th percentile of the shuffled distribution (19/421, below the chance level of 0.05, equal to 21/421). The results are also here similar when changing the temporal bin size (18/421 and 19/421 cycles, for bin size of 1 s and 2 s respectively, chance level of 0.05: 21/421). These results are presented collectively in line 281-284 of the main text, with additional text in the figure legend and in the Methods (line 2519-2534).

While in the presence of traveling waves the cumulative distance travelled by the COM might be smaller or larger than chance, depending on the specific type of travelling wave, our results show that the great majority of individual cycles are within what is to be expected by chance (the number of cycles not within this range is no larger than expected by chance). These findings thus further support the notion that the MEC population oscillation, within individual cycles of the oscillation, does not manifest as a travelling wave.

This leads me to the final, and biggest problem related to the analyses conducted by the authors regarding traveling waves, namely, that as they describe in the Methods in lines 1960-1993, they pooled the data across the entire session to examine the topography. This pooling would reveal topography if at each oscillation cycle the wave travels in the same direction, but it would fail to reveal traveling waves if at each cycle the wave travels at a

different (random) direction, as was described in many previous studies in other brain regions (some of which the authors cited). Therefore, all the analyses performed by the authors regarding traveling waves, as well as the additional analyses that I proposed above, must be repeated separately on individual oscillation cycles.

I realize that if the directions of the traveling waves would have been completely random, then the authors would likely not obtain repeated activation sequences across tens of minutes; but perhaps the directions of the waves on the cortex are variable but not entirely random, i.e. span a certain sector of angles, which could still yield the sequences they see, yet smear their analyses of topography. In fact, it is possible that those sessions where they found nice oscillations were sessions in which traveling waves traveled at a narrow set of directions (narrow sector), while those sessions where they could not detect oscillations were sessions in which traveling waves traveled at a wide (random) set of directions – and would in fact be revealed as traveling waves if the analysis was done for each oscillation-cycle separately.

In short, all the topography analyses should be repeated separately per oscillation-cycle, in addition to the analyses pooled across the entire session.

Thanks again for the insightful comments. We have now performed the cycle-by-cycle analyses suggested by the Reviewer and, all in all, don't find any evidence for travelling waves. As reported already in the above sections, performing analyses for individual cycles instead of pooling across cycles did not show correlations between anatomical distance and phase distance, and there was no coherent movement of COM across cycles. This speaks against waves moving in subsets of directions, as this should have either reduced the cumulative distance between temporally consecutive COMs within cycles (for example if the travelling wave were circular, or even linear) or increased this distance (if the traveling wave described a more complicated trajectory in the FoV, for example a spiral). At the same time, the correlation between anatomical distance and phase distance would have increased. We have added a number of new panels in Figure 3 and in two new Extended data Figures (8 and 9) to illustrate these analyses, both already described in the previous sections. Statistical analysis is presented along with the figures and in the main text. We have also made a new video (Movie 1) showing activity in the FoV across several consecutive cycles and showing how cells are anatomically organized within the periodic sequences. We hope the new video will be helpful to further visualize the lack of anatomical structure.

We want to thank the Reviewer for raising these important points, as well as for making precise suggestions. We would like to add two final remarks in conclusion. First, we completely agree with the point that traveling waves may not always propagate in the same direction. However, the notion that travelling waves move in different directions across oscillatory cycles would not be consistent with the almost fixed temporal ordering in the cells' firing across cycles of the oscillation. It would be hard to imagine how the same sequence could unfold if waves moved in different directions, and therefore activate different cells, on successive cycles. Second, we would like to remark that with our current data and analysis we don't rule out that topographical organization *could* exist at shorter time scales, for example for the theta rhythm.

Functional significance: The authors proposed in the abstract and in the discussion that these minute-scale sequences can be used as a "scaffold" for processes such as navigation and

episodic memory formation. I don't quite understand what a "scaffold" is, and therefore I could not quite imagine what is the possible usefulness of such ultraslow oscillations, in mechanistic terms.

We thank the Reviewer for pointing out the lack of clarity in the term 'scaffold'. After revisiting the text, we decided to remove this term from the manuscript, given that it is vague and subject to different interpretations. Our hypothesis was and is that ultraslow periodic sequences might serve as a basis ('template') for the formation of new patterns of neuronal activity, for example new sequences. Below we present three examples from the literature that illustrate the idea. In these examples, the template of sequences of neural activity (not periodic and not ultraslow) is thought to enable the formation of new sequences of population activity during encoding of one-time experiences, or upon learning.

1. Spontaneous neural activity in the visual cortex recorded in the absence of sensory stimulation, contains (non-periodic) sequences of activity. Parts of those sequences recur during sensory evoked experience (Carrillo-Reid et al., 2015, ref. 36), suggesting that the sensory-independent sequence may serve as a template for the formation of snippets of sequences in sensory-evoked conditions.
2. Studies of 'preplay' in hippocampal place cells have suggested that sequences expressed independently of experience may serve as templates for unique sequences generated during behavior (and replayed during rest after the behavior) (ref. 97). Sequences can be observed prior to experience ('preplay') and those sequences may serve as templates for the subsequent formation of sequences during specific experience and subsequent replay.
3. Complex sequences in the RA of zebra finches during song production are enabled by simple sequences upstream in the HVC, and plasticity between HVC and RA is thought to support song learning (ref. 98).

These examples show that sequences might facilitate and serve as a template for the formation of new sequences in the neural population during encoding of experience. In order to determine whether such parent sequences could serve as template for expression of a larger variety of dynamical regimes, i.e. beyond the original template sequence, we followed the Reviewer's advice and developed a simple model. The model has two components: (1) the template, i.e. the sequence, and (2) one output unit. The weights between these two components can be trained so that the output neuron performs two different types of activity patterns. We chose to use only one output unit in order to illustrate our hypothesis in the simplest possible scenario, although our conclusions can be generalized to neuronal populations consisting of several output units.

To test the functional role of the observed ultraslow sequences as a template in a biologically plausible scenario, we used as 'target activity pattern', or 'target response' for the output unit, a ramp of activity, because ramping activity is ubiquitous across brain regions, such as during decision making tasks in high-level visual and parietal areas (ref. 57) or foraging tasks in the lateral entorhinal cortex (ref. 58). By training our model (i.e. by changing the weights between the input units generating the template, and the output unit), we found that a ramp of activity can be generated if ongoing sequences serve as inputs. In order to generalize this idea and evaluate the model in a larger repertoire of activity patterns, we also considered a less stereotyped pattern, in which the activity varies as a function of time in a noisy manner, with the activity sometimes increasing and sometimes decreasing. In order to generate a signal with these characteristics, we used an Ornstein-Uhlenbeck process, which is widely used in the literature of computational neuroscience to emulate noisy signals in the brain.

We found that the realization of the Ornstein-Uhlenbeck process was also correctly reproduced in our model.

The model extends the knowledge of the examples above by showing that ongoing sequences can generate a variety of new activity patterns in downstream neurons. In addition, the model enabled an analysis at multiple temporal scales: We found that if the ongoing sequences are slow enough or slower than the pattern of activity that is to be generated, then the sequences can serve as a template. Otherwise, they can't. This temporal component provides an extra meaning to the template: not only does it facilitate certain responses, but it does so only at certain time scales.

The model has implications for how we envisage that stereotyped sequences in MEC may be used to create new sequences or patterns in regions that read out activity from MEC neurons participating in ultraslow oscillatory sequences. Our working hypothesis is that while the ultraslow sequences frequently may proceed periodically and without interruption under minimalistic conditions, activity during foraging in an open field is characterized more by snippets of such sequences, with frequent resetting and interruption. Preliminary data suggest that this may be the case (RL Fig. 2). Such snippets of MEC sequences, consisting of neurons firing in a fixed order, could project to the hippocampus, or other areas, and act there as a template to enable the formation of a large variety of new sequences. This hypothesis is consistent with previous work suggesting that the hippocampus can be thought of as a context-dependent sequence generator (ref. 72). A variety of hippocampal sequences could be formed by using the oscillatory sequences as a template. We showed in our model that activity patterns in downstream neurons can be generated via plasticity in connections between input and output levels. By extension, the parent sequences in MEC may give rise to new sequences downstream in the hippocampus, in reminiscence to the relationship between RA and HVC sequences during zebra finch song learning (ref. 98). In general, our model shows how having a sequential template in the MEC might enable a diversity of population activity patterns (not only sequences) in downstream regions.

We would like to remark that we decided to use an extremely simple model to illustrate our hypothesis in the simplest possible architecture. However, more sophisticated models (ref. 96), in which several networks are connected in a recurrent and feedforward manner, are also able to generate activity patterns using, as mechanism, a sequential template. If parameters of the network are precisely tuned, the network can also perform storage and replay of the learned patterns. Our conclusions are thus generalizable to neuronal populations, in which several output units could, collectively, generate specific patterns of population activity. If the Reviewer and the Editor consider that the model should have more levels of complexity, we will be happy to develop it further, for example by adding more output units or more patterns of target activity. Given the scope of this work, it is possible, however, that this would fit better for a separate computational study than a somewhat hidden Extended data Figure in the present paper.

In the revised manuscript, we have added a new section on potential functions of the ultraslow sequences, which contains the computational model (line 409-426 and a new Extended data Figure, number 14), we have rewritten the final paragraph in the Discussion (where we discuss functions of the ultraslow sequences, lines 492-500 in particular), and we have modified the concluding sentence of the Abstract to reflect the new thoughts on functional implications.

Likewise, in the discussion (line 449 onward) the authors propose that another function of these ultraslow oscillation could be to prevent different cell classes from drifting apart; I

could not understand drifting apart from what, and why does it need to be prevented; so again, it was hard for me to understand concretely what exactly is the proposed function.

Thanks for pointing out the lack of clarity. We revised the text starting on line 487 (487-492 in particular) to clarify this point. We meant that the population oscillation could allow functional cell classes, such as grid cells and head direction cells, to maintain a constant set of phase relationships between each other over time. If the phase of grid cells or head direction cells drifted independently from each other, it would rapidly lead to detrimental readouts of the position representation. These ideas are investigated in the context of grid cell modules coordination in the two papers we cite: Mosheiff and Burak 2019 and Waaga et al 2022 (refs. 94, 95).

Likewise, I was unable to understand the other functional proposals in the discussion. I suggest that the authors add a modeling chapter to their manuscript, where they would show using concrete simulations of some mathematical mechanistic model, e.g. a neural-network model, what could be the potential functional usefulness of such ultraslow waves/sequences. Without such concrete modeling effort, I find it truly difficult to understand the potential function of these oscillations.

We followed the Reviewer's suggestion and developed a neural network model that illustrates one potential function of the ultraslow sequences. As explained in the sections above, the model shows how stereotyped sequences of activity across a population of neurons may enable the formation of new slowly developing patterns of firing in downstream neurons receiving input from the neurons participating in the parent sequence. The model is explained in a new section, from lines 409 to 426, and results are presented in Extended data Figure 14. The model is a perceptron model showing that a readout unit can generate two different sets of slowly developing activity patterns: a ramp of activity – deterministic and emulating a cell with increasing activity over time (ref. 57 and 58 for examples of such cells in the brain) - and an Ornstein-Uhlenbeck process – stochastic and emulating a slow internal latent variable that the cell would keep track of. These slowly developing activity patterns emerge only if there are ongoing sequences spanning a correspondingly slow time scale upstream.

Additional major comments:

Related to the comment above: On lines 450-451 the authors propose that these ultraslow oscillations synchronize/modulate faster oscillations such as theta or gamma. This is an empirical question, which the authors can and should analyze directly: Do these ultraslow oscillations modulate the amplitude of theta or gamma oscillations in MEC?

In our calcium imaging dataset it was not possible to analyse whether ultraslow oscillations modulate the amplitude of theta or gamma because the temporal resolution of imaging data is not high enough for performing these analyses. The time scale of the theta rhythm (120 ms per cycle) is beyond what GCaMP6m fluctuations can reliably capture with a rise time of ca. 300 ms (Chen et al 2013). The Reviewer's question can be addressed with Neuropixels recordings (which provide access to the LFP at a fine temporal resolution, even for gamma rhythms) but we prefer to collect more data to address this substantial issue. However, while investigating nested rhythms is beyond the scope of the current paper (please see the communication with the Editor described at the start of this letter), we would

like to draw the Reviewer's attention to our response to comment 1.IV from Reviewer 1, in which we show preliminary data that seems to indicate that ultraslow oscillations are indeed nested with theta rhythms (RL Fig. 1; we haven't performed the analysis for gamma rhythms yet, and this will be part of future studies). While we do not include these analyses in the manuscript, we hope the present letter will be published along with the paper, giving readers an initial impression of what nesting between slow and fast rhythms might look like. Also note that in order to de-emphasize the speculation on the modulation of rhythms at different time scales in the revised manuscript, we removed the reference we originally had in the Main section (line 56).

Because there is no specific task in these experiments, perhaps these oscillations reflect slow oscillations of attention, or some other cognitive variables? This should be discussed.

While in general cognitive processes are unlikely to fluctuate with the strict periodicity of the ultraslow oscillations shown here, we do agree with the Reviewer that factors external to the entorhinal circuit, such as fluctuations in arousal-related functions mediated by neuromodulatory inputs from the brain stem, expressed for example by changes in pupil size and blood flow (refs. 74,75), could impact attentional operations. We now elaborate on this in the Discussion, in lines 467-470.

Threshold used for the oscillation score was 0.72. I could not quite understand why 0.72. Please show that the results in the paper (in all figures) are robust to other choices of the threshold.

Thank you for pointing out the lack of clarity in the justification for the choice of threshold. In Extended data Fig. 5d we show that the distribution of oscillation scores is bimodal, with a group of sessions having very low oscillation scores ($n = 12$, illustrated in red in the figure), and a second group having high scores ($n = 15$, illustrated in green). The threshold that we used (0.72) corresponded to the smallest oscillation score within the group with high scores. Because the data was bimodal, any threshold between 0.27 and 0.72 would have led to the same results presented in the manuscript, indicating that our findings are robust to other choices of thresholds. We now comment on this in the Methods section, in lines 2191-2193.

Line 1640-1646: Shuffling: Why did the authors randomly shuffled the time-bins for each cell? This destroys the normal physiological temporal structure of activity in each cell. It seems to me that a much better shuffling procedure (and much more common in the field) would be to rigidly and circularly shift the activity of each cell, making sure to shift by a different random amount for each cell.

We thank the Reviewer for raising this point. We decided originally not to circularly shuffle the calcium events for individual cells because this would preserve the periodicity each cell shows. In a new figure, Extended data Fig. 3, we now demonstrate both types of shuffling: circular (oscillation is preserved) and shuffling where inter-event trials are disrupted (oscillation is disrupted), as before. We show the autocorrelation and the power spectral density for one example cell of each of the 15 oscillatory sessions – both experimental data and the two types of shuffled data (Extended data Fig. 3a-d), as well as the mean across the cell population (Extended data Fig. 3e,f).

At the single cell level, we now mention in the manuscript, in line 79, that when using a new shuffling procedure that preserves a subset of the inter calcium event intervals (see comment from the

Reviewer, below, and Methods, line 1918), 91% of the cells present oscillatory activity (see also response to third comment below).

At the network level, in Extended data Fig. 4c, left column, middle row, we now show the rasterplot obtained after circularly shuffling the calcium activity of each cell separately, and then applying the PCA method for sorting the population activity. This new plot shows that when the activity of individual cells is circularly shuffled the sequences are no longer detectable.

Fig. 1b, right: Given the wide heterogeneity of oscillation frequencies for different neurons (seen in the right panel), how is it possible that a coherent population oscillation emerges? This panel has left me quite perplexed, as it seems to contradict the rest of the paper.

We agree with the Reviewer that the apparent continuum of frequencies observed in that panel (repeated here in RL Fig. 11, for convenience), raised the question of how a coherent population oscillation can emerge in the network. However, when looking at the distribution of frequencies at which the PSD of single-cell autocorrelations peaked (RL Fig. 12) in that session, the apparent continuum is reduced to two well defined maxima: one between 0.0055 and 0.0085 Hz (encompassing 301 cells out of 484, 62%), and a second one between 0.0155 and 0.0185 Hz (41 cells out of 484, 8%). The first maximum coincides with the population oscillation frequency (0.0066 Hz). The second maximum contains the first harmonic of the frequencies in the first maximum. This discreteness in the frequencies is masked in the visualization of the stacked autocorrelations, where a continuum is hinted at by the way the cells are sorted (according to increasing frequency).

RL Figure 11: Stacked autocorrelations of single-cell calcium activity for the example session shown in Fig. 1b. Each row is the z-scored autocorrelation of one cell's calcium activity. Neurons are sorted according to the frequency at which the PSD peaks.

RL Figure 12: Histogram of frequencies at which the PSD of single cells' autocorrelations peaked ($n = 484$, same session as in Fig. 1b and Fig. 2b).

That said, we note that the maxima in the distribution of RL Fig. 12 have a certain width. These widths are associated to the variability in the exact frequency of single cells. For example, some PSDs peaked at a frequency of 0.0066 Hz, while others did so at a frequency of 0.0075 Hz. However, in many cases the PSDs were wide enough to exhibit high power in neighboring frequencies, as shown in RL Fig. 13, providing further support to the frequencies being rather clustered among a subset of values, with some slight variability around those values.

RL Figure 13: Similar to Fig. 1c, but for another example cell in the same recording. The PSD peaks at 0.0264 Hz (red dashed line in the left panel), equal to 4 times 0.0066 Hz, the frequency of the population oscillation. The peak is much wider than in (c), corresponding to a weaker oscillatory pattern in the autocorrelation. These plots were previously presented in the manuscript, but we have now removed them to instead include example cells from Neuropixels recordings (see comments from Reviewer 1 and 2).

When all cells were analyzed ($n = 6231$ cells pooled across 15 oscillatory sessions), we found that, as expected, for approximately half of the data the oscillatory frequency at the single-cell level was very similar to the frequency at the population level. This finding points to a small variability in the frequency of single cell activity, as expected in presence of recurring sequences. This data is presented in Extended data Fig. 7c, where we show the distribution of single-cell oscillatory frequency divided

by the population oscillation frequency of the session. In this figure, a value of 1 indicates that single-cell and population frequency coincide.

Since the frequencies tend to be similar across cells and rather clustered around one value (and its harmonics), and because the panel of stacked autocorrelations sorted with increasing frequency (RL Fig. 11) suggests a continuum that we didn't find in the quantification of our data, we decided to remove the questioned panel from Fig. 1, as well as the ones corresponding to PaS and VIS from Fig. 6, and instead let Extended data Fig. 7c speak to the variations in single-cell frequency. We have also added text in the Methods (line 1946) to better explain the variations in single-cell frequencies.

Fig. 1c,d,e: The authors show only 3 examples in the entire paper, and should add many more examples in the Extended Data Figures. In addition, the authors must add shuffles in these examples (and other examples they will add), so that the reader can see if the data is indeed more periodic than shuffles.

We agree with the Reviewer that in the original manuscript we presented too few examples. We have now made a new Extended data figure (number 3) with one example cell from each oscillatory session. We include plots of autocorrelations and power spectra densities for (i) experimental data corresponding to one example cell (in black), (ii) shuffled data obtained by circularly shuffling the calcium events of the analysed cell (in blue), (iii) shuffled data obtained by randomly shuffling the calcium events of the analysed cell (in red), which destroyed the inter calcium event intervals. In addition, we also present, for each oscillatory session, the mean autocorrelation and the mean power spectral density calculated over all cells in the session. These plots were made for both experimental (in black) and shuffled data (in red), as requested by the Reviewer. In addition, now we also include 2 example cells from Neuropixels recordings in Fig. 1. We also refer to these new plots in the main text, in line 80.

As we point out above and show in Extended data Fig. 3a, circularly shuffling the deconvolved calcium events does not destroy the oscillations, since this shuffling procedure preserves the inter calcium event intervals, which both enable the oscillations, and define the frequency of the oscillation. Yet, for completeness we decided to include the circular shuffling too. In order to have a reference of what should be expected by chance, we also considered the second shuffling procedure, where we shuffled the calcium events without preserving the inter calcium event interval.

We hope that this new figure illustrates what single cells activity looks like across oscillatory sessions. Thank you for this suggestion.

On a related note: What is the percentage of cells with spectral peak that protrudes above 95% or 99% of the shuffles (after Bonferroni-correcting for the number of frequency bins)?

We thank the reviewer for raising this question. In order to perform this analysis we considered two extreme and opposite shuffling procedures (see also responses to comments above): On the one hand, given that circularly shuffling the data preserves all inter calcium events, taking this approach would preserve the shape and the position of the peak in the power spectral density (PSD) calculated on experimental data. On the other hand, destroying the inter calcium event intervals by assigning a random position to each calcium event in the time series would lead to a flat PSD (Extended data Fig. 3d). In the latter approach, all cells would be trivially classified as oscillatory. To bridge these two approaches we developed a new shuffling procedure. For each cell we divided the calcium activity

vector into n epochs of length W , with $n = T/(W \cdot SF)$, where T is the total number of time bins sampled at a frequency $SF = 7.73$ Hz (i.e. bin size = 129 ms). We next shuffled those epochs (and preserved the ordering of the time bins within each epoch). This method preserved the inter calcium event interval, but at the same time disrupted the periodicity. In the limit where $W = 129$ ms, this method coincides with shuffling all calcium events without preserving the inter calcium event intervals; in the limit where $W = T/SF$, this method is equivalent to circularly shuffling the data. For each of the 200 shuffled realizations we calculated the PSD. Following the Reviewer's suggestion, we calculated the fraction of cells for which the peak of the PSD was above the 95th percentile of a shuffled distribution built with the values of the PSDs calculated on shuffled data at the frequency at which the PSD computed on experimental data peaked. Because we determined significance for the peak of the PSD (as the Reviewer suggested), we didn't apply a Bonferroni correction.

Here we present the results for 5 different epoch lengths.

$W = 1$ s: 6226 oscillatory cells out of 6231 (99%)

$W = 10$ s: 6153 oscillatory cells out of 6231 (99%)

$W = 20$ s: 5695 oscillatory cells out of 6231 (91%)

$W = 50$ s: 4642 oscillatory cells out of 6231 (74%)

$W = 100$ s: 3521 oscillatory cells out of 6231 (56%)

When W is below the typical duration of the sequences ($W < 50$ s), the great majority of cells are classified as having a peak in the PSD. As expected, when W is similar to the duration of the sequences ($W \geq 50$ s), the fraction of oscillatory cells quickly drops. This fraction is no longer significantly above a chance level of 5%.

We now comment on this in line 79 of the Results section and in lines 1918 to 1945 of the Methods.

Extended Data Fig. 3d: Even though these examples are supposed to show the 15 oscillatory sessions, I see that about half of these "oscillatory" sessions do not show clear ring structure in the plot of PC1 vs. PC2. For example, a ring is missing in Mouse 59914 session 17, Mouse 60355 session 18, Mouse 60585 session 17, etc. Why is that the case? Also: if there is no ring structure here, indicating lack of an oscillation, then how did the authors extract the 'phase' or angle of the oscillation, which they used for some of their analyses and indices?

We agree with the Reviewer that in those sessions the ring is less evident. This is because there were more variations from sequence to sequence, which resulted from the rings that corresponded to each sequence not completely overlapping in the PC1 vs PC2 plane. Please note that recovering rings with PCA is already challenging due to PCA being a linear method. We were therefore positively surprised that the data was so striking that even with PCA we could recover the circular topology of the manifold. Using a non-linear method would have helped in visualizing the ring (as in Fig. 2e right), but we decided not to do this because non-linear methods require more fine-tuning and are usually harder to interpret. For all sessions that were classified as oscillatory sessions we extracted the phase as the arctangent of the ratio between the neural activity projected onto PC1 and PC2. This quantity is always defined (see sentence added in the Methods, line 2122) and in the case of the oscillatory sequences it tracks the progression of the sequences. We have now referred to these distortions in lines 139 to

142, and we have justified the choice of PCA in the Methods (lines 2112-2113).

Minor comments:

Lines 250-251: "Extended Data Figs 6j,6k,6l": this should probably be Extended Data Figs 7j,7k,7l.

Line 286: "three of more ensembles": should be "or".

Corrected, thank you.

Line 809, one sample t-test: Why t-test? You plot it as median (boxplot), so a Wilcoxon test on the median seems more appropriate.

We agree with the Reviewer. We changed the t-test by a Wilcoxon signed rank test and tested for the median of the distribution of percentages of locked neurons. This information is now provided in the legend of Fig. 3b.

Lines 832-833 (figure legend of Fig. 3f): why the tildes (\sim), in ~ 0 rad and $\sim \pi$ rad? Why approximately? The authors should provide the exact definition, not approximate definition. The same goes for the figure legend of Fig. 3h: what is the exact definition?

We decided to remove this analysis from the paper and replace it by the more thorough analysis suggested by the Reviewer (see our comments to the section of the RL on traveling waves).

Line 867: "Time bins colored in blue": this seems more like aquamarine or some other greenish color, not blue.

We agree with the Reviewer and we changed the color to aquamarine (legend of Fig. 5a).

Line 915: "Probability is shown on a log-scale": looking at the figure, the y-axis seems actually to be on a lin-scale.

While in the original plot the probability was indeed shown on a log-scale, we agree with the Reviewer that it didn't look like that. We have now added more ticks to indicate the logarithmic nature of the scale.

Line 1048, "lower than 10% of 2π ": I can see actually examples of the phase going down by more than 10%, e.g. at $t=250$ s.

We thank the Reviewer for the thorough examination of our figures. Indeed, at $t = 250$ s there is a jump in the phase of the oscillation that is larger than 0.63 rad (10% of 2π). This jump divided what is perceived as one sequence in Extended data Fig. 6a into two sequences, each of which entrains a subset of the network. In the original version of this figure we used a very thick cyan line to color the segments of the phase that belonged to individual cycles, which gave the impression that these segments belonged to only one sequence. In RL Fig. 14 we show the same sequence with a thinner

line, where it can be noticed the discontinuity indicated by the Reviewer. We also changed the line width in the new version of the figure (Extended data Fig. 6a).

RL. Figure 14: Snippet of session illustrated Extended data Fig. 6a. Top: Rasterplot. Bottom: Phase of the oscillation. By using a thinner cyan line we show how the phase discontinuity breaks the sequence down into two sub-sequences.

Line 1122: “The preferred phases cover the entire range of phases...”: This was not the case in 4 of the sessions, in particular all the 3 sessions in the rightmost column of this panel, as well as Mouse 60585 session 18. What does this mean?

We thank the Reviewer for raising this point. We agree with the Reviewer that from visual inspection it seems that in some of the sessions the preferred phases might not cover the entire range of phases, from $-\pi$ to π . To quantitatively probe this, we identified the smallest and largest preferred phase in each of the sessions (Table 1). We found that in each of the sessions the smallest preferred phase was very close to $-\pi$, and the largest preferred phase was very close to π . This information shows that the extremes of the range (i.e. $-\pi$ and π) are relatively well sampled in our data. In Extended data Fig. 7d we further show that in each session the preferred phases were distributed more uniformly than expected by chance. Altogether, these two pieces of results suggest that the range $[-\pi, \pi)$ is uniformly sampled in each oscillatory session. To clarify this point, we have replaced the sentence ‘*The preferred phases cover the entire range of phases*’ by the range of smallest and largest preferred phases.

Animal #	Session #	Smallest preferred phase (rad)	Largest preferred phase (rad)
60355	18	-3.12	3.10
60355	19	-3.11	3.14
60355	20	-3.14	3.13
60585	16	-3.14	3.11
60585	17	-3.11	3.13
60585	18	-3.12	3.10
60584	16	-3.13	3.13
60584	17	-3.14	3.14
60584	18	-3.14	3.14
60584	19	-3.11	3.13
59914	16	-3.13	3.14
59914	17	-3.13	3.14

59914	18	-3.11	3.11
59914	19	-3.14	3.12
59914	21	-3.13	3.08

Table 1: Information about smallest and largest preferred phased in each of the 15 oscillatory sessions.

Line 1436: twice “with with”.

Corrected, thank you.

Lines 1510-1528: The authors write that thy used Dil to mark the corner of the prism. But in other places in the paper they wrote they used a different substance, DiL. Please clarify across the paper which substance was used, Dil or DiL.

We thank the Reviewer for identifying this mistake. Dil and DiL referred to the same substance. We have now unified the terminology and refer to it always as Dil (both in the Methods and in the legends of Extended data Fig. 1 and 11).

Line 1938: “the number calcium events”: should be “the number of calcium events”.

Corrected, thank you.

Fig. 3f,h: Fonts are too small on the x-axis: please increase the font size.

These figures are no longer included in the manuscript, as they were replaced by the new analyses suggested by the Reviewer.

Fig. 6i, left panel: Is there really a significant difference between the red and blue lines in the first three error bars (n = 2, 3 and 4 ensembles)? It says ‘’, ‘**’ and ‘***’, but looking at the error bars, the difference does not seem significant to me. Am I missing something?*

We agree with the Reviewer that the error bars corresponding to shuffled data and recorded data tend to overlap. The degree of overlap, however (especially for ensembles 2 and 3), seems to be magnified due to the log scale in the plot (Fig. 6i). In order to test for significance and determine whether the probability of sequential activation in recorded data was significantly above shuffled data, we ran a one-tailed Wilcoxon rank-sum test. Here are the obtained p-values:

2 ensembles: $p = 0.9981$ – n.s. (i.e. the probability of finding 2 ensembles that are sequentially activated is larger in shuffled data than in recorded data).

3 ensembles: $p = 0.0058 > 0.001$ (**)

4 ensembles: $p = 0.0006 < 0.001$ (***)

5 ensembles: $p = 0.0359 < 0.05$ (*)

6 ensembles: $p = 0.0007 < 0.001$ (***)

7 ensembles: $p = 0.0112 < 0.05$ (*)

We hope this information clarifies how significance was reported. We believe that the confusion comes from the fact that in the first version of the manuscript we didn't report the statistics for 2 ensembles. In the legend of Fig. 6 we have now added the statistics for 2 ensembles, both for PaS and VIS. We didn't include all p values presented above to prevent the legend from becoming too

long and hard to read. However, we would be happy to include this information if the Reviewer considers it important for a correct interpretation of the figure.

References:

- Zong, W., Obenaus, H. A., Skytøen, E. R., Eneqvist, H., de Jong, N. L., Vale, R., ... & Moser, E. I. (2022). Large-scale two-photon calcium imaging in freely moving mice. *Cell*, *185*(7), 1240-1256.
- Jacobsen, R. I., Nair, R. R., Obenaus, H. A., Donato, F., Slettmoen, T., Moser, M. B., & Moser, E. I. (2022). All-viral tracing of monosynaptic inputs to single birthdate-defined neurons in the intact brain. *Cell reports methods*, *2*(5).
- Pachitariu, M., Stringer, C., & Harris, K. D. (2018). Robustness of spike deconvolution for neuronal calcium imaging. *Journal of Neuroscience*, *38*(37), 7976-7985.
- Berens, P., Freeman, J., Deneux, T., Chenkov, N., McColgan, T., Speiser, A., ... & Bethge, M. (2018). Community-based benchmarking improves spike rate inference from two-photon calcium imaging data. *PLoS computational biology*, *14*(5), e1006157.
- Ray, S., Naumann, R., Burgalossi, A., Tang, Q., Schmidt, H., & Brecht, M. (2014). Grid-layout and theta-modulation of layer 2 pyramidal neurons in medial entorhinal cortex. *Science*, *343*(6173), 891-896.
- Tang, Q., Burgalossi, A., Ebbesen, C. L., Ray, S., Naumann, R., Schmidt, H., ... & Brecht, M. (2014). Pyramidal and stellate cell specificity of grid and border representations in layer 2 of medial entorhinal cortex. *Neuron*, *84*(6), 1191-1197.
- Lee, H. S., Ghetti, A., Pinto-Duarte, A., Wang, X., Dziejczapolski, G., Galimi, F., ... & Heinemann, S. F. (2014). Astrocytes contribute to gamma oscillations and recognition memory. *Proceedings of the National Academy of Sciences*, *111*(32), E3343-E3352.
- Baird-Daniel, E., Daniel, A. G., Wenzel, M., Li, D., Liou, J. Y., Laffont, P., ... & Schwartz, T. H. (2017). Glial calcium waves are triggered by seizure activity and not essential for initiating ictal onset or neurovascular coupling. *Cerebral cortex*, *27*(6), 3318-3330.
- Chen, T. W., Wardill, T. J., Sun, Y., Pulver, S. R., Renninger, S. L., Baohan, A., ... & Kim, D. S. (2013). Ultrasensitive fluorescent proteins for imaging neuronal activity. *Nature*, *499*(7458), 295-300.

Reviewer Reports on the First Revision:

Referees' comments:

Referee #1 (Remarks to the Author):

I thank the authors for the remarkably thorough and extensive revisions to their manuscript. Most of my queries have been answered, and I understand the rationale behind leaving some answers for later, given the study's richness. I thus accept that some questions remain unanswered and look forward to following studies.

I have a few remaining very small questions and remarks, which I list below.

Line 53. "We directed our search": the writing seems to suggest that the authors were looking for such a phenomenon. Was it so, or did it rather start as an unexpected observation? If so, my personal preference would be to present it as such. As the grid cell discovery certainly proved, what we don't expect is usually more interesting!

Lines 83-88. Thank you for these new data, which represent a very substantial amount of work. I found them very interesting, not so much for the fact that they support the Ca imaging results (they do, but do not seem to be a perfect match), but for the apparent differences with the Ca results. For example, the frequency of the "periodic sequences" seems very significantly higher than for Ca imaging results (the time bases in Fig 2a-c and d are different by a factor of 2). Is it noise as suggested, or could it be something else, such as higher detectability of events?

Also, the sequences seem less smooth than with Ca imaging.

I wonder whether it might not be useful to mention the most noteworthy differences in the results section or the discussion, in case they might eventually prove to be real.

Line 429: "periodic sequences": the shortcut is used throughout, but could potentially be misinterpreted. Would "sequences that repeat periodically", stated at least once at the beginning of the discussion, help clarify?

Once again, many thanks for all the added experimental and analytical work. This is a beautiful study.

Referee #2 (Remarks to the Author):

The revision has addressed my comments. Overall, the addition of neuropixel data, the extension of the analysis to the parasubiculum and the visual cortex, and further quantitative testing of oscillations strengthened the work greatly. It is a very impressive work and indicates the presence of cortex-wide ultra-slow oscillatory activity that can be identified by large-scale neuronal assembly recordings.

Jozsef Csicsvari

Referee #3 (Remarks to the Author):

I applaud the authors for the very extensive work that they have done in this major revision. Based on the (multiple) new analyses that they conducted, I am now convinced that the oscillatory sequences cannot be explained via traveling waves, which was my biggest concern. I also appreciate the clarifications regarding the possible functional roles of these ultraslow sequences (including the new modeling effort) – which I believe increase the biological importance of this study. Together with the other major work that the authors did for this revision, I think that the manuscript was now substantially improved – and I now believe that they report here a truly new phenomenon, which may have interesting biological implications. I am therefore happy to recommend publication.

Author Rebuttals to First Revision:

To all Reviewers:

Thanks to all Reviewers for their encouraging feedback and kind words. Below we provide a point-to-point response to the Reviewer and Editor comments (comments in *italics*).

General comment to the Editors:

Your manuscript, "Minute-scale oscillatory sequences in medial entorhinal cortex", has now been seen by our referees, and in the light of their advice I am delighted to say that we can in principle offer to publish it.

Thank you very much for the willingness to publish our paper. We are delighted!

The typical length of an 8-page article with 5 modest (quarter-page) display items is 4300 words. [...] I can give you 12 ED figs in order to assist with the process of trimming this down, but the ED should also be well-organized and used efficiently. Also, the reference section is much too long, so please be sure that all references that pertain to the methods are included with the methods section as opposed to the regular reference list of the main paper.

We thank the Editor for the thorough information on how to format the manuscript. After trimming the manuscript down very extensively, the document now has 4551 words, 50 references, 5 main Figures and 12 Extended data Figures (EDF) (the version we submitted in early August had 9562 words, 103 references, 6 main Figures and 14 Extended data Figures). Reducing the paper by this extent include the following actions:

- We had to remove this sentence from the Discussion (which made reference to one of the points raised by Reviewer 1 in the first round of revision): *Ultraslow oscillations might also be modulated by glial networks through the regulation of astrocyte-dependent synaptic connectivity*^{76,77}. We feel that removing this sentence is justified, given that it is entirely speculative and the paragraph on mechanisms was insufficiently focused. We are happy to put it back (or into methods) if you feel otherwise. We have not made a supplementary file.

- We removed figure panels that were not informative, or panels that were duplicates of other figures. We didn't remove any data. We no longer include figures for the analysis on differences in preferred phases in small neighborhoods (previously shown in EDF9 a,b), because this analysis, relevant to travelling waves, revealed the same findings, with very similar methods, as the analysis previously reported in Extended Data Fig. 8b,c. All quantifications of the duplication are still reported in the text, however (legend of EDF 8) - they are just not shown in a figure.

- We moved a lot of methods description and some of the quantification that were originally presented in the main text and in the legends to the legends of Extended data Figures or to the Methods section.

We have produced source data files as requested for all graphical source data where this was possible (not including highly pixelated two-dimensional color maps or high-resolution raster plots, which all include data of an amount far exceeding the 30 MB limit for graphical source data). Please let us know if you think there is a way to present such data in Nature-compatible excel-sheet format.

The legend of Fig. 3 has 399 words, which exceeds the 300-word count limit. We couldn't compress the legend further without making it hard to interpret for the reader.

We will provide the DOIs during the proofs.

Reviewer comments:

REVIEWER #1

I thank the authors for the remarkably thorough and extensive revisions to their manuscript. Most of my queries have been answered, and I understand the rationale behind leaving some answers for later, given the study's richness. I thus accept that some questions remain unanswered and look forward to following studies.

I have a few remaining very small questions and remarks, which I list below.

We thank the Reviewer for the constructive feedback in the previous round, and for the positive comments in this round of revision.

Line 53. "We directed our search": the writing seems to suggest that the authors were looking for such a phenomenon. Was it so, or did it rather start as an unexpected observation? If so, my personal preference would be to present it as such. As the grid cell discovery certainly proved, what we don't expect is usually more interesting!

We thank the Reviewer for bringing this up. In order to deemphasize the active search for ultraslow oscillations in the motivation of the study, in line 52 of the Introduction we now write 'To rule out variations in external stimuli as sources of modulation, we let head-fixed mice run on a rotating wheel for 30 or 60 minutes, in darkness and with no scheduled rewards'.

Lines 83-88. Thank you for these new data, which represent a very substantial amount of work. I found them very interesting, not so much for the fact that they support the Ca imaging results (they do, but do not seem to be a perfect match), but for the apparent differences with the Ca results. For example, the frequency of the "periodic sequences" seems very significantly higher than for Ca imaging results (the time bases in Fig 2a-c and d are different by a factor of 2). Is it noise as suggested, or could it be something else, such as higher detectability of events? Also, the sequences seem less smooth than with Ca imaging. I wonder whether it might not be useful to mention the most noteworthy differences in the results section or the discussion, in case they might eventually prove to be real.

We agree with the Reviewer's observations. The sequences look less clear in Neuropixels recordings. We hypothesize that the differences between both techniques are due to differences in cell sampling, with Neuropixels probes targeting multiple layers and from more ventral locations. We also observed many interneurons in the Neuropixels recordings (firing rate > 70 Hz), and the dynamics of these cells is likely to be better accounted for in Neuropixels than in the calcium imaging data. Finally, there could also be differences in the training and behaviour of the animals.

We now include information about the duration of the sequences in the legend of EDF4, lines 1986 and 1991. We also added a comment on the differences between Neuropixels and calcium imaging sequences in the Discussion, lines 228-230. Finally, we also comment briefly on these differences in the Results section, lines 102-104

Line 429: “periodic sequences”: the shortcut is used throughout, but could potentially be misinterpreted. Would “sequences that repeat periodically”, stated at least once at the beginning of the discussion, help clarify?

We thank the Reviewer for this suggestion, which is now addressed in line 225 of the Discussion.

Once again, many thanks for all the added experimental and analytical work. This is a beautiful study.

We thank the Reviewer their kind words and for their enthusiasm about our findings and manuscript.

REVIEWER #2

The revision has addressed my comments. Overall, the addition of neuropixel data, the extension of the analysis to the parasubiculum and the visual cortex, and further quantitative testing of oscillations strengthened the work greatly. It is a very impressive work and indicates the presence of cortex-wide ultra-slow oscillatory activity that can be identified by large-scale neuronal assembly recordings.

Jozsef Csicsvari

We thank the Reviewer for his positive feedback and enthusiasm about our findings, and we appreciate his comments and suggestions from the previous round of revision.

REVIEWER #3

I applaud the authors for the very extensive work that they have done in this major revision. Based on the (multiple) new analyses that they conducted, I am now convinced that the oscillatory sequences cannot be explained via traveling waves, which was my biggest concern. I also appreciate the clarifications regarding the possible functional roles of these ultraslow sequences (including the new modeling effort) – which I believe increase the biological importance of this study. Together with the other major work that the authors did for this revision, I think that the manuscript was now substantially improved – and I now believe that they report here a truly new phenomenon, which may have interesting biological implications. I am therefore happy to recommend publication.

We thank the Reviewer for the very kind words and for the feedback from the previous round, which strengthened the quality of our paper.